# The impact of imputation quality on machine learning classifiers for datasets with missing values

Tolou Shadbahr [1,23], Michael Roberts [2,3,23✉], Jan Stanczuk[2,23], Julian Gilbey [2,23], Philip Teare[3,23], Sören Dittmer[2,4], Matthew Thorpe[5], Ramon Viñas Torné[6], Evis Sala [7], Pietro Lió [5], Mishal Patel[3,8], Jacobus Preller [9], AIX-COVNET Collaboration*, James H. F. Rudd [10], Tuomas Mirtti [1,11,12], Antti Sakari Rannikko[1,12,13], John A. D. Aston[14], Jing Tang [1] & Carola-Bibiane Schönlieb[2]

## Abstract

**Background** Classifying samples in incomplete datasets is a common aim for machine learning practitioners, but is non-trivial. Missing data is found in most real-world datasets and these missing values are typically imputed using established methods, followed by classification of the now complete samples. The focus of the machine learning researcher is to optimise the classifier's performance.

**Methods** We utilise three simulated and three real-world clinical datasets with different feature types and missingness patterns. Initially, we evaluate how the downstream classifier performance depends on the choice of classifier and imputation methods. We employ ANOVA to quantitatively evaluate how the choice of missingness rate, imputation method, and classifier method influences the performance. Additionally, we compare commonly used methods for assessing imputation quality and introduce a class of discrepancy scores based on the sliced Wasserstein distance. We also assess the stability of the imputations and the interpretability of model built on the imputed data.

**Results** The performance of the classifier is most affected by the percentage of missingness in the test data, with a considerable performance decline observed as the test missingness rate increases. We also show that the commonly used measures for assessing imputation quality tend to lead to imputed data which poorly matches the underlying data distribution, whereas our new class of discrepancy scores performs much better on this measure. Furthermore, we show that the interpretability of classifier models trained using poorly imputed data is compromised.

**Conclusions** It is imperative to consider the quality of the imputation when performing downstream classification as the effects on the classifier can be considerable.

## Plain language summary

Many artificial intelligence (AI) methods aim to classify samples of data into groups, e.g., patients with disease vs. those without. This often requires datasets to be complete, i.e., that all data has been collected for all samples. However, in clinical practice this is often not the case and some data can be missing. One solution is to 'complete' the dataset using a technique called imputation to replace those missing values. However, assessing how well the imputation method performs is challenging. In this work, we demonstrate why people should care about imputation, develop a new method for assessing imputation quality, and demonstrate that if we build AI models on poorly imputed data, the model can give different results to those we would hope for. Our findings may improve the utility and quality of AI models in the clinic.

[1] Research Program in Systems Oncology, Faculty of Medicine, University of Helsinki, Helsinki, Finland. [2] Department of Applied Mathematics and Theoretical Physics, University of Cambridge, Cambridge, UK. [3] Data Science & Artificial Intelligence, AstraZeneca, Cambridge, UK. [4] ZeTeM, University of Bremen, Bremen, Germany. [5] Department of Mathematics, University of Manchester, Manchester, UK. [6] Department of Computer Science and Technology, University of Cambridge, Cambridge, UK. [7] Department of Radiology, University of Cambridge, Cambridge, UK. [8] Clinical Pharmacology & Safety Sciences, AstraZeneca, Cambridge, UK. [9] Addenbrooke's Hospital, Cambridge University Hospitals NHS Trust, Cambridge, UK. [10] Department of Medicine, University of Cambridge, Cambridge, UK. [11] Department of Pathology, University of Helsinki and Helsinki University Hospital, Helsinki, Finland. [12] iCAN-Digital Precision Cancer Medicine Flagship, Helsinki, Finland. [13] Department of Urology, University of Helsinki and Helsinki University Hospital, Helsinki, Finland. [14] Department of Pure Mathematics and Mathematical Statistics, University of Cambridge, Cambridge, UK. [23]These authors contributed equally: Tolou Shadbahr, Michael Roberts, Jan Stanczuk, Julian Gilbey, Philip Teare. *A list of authors and their affiliations appears at the end of the paper. ✉email: michael.roberts@maths.cam.ac.uk

Datasets with missing values are ubiquitous in many applications and feature values can be missing for many reasons. This may be due to incomplete or inadequate data collection, corruption of the dataset, or differences in the recording of data between different cohorts or sources. Furthermore, these reasons do not always remain constant or consistent across data sources. Each of these sources of missingness can also introduce different types of missingness, e.g., missing completely at random (MCAR), missing at random (MAR) and missing not at random (MNAR)[1,2].

Training machine learning classification models is non-trivial if the underlying data is incomplete, as many methods require complete data[3]. Simply excluding incomplete samples could lead to both a large reduction in statistical power and also risks introducing a bias if the cause of the missingness is related to the outcome. The typical solution to this problem is to follow a two-stage process, firstly imputing the missing values and then using a machine learning method to classify the now-complete dataset. There is extensive literature discussing imputation methods, most of which is focussed on the imputation of data at a single time-point (as we focus on). However, more nuanced scenarios have also been considered, in particular for the imputation of long-itudinal data[4,5], imputation of decentralised datasets using a federated approach[6] and imputation of variables which have a hierarchical (multi-level) relationship to one another[7].

Despite the two-stage methods forming the bedrock of approaches to making predictions from incomplete data[8–11], it is still unclear which approaches perform 'best' and how this should be measured. In particular, it is unclear how the classification model's performance is influenced by the underlying imputation method and how performance is affected by levels of missingness in the data. When fitting a model to incomplete data using a two-stage approach, the primary aim of many studies is to optimise the downstream classification performance, rather than carefully assessing if the imputed data reflects the underlying feature distribution. However, the latter is profoundly important, as a model trained using poorly imputed data could assign spurious importance to particular features. When we artificially induce missingness into a complete dataset, the quality of an imputation method is typically measured by comparing the imputed values with the ground truth using common metrics such as the (root/normalised) mean square error (MSE)[3,6,12–20], mean absolute (percentage) error[3,14–16] or (root/normalised) square deviance[4,6,12]. However, as other authors have commented[7], optimal results for some metrics are achieved even when the distribution of the imputed data is far from the true distribution, see Supplementary Fig. 1.

There are many different discrepancy scores for measuring imputation quality considered in the literature, with a distinction made for those used with categorical data and those used for continuous data. For example, for categorical data, one could consider the proportion of false classifications and Cramér's V metric. For continuous data, there are many additional metrics and discrepancy scores such as the two-sample Kolmogorov–Smirnov statistic, the (2-)Wasserstein distance (Mallows' $L^2$) and the Kullback–Leibler divergence. In particular, the paper of Thurrow et al.[18] considers imputation quality in detail, reporting many of these discrepancy scores, on a feature-by-feature basis to identify distributional differences between the imputed and true missing values.

In this paper, we systematically and carefully address two open research questions. Firstly, for two-stage methods, does the optimal configuration (e.g., imputation method and classifier combination) for a particular dataset, ensure optimal classifying performance for the incomplete data? To address this, we evaluate the performance of several two-stage methods for classifying incomplete data. Using multi-factor ANOVA analysis, we quantify how the downstream classification performance is influenced by the imputation method, classification method and data missingness rate. Secondly, how faithfully do different data imputation methods reproduce the distribution of the underlying dataset? This is a crucial question and requires careful and extensive evaluation. We assess imputation quality using standard discrepancy scores, such as RMSE, MAE and the coefficient of determination ($R^2$), and feature-wise discrepancy scores using the Kullback–Leibler divergence, Kolmogorov–Smirnov statitstic and Wasserstein distance. In addition, we introduce a class of discrepancy scores inspired by the sliced Wasserstein distance[21] for evaluating how well-imputed data faithfully reconstructs the overall distribution of feature values. We demonstrate how the new proposed class of measures is more appropriate for assessing imputation quality than existing popular discrepancy statistics.

Moreover, we explore the link between imputation quality and downstream classification performance and show the remarkable result that a classifier built on poor imputation quality can actually give a satisfactory downstream performance. We postulate that this could be due to the ability of powerful classifier methods to overcome issues with the imputed data (as imputation leads to noise injection into the dataset). Training machine learning-based models on such noisy data can be viewed as a form of data augmentation (known to improve the gen-eralisability and robustness of the models[22]). The stability of different imputation methods is explored and we found that the algorithms consisting of neural network components can give highly variable imputation results. In addition, in our experiments, we find that the popular discrepancy measures used to assess imputation quality are uncorrelated from the downstream model performance, whereas the measures which consider distributional discrepancies do show a correlation. Finally, we demonstrate how high-performing classification models trained using poorly imputed data assign spurious importance to particular features in the dataset. We release a codebase to the community, along with the datasets considered, in a GitLab repository[23]. This provides a framework for practitioners to allow for easy, reproducible benchmarking of imputation method performance, for evaluating a wide range of classifiers along with assessing imputation quality in a completely transparent way.

## Methods
In this section, we briefly introduce the datasets used in the study, the benchmarking exercise and the methods for assessing imputation quality.

**Datasets**. In this study, we focus on five datasets denoted as Breast Cancer, MIMIC-III, NHSX COVID-19, Simulated (N) and Simulated (N,C). The first three of these are derived from real clinical datasets, while the final two are synthetic. The MIMIC-III, Simulated (N) and Simulated (N,C) datasets are complete (i.e., do not contain missing values), giving us control over the induced missingness type and rate. The Breast Cancer and NHSX COVID-19 datasets exhibit their natural missingness. Permission has been obtained from the data originators, where required to use the clinical data in this study.

Simulated (N) is a synthetic dataset created using the `scikit-learn`[24] function `make_classification`, giving a dataset with 1000 samples, 25 informative features, no redundant features and no useless features. This allows us to perform a simulation study to determine which factors can potentially influence classification after imputation. Simulated (N,C) is another synthetic dataset created in a similar manner but now containing categorical, ordinal and uninformative variables in addition to continuous variables. More details are provided in

the Supplementary Notes. MIMIC-III. The Medical Information Mart for Intensive Care (MIMIC) dataset[25] is a large, freely available database comprising de-identified health-related data from patients who were admitted to the critical care units of the Beth Israel Deaconess Medical Center in the years 2001–2012. Following the preprocessing detailed in the Supplementary Notes, we obtain the MIMIC-III dataset used in this paper. This contains data for 7214 unique patients with 14 clinical features and a survival outcome recorded for each patient. Breast Cancer. This dataset is derived from an oncology dataset collected at Memorial Sloan Kettering Cancer Center between April 2014 and March 2017[26]. The dataset contains genomic profiling of 1918 tumour samples from 1756 patients with detailed clinical variables and outcomes for each patient and the therapy administrated over the time of treatment. The Breast Cancer dataset is obtained after the preprocessing described in the Supplementary Notes and has 16 features for 1756 patients with the natural missingness of the data retained. NHSX COVID-19. This dataset is derived from the NHSX National COVID-19 Chest Imaging Database (NCCID)[27] which contains clinical variables and outcomes for COVID-19 patients admitted in many hospitals around the UK. The dataset is continually updated by the NHSX; the download we performed was on August 5, 2020 which contained data for 851 unique patients. The NHSX COVID-19 dataset contains 23 features for 851 patients.

*Data preprocessing*. Among the five datasets we consider, Simulated (N) and MIMIC-III contain only numerical features, whereas Breast Cancer, NHSX COVID-19 and Simulated (N,C) contain numerical, categorical (single and multi-level) and ordinal features. In preprocessing, the categorical features are one-hot encoded, and the ordinal features are coded with integer values before the imputation task.

*Outcome variables*. For the MIMIC-III, Breast Cancer and NHSX COVID-19 datasets we use survival status as the outcome of interest, the Simulated (N) and Simulated (N,C) datasets outcome is a binary variable generated at the data synthesis stage by the `make_classification` function in `scikit-learn`[24].

**Dataset partitioning**. In our experiments, we simultaneously compare many combinations of imputation and classification methods, each with several hyperparameters that can be tuned. Therefore, we must be careful to avoid overfitting. To address this, we partition each dataset at two levels. At the first level, we randomly split the datasets three times to give different development and holdout cohorts and, at the second, we randomly partition each of the development sets into fivefolds (for cross-validation) with four used for training and one for validation, see Supplementary Fig. 2 and the Supplementary Notes for more details.

*Induced missingness*. The Simulated (N), Simulated (N,C) and MIMIC-III datasets are complete and, therefore, we can induce different missingness rates in the development and holdout datasets. We randomly removed 25% and 50% of entries from each of the development and holdout datasets. This allows us to compare four different scenarios for development-holdout missingness rates: 25–25%, 25–50%, 50–25% and 50–50%.

**Imputation methods**. Data imputation is the process of substituting missing values in a dataset with new values that are, ideally, close to the true values which would have been recorded. To formalise the notation, we consider a dataset $\mathcal{D}$, consisting of $N$ samples $\mathbf{x}_i \in \mathbb{R}^d$ drawn from some (unknown) distribution.

Some of the elements in these samples may be missing and we impute them to give the complete samples $\hat{\mathbf{x}}_i$. Whilst it is not possible to be certain about the true value of the missing entries, it is desirable that the uncertainty in the imputed values be considered in any downstream task which relies on the imputed data[28] as the uncertainty of a variable increases after data imputation[29].

In general, imputation methods fall into two categories, namely 'single imputation' and 'multiple imputation'. In the case of single imputation, plausible values are imputed in place of the missing values just once, whereas for multiple imputation methods[30,31], imputation is performed multiple times to generate a series of imputed values for each missing item. For imputation methods that have some stochastic nature, this allows for insight into the uncertainty of the imputed values. For multiple imputation methods, there are two approaches for performing a downstream classification task. The first approach involves pooling the multiple imputation results and creating a single, summary, dataset from which we perform the classification task. The second approach, which we follow in this paper, is to perform the classification task on each of the multiple imputation results separately, before pooling the outputs of the classification model. The final output is determined using e.g., averaging or majority vote. See van Buuren[7], for a comparison of the approaches and justification for the latter being preferable.

In our study, we consider five popular imputation methods from the literature, namely, mean imputation, multivariate imputation by chained equations (MICE)[32,33], MissForest[34], generative adversarial imputation networks (GAIN)[35] and the missing data importance-weighted autoencoder (MIWAE)[36]. The interest of this paper is not to assess all imputation methods, but to draw attention to the importance of assessing imputation quality when fitting classification models to incomplete data. The five methods we consider represent some of the most popular imputation methods, using statistical and machine learning-based methods, but we also highlight there is a vast literature describing additional imputation methods, such as AMELIA[37], K-Nearest Neighbour imputation[38], and fractional hot deck imputation (FHDI)[39] and its parallel variation, P-FHDI[40].

**Classification methods**. Classification is the process of grouping items with similar characteristics into specific classes. In our case, the classification methods take an input sample $\hat{\mathbf{x}}_i \in \mathbb{R}^d$ and output a particular class label $\ell_i \in \{0, 1\}$. The classification methods we consider in this paper are logistic regression, Random Forest, XGBoost, NGBoost and an artificial neural network (for more details, see the Supplementary Notes.)

**Hyperparameter optimisation**. Each of the classifiers that we consider (except for logistic regression) has several tuneable hyperparameters, with classifier performance highly dependent on them. In order to identify the best classifier for each dataset, we perform an exhaustive grid search over a wide range of hyperparameters. For each configuration of a dataset (train/test missingness level, imputation method, holdout set, validation set and each of the multiple imputations), we fit a classifier, exhaustively, over a large range of hyperparameters as detailed in Supplementary Table 1. To identify the optimal hyperparameter choice for each of these configurations, we evaluate the model on each of the five validation folds. The hyperparameter choice, which results in the best average performance over these validation folds, is selected as the optimal model configuration (i.e., for each holdout set H1–H3 (Supplementary Fig. 2) we obtain an optimal model). This exhaustive grid search required millions of experiments to ensure the comparisons are fair.

**ANOVA analysis**. One of the key aims of this paper is to identify and quantify the influence of key factors on the performance of the downstream classification. To quantify the impact, we used a generalised linear model (binomial logistic regression-based) to perform multi-factor ANOVA for the holdout AUC of the optimal classifiers named in the Classification methods section. Firstly, to identify the most influential factors in the models built on the real clinical models, we pool the results for the MIMIC-III, Breast Cancer and NHSX COVID-19 datasets and perform a logistic ANOVA, whilst additionally assessing each of the datasets individually. After constructing the ANOVA model including all factors and their interaction, we excluded factors not significant at the 1% level using backward elimination (see the Supplementary Notes for more details).

**Measuring the quality of imputation**. In addition to determining how the imputation method, missingness rates and datasets affect the downstream classification performance, we are also interested in exploring how the quality of the imputation affects the downstream classification performance. However, there is no widely accepted approach for measuring the quality of imputation. In this paper, to assess imputation quality, we follow[18,41] by taking a complete dataset, introducing missingness artificially, imputing the resulting incomplete dataset and then computing a discrepancy statistic between the original dataset and the imputed dataset.

For this purpose, we induced missingness completely at random (MCAR) with rates 25% and 50% into the MIMIC-III, Simulated (N) and Simulated (N,C) datasets. In order to quantitatively assess how well an imputation method reconstructs the missing values, we must define a discrepancy score which achieves a low value when the distribution of imputed values closely resembles that of the true values. Explicitly, our aim is to compute a distance between the original samples $\mathcal{D} = \{\mathbf{x}_i\}_{i=1}^N$ and the imputed samples $\hat{\mathcal{D}} = \{\hat{\mathbf{x}}_i\}_{i=1}^N$ that measures the quality of the imputation.

In this paper, we consider many of the popular discrepancy statistics used in the literature for measuring imputation quality. These typically fall into two classes: (A) measures of discrepancy between the imputed and true values of individual samples and (B) measures of discrepancy in distributions of individual features for the imputed and true data. However, we strongly believe that the class of measures which would be of most practical value to practitioners is (C) measures of discrepancy for imputed and true data across the whole data distribution. In the literature, we were unable to find any examples of discrepancy measures of this type, and we propose such a class in this paper.

(A) Sample-wise discrepancy: In much of the literature, the quality of imputation is determined by measuring the discrepancy in the real and imputed values sample-by-sample, then summarising over all samples in the dataset. In this paper, we consider the three discrepancy statistics, Root mean square error (RMSE), mean absolute error (MAE) and the coefficient of determination ($R^2$). These statistics compare the imputed values explicitly to the true values.

(B) Feature-wise distribution discrepancy: In the literature, some authors have considered discrepancy measures to quantify how faithfully the distributions of individual features are reconstructed. In particular, Thurow et al.[18] consider several distribution measures on a feature-by-feature basis, including the Kullback–Leibler (KL) divergence, the two-sample Kolmogorov–Smirnov (KS) statistic and 2-Wasserstein (2W) distance. We report results for all of these in this study. As these discrepancies are measured feature-by-feature, we get many scores for each dataset and report the minimum, maximum and median discrepancy for each distance's overall features. More details about these statistics, including definitions and the implementations used, can be found in the Supplementary Notes.

(C) Sliced Wasserstein distance: In Fig. 1, we show that simply considering the feature-by-feature marginal distributions is not sufficient to quantify how well a high-dimensional data structure has been imputed. The marginal distributions of the imputed data (directions 1 and 2) match that of the original data perfectly but do not identify the discrepancy of the distributions shown in Fig. 1a, b. This motivates us to consider a new measure, which harnesses the multi-dimensional nature of the data to better identify distribution differences like this.

Modelling the discrepancy between imputed and true data in high dimensions is challenging for two key reasons. Firstly, the curse of dimensionality results in computations that are infeasible for high-dimensional datasets. Secondly, high-dimensional (complete) datasets are very sparse (with low density) in the space $\mathbb{R}^d$ unless there are an unrealistically large number of samples. We address both of these issues by repeatedly projecting the entire data distribution to random one-dimensional subspaces, thereby increasing the density of the data while considering more axes than those simply defined by the features themselves. We then consider the distance of the imputed data from the original data in the random direction, commonly known as the sliced Wasserstein distance[42,43]; this gives a distribution of distances across all randomly chosen directions. Initially, we perform the following steps:

Step 1: Determine partitions and random directions. Choose $M$ random unit vectors (directions) $\mathbf{n}_r \in \mathbb{R}^d$, $r = 1, \ldots, M$, where $M \geq d$. Choose $P$ random partitions of the index set $\{1, 2, \ldots, N\}$ into two equally-sized subsets $I_p$ and $J_p$ for $p = 1, \ldots, P$ (where $N$ is the number of samples). If $N$ is even, then $I_p$ and $J_p$ are the same sizes. In the case that $N$ is odd, these subsets have sizes $(N + 1)/2$ and $(N − 1)/2$, respectively. In our experiments, we set $P = 10$ and to account for the dimensionality differences between MIMIC-III, Simulated (N) and Simulated (N,C), we used $M = 50$ and $M = 90$, respectively.

Step 2: Calculate projections of data. For each $r$, we project all data onto the one-dimensional subspace of $\mathbb{R}^d$ spanned by $\mathbf{n}_r$; this gives the projected original data $\mathbf{x}_i.\mathbf{n}_r$ and projected imputed data $\hat{\mathbf{x}}_i.\mathbf{n}_r$.

Step 3: Calculate sliced Wasserstein Distances. For each pair $(r, p)$, with $r \in \{1, \ldots, M\}$ and $p \in \{1, \ldots, P\}$, we calculate (2-) Wasserstein distances between the projected original and imputed data. The data are normalised by dividing by the standard deviation, $s$, of the projected data $\mathbf{x}_i.\mathbf{n}_r$ for $i \in I_p$. The datapoints in $I_p$ are taken as the 'true' distribution and we determine the distance $w(r, p)$ of the remaining data in $J_p$ from this. This is our baseline distance inherent to the data. We then calculate $\hat{w}(r, p)$, the 2-Wasserstein distance between $\mathbf{x}_i.\mathbf{n}_r/s$ for $i \in I_p$ and the imputed data $\hat{\mathbf{x}}_j.\mathbf{n}_r/s$ for $j \in J_p$.

See the Supplementary Notes for a step-by-step example of the calculation of the sliced Wasserstein distance.

Over all $(r, p)$ pairs, these steps result in two distributions of distances, for $w(r, p)$ and $\hat{w}(r, p)$, as illustrated in Fig. 2. Firstly, these can be regarded as probability distributions, allowing us to compute the same class B feature-wise discrepancy scores discussed previously. Secondly, we evaluate the relative change in the Wasserstein distance due to imputation for each $r$ and $p$, namely $\hat{w}(r, p)/w(r, p)$, allowing us to quantify how much different imputation methods induce discrepancy in the data distribution. Finally, we assess the stability of the different imputation methods by exploring the variance in the induced sliced Wasserstein distance across repeated imputations.

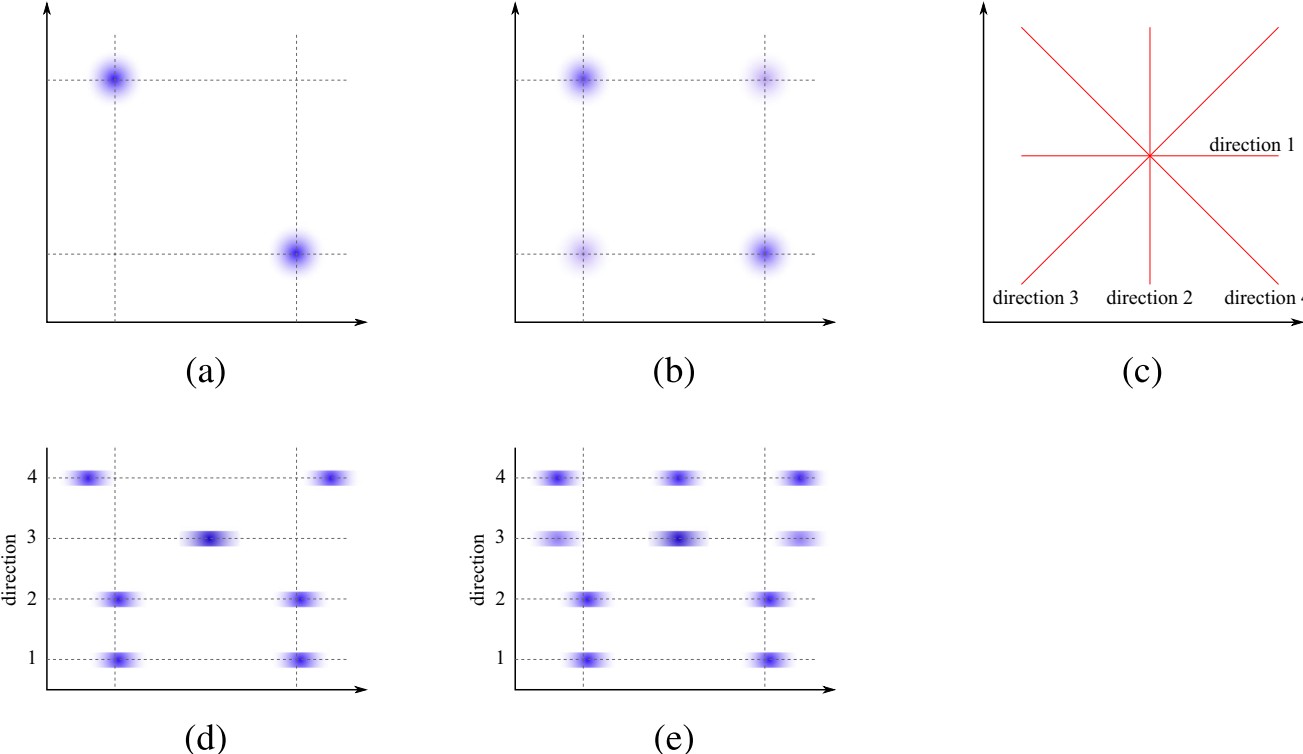

**Fig. 1 The effects of projecting imputed data in multiple directions. a** The underlying 2-dimensional data distribution; **b** the distribution of imputed data; **c** some example directions: 1 and 2 are in the direction of the features, directions 3 and 4 are not; **d, e** show the original and imputed data distributions projected onto the four directions shown in (**c**): their marginals (directions 1 and 2) are indistinguishable but the distributions are clearly different when projected in directions 3 and 4.

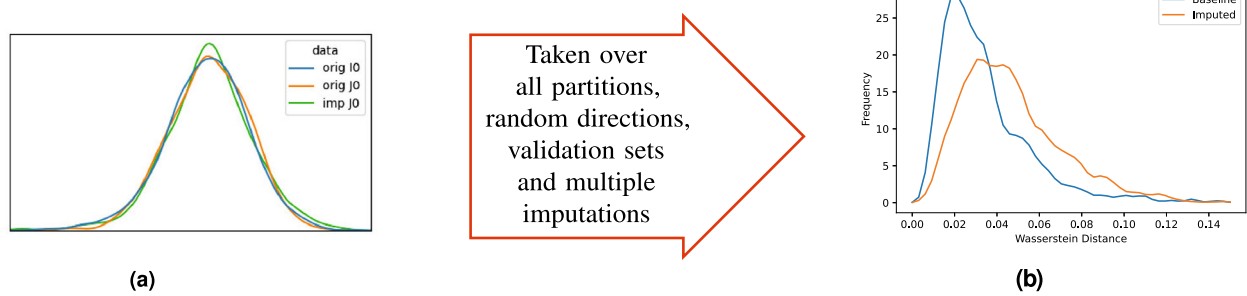

**Fig. 2 Procedure for deriving the sliced Wasserstein discrepancy statistics. a** Example density plot of the projection onto $\mathbf{n}_r$ of the original data in $I_p$ (blue) and $J_p$ (orange) and the imputed data in $J_p$ (green). **b** Density plot of the sliced Wasserstein distances for the original and imputed data.

**Downstream effects of imputation quality**. A critical question that this paper aimed to answer is: in what way does the quality of the data imputation affect the downstream classification performance? Firstly, we determined whether there is a correlation between the discrepancy scores in classes A–C and the performance of the classification model. Secondly, we investigated whether using poorly imputed data in the training of a classification model would affect the importance of the feature to each classifier. To answer this, we analyse the models fit to the Simulated (N) and Simulated (N,C) datasets. As the features are of equal importance by design, the clusters of values within each feature are normally distributed, centred at vertices of a hypercube and separated[24]. For each classifier, we identify two models which perform well at the classification task where one is trained using poorly imputed data and the other is fit to data that is imputed well. See the Supplementary Notes for details on the model selection. To assess the feature importance for each of the trained models, we used the popular Shapley value approach of Lundberg et al.[44]. For each feature, this assigns an importance to every value, identifying how the model output is affected on the basis of the value of this feature. A positive importance value indicates the value influences the model towards a positive class. Due to the design of the Simulated (N) dataset, we expect the distribution of the Shapley values to be symmetric, as the clusters are separated by a fixed distance and are normally distributed within those clusters. Using the skewness, we measure the symmetry of the feature importance values for each feature for all models.

**Reporting summary**. Further information on research design is available in the Nature Portfolio Reporting Summary linked to this article.

## Results

**Classifier influence on downstream performance.** In Fig. 3a, we show how the classifier affects the downstream performance in terms of the Area Under the Receiver Operating Characteristic (AUC-ROC) curve for each dataset across the different train and test missingness rates. For the MIMIC-III, Simulated (N) and Simulated (N,C) datasets, the performance of a classifier trained using the complete original dataset always exceeds that of those built on imputed data (with the sole exception of the Neural Network for MIMIC-III). The performances for the Simulated (N) and Simulated (N,C) datasets are much higher than for the real datasets as, by design, there is a direct link between the outcome and the feature values. For the Simulated (N), Simulated (N,C) and MIMIC-III datasets, we see that increasing the train and test missingness rates leads to a decline in performance. For all classifiers with a fixed train missingness rate, a change in the

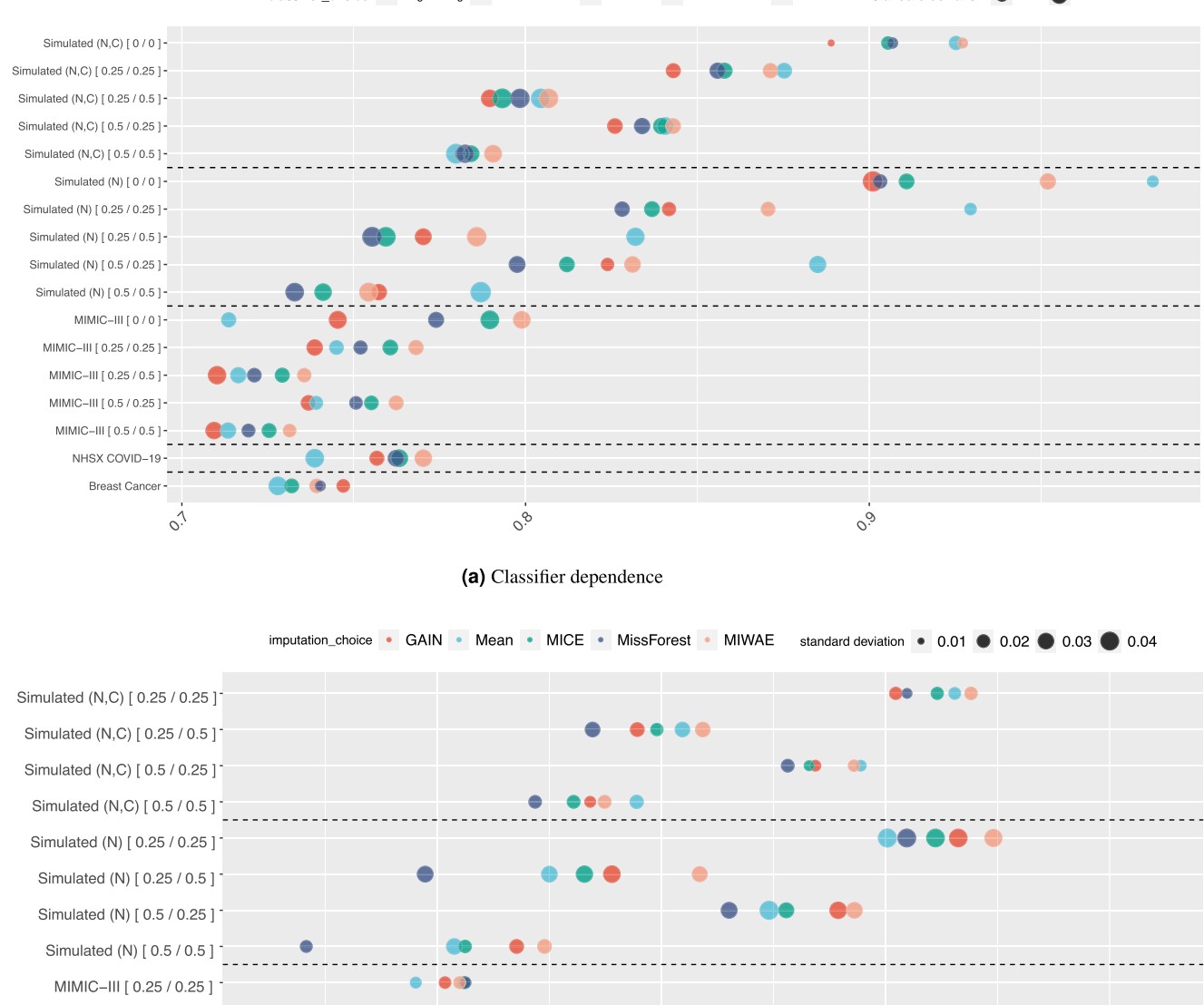

**(a)** Classifier dependence

**(b)** Imputation dependence

**Fig. 3 Dependence of downstream classification AUC performance on classification and imputation methods.** These plots show the dependence of the downstream performance on the **a** classification and **b** imputation methods. For the Simulated (N), Simulated (N,C) and MIMIC-III datasets, we show performance for 25% and 50% levels of missingness in the development and test data. Datasets are separated by dashed lines from each other. The size of each marker corresponds to the standard deviation in the results, and the values after each dataset name indicate the [train/test] missingness.

test missingness rate affects performance drastically compared to changing the train missingness rate for a fixed test missingness rate. For each imputed dataset, we see that when varying the missingness rates of the development and test data, the ranking of each classifier's performance is almost consistent, e.g., for the Simulated (N) dataset, the Neural Network always performs the best while logistic regression and the Random Forest perform worst (logistic regression has higher variance). The XGBoost classifier performs best for the MIMIC-III and NHSX COVID-19 datasets, with consistency in performance ranking at all missingness rates. The Breast Cancer dataset follows the trend of the other real-world datasets, with the important exception that the worst-performing classifier for the other datasets, namely logistic regression, performs the best here. For most datasets, the variance of the classifier performance is similar across the classifiers, the exception being the Breast Cancer dataset for which the Random Forest has a small variance and Neural Network a relatively large variance. In the Supplementary Figs. 5–9 we also present performance in terms of accuracy, Brier score, precision, sensitivity/recall and specificity.

**Imputation influence on downstream performance**. In Fig. 3b, we show the dependence of the downstream classification performance on the imputation methods used to generate the complete datasets. For the Simulated (N) and NHSX COVID-19 datasets, at all levels of development and test missingness rates, MIWAE imputation gives the best downstream performance. For the Simulated (N), Simulated (N,C) and MIMIC-III datasets, as the missingness rate increases in the test dataset, the performance of models trained using imputed data declines drastically. In the real-world datasets, there is no 'best' imputation method that leads to a model which outperforms the others although MIWAE gives a consistently high-performing model, being either first or second best. For the MIMIC-III dataset, we see that the simple mean imputation method gives the worst downstream performance for all missingness rates but for NHSX COVID-19 is competitive with the best-performing MIWAE method, potentially due to mode collapse of the deep learning methods. We observe a large performance difference between the synthetic and real-world datasets when missingness rates are low, which decreases considerably as the train and test missingness rates increase. Moreover, in the real-world datasets, we observe that the difference in performance for different imputation methods is marginal, and the choice of the imputation method does not play an important role in the classification performance. The variance in the performances is consistent across the real-world datasets, in the range [0.01, 0.02], but is markedly higher for all levels of missingness in the Simulated (N) dataset, in the range of [0.02, 0.04]. In Supplementary Figs. 5–9, we illustrate the effect of the different imputation methods on the downstream accuracy, Brier score, precision, sensitivity/recall and specificity.

**ANOVA for downstream classification performance**. In Fig. 4, we show the significant factors identified for the pooled data and individual datasets. For the pooled data, it is clear that the missingness rate of the test set explains most of the deviance in the results of our classification models, confirming the observations previously derived from Fig. 3. The dataset under consideration and the classification method used are the next most important factors. The ANOVA is also performed for the individual datasets, shown in Fig. 4b–d. For the Simulated (N) and Simulated (N,C) datasets, we see that many factors affect the classification performance, but surprisingly the imputation method itself has a relatively small impact on the deviance of downstream classification. Only the Simulated (N), Simulated

(N,C) and MIMIC-III datasets have induced missingness, and we see that the test set missingness rate is the most significant source of deviance in the downstream classification performance, followed by the choice of classification method, imputation choice and train missingness rate. For both the NHSX COVID-19 and Breast Cancer datasets, the classification method is the primary source of deviance. Indeed for the Breast Cancer dataset, it is the only significant factor at the 1% level. The results of these ANOVA analyses are provided in the Supplementary Tables 2–6.

**Comparing imputation quality**

*Sample-wise statistics*. In Fig. 5 and Supplementary Figs. 10–12, the sample-wise discrepancy scores are shown for the MIMIC-III, Simulated (N) and Simulated (N,C) datasets for different train and test missingness rates. RMSE and MAE are generally consistent across the different imputation methods for fixed train and test missingness rates. There is a minimal performance difference between holdout sets. Using these sample-wise measures for the MIMIC-III dataset, the MissForest imputation method performs the best overall, followed by GAIN. The gap between the performance of MissForest and GAIN narrows as the train and test missingness rates increase. MICE imputation performs worst at all train and test missingness rates. For the Simulated (N) dataset, mean imputation tends to perform the best, whilst MICE is consistently the worst. For the Simulated (N,C) dataset, the best-performing methods are MissForest and mean imputation, whilst MICE is the worst. MICE generally attains the lowest $R^2$ score at all missingness rates for all datasets. The poor performance of MICE by the RMSE metric can also be seen in Supplementary Fig. 1, although, qualitatively, the distribution is better recreated using this imputation method.

*Feature distribution metrics*. In Fig. 6 and Supplementary Figs. 13–23, we show the minimum, median and maximum feature-wise discrepancies statistics for the MIMIC-III, Simulated (N) and Simulated (N,C) datasets. In general, the mean imputation method is the worst in all metrics for minimum, median and maximum at all rates of train and test missingness in all datasets. MissForest performs best by sample-wise RMSE and MAE, and very competitively for the feature-wise discrepancy scores. MICE imputation, which performed worst by the sample-wise discrepancy scores, is the best-performing method by the Kolmogorov–Smirnoff statistic and Wasserstein distance for all missingness rates across all the minimum, median and maximum discrepancies. It is generally the best method by Kullback–Leibler divergence, with MissForest and MIWAE also competitive. For GAIN, the minimum discrepancies are competitive with the other imputation methods. However, when considering the maximum discrepancy, we note that the difference in the performance of the mean and GAIN methods narrows considerably. Increasing the test missingness rate leads to a drastic increase in the feature-wise distances, whereas there is a more subtle increase in the distances with an increase in the train missingness rate from 25 to 50%.

*Sliced Wasserstein distribution metrics*. In Fig. 7 and Supplementary Figs. 24–29, we show the discrepancies between the distributions of the sliced Wasserstein distances, measured using the Kullback–Leibler divergence, the Kolmogorov–Smirnov statistic and the Wasserstein distance for the imputation methods at different train and test missingness rates for the MIMIC-III, Simulated (N) and Simulated (N,C) datasets. For the MIMIC-III dataset, over all discrepancy scores and missingness rates, the MICE imputation method shows a clear dominance, with MissForest and MIWAE are competitive and poor performance is observed with mean and GAIN imputation. For the Simulated

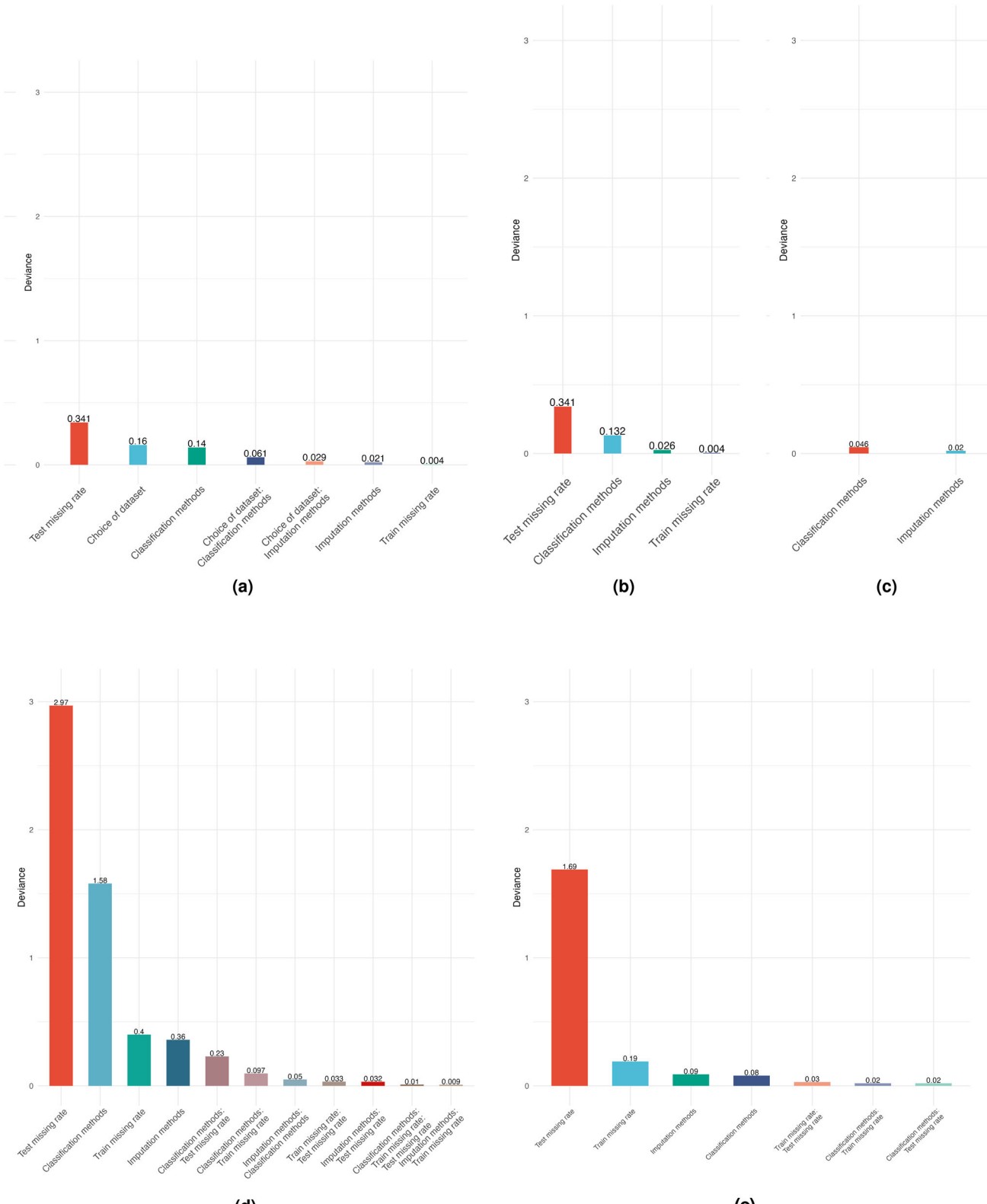

**Fig. 4 Pooled and dataset-segregated ANOVA analysis.** In these plots, we show the significant factors in the ANOVA analysis for the **a** pooled dataset ($n = 1050$), **b** MIMIC-III dataset ($n = 300$), **c** NHSX COVID-19 dataset ($n = 75$), **d** Simulated (N) dataset ($n = 300$) and **e** Simulated (N,C) dataset ($n = 300$). Note that we do not display for Breast Cancer, as the choice of the classifier is the only significant factor).

(N) dataset, the MICE method again performs the best overall by all measures at 25% train missingness but its relative performance is unclear for 50% train missingness. For the Simulated (N,C) dataset, MICE and MIWAE are generally the best-performing methods. At the 25% test missingness rate, the MIWAE imputation method is competitive with MICE, sometimes outperforming it. For MIMIC-III and Simulated (N), mean imputation performs the worst, with GAIN found to generally

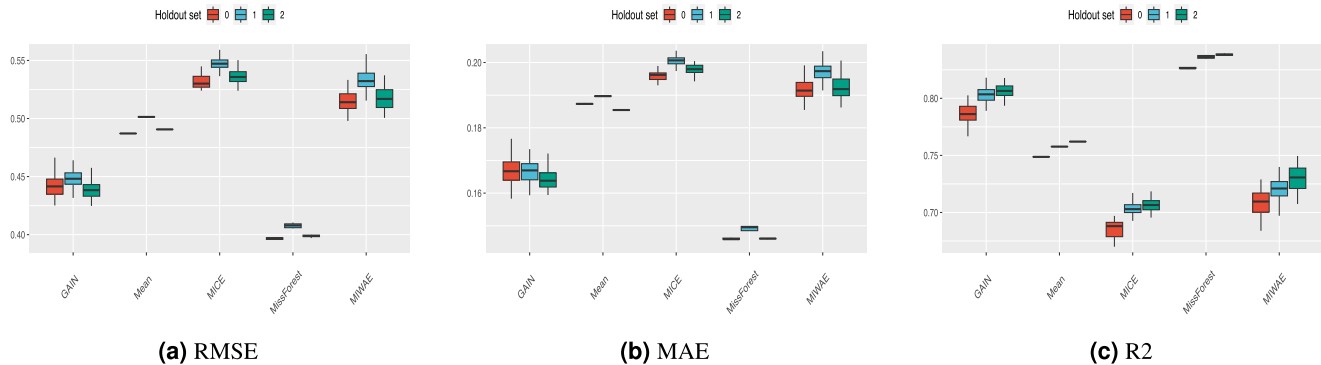

**(a)** RMSE **(b)** MAE **(c)** R2

**Fig. 5 Sample-wise statistics for the MIMIC-III dataset with 25% train and test missingness rates.** Results for **a** RMSE, **b** MAE and **c** $R^2$ with $n = 750$ for all boxplots. Note that for presentation purposes, the scales between plots are different. Whiskers extend to the extreme values, with outliers omitted that are above 1.5 times the interquartile range from the median (horizontal line). Note that for presentation purposes, the scales between plots are different.

**Fig. 6 The feature-wise statistics for the MIMIC-III dataset with 25% train and test missingness rates.** Discrepancy scores for the **a–c** Kullback–Leibler, **d–f** Kolmogorov–Smirnov and **g**, **h** Wasserstein methods. **a**, **d**, **g** are the minimum scores over all features, **b**, **e**, **h** are the median scores and **c**, **f**, **i** are the maximum scores. Each boxplot consists of $n = 210$ datapoints. Whiskers extend to the extreme values, with outliers omitted that are above 1.5 times the interquartile range from the median (horizontal line).

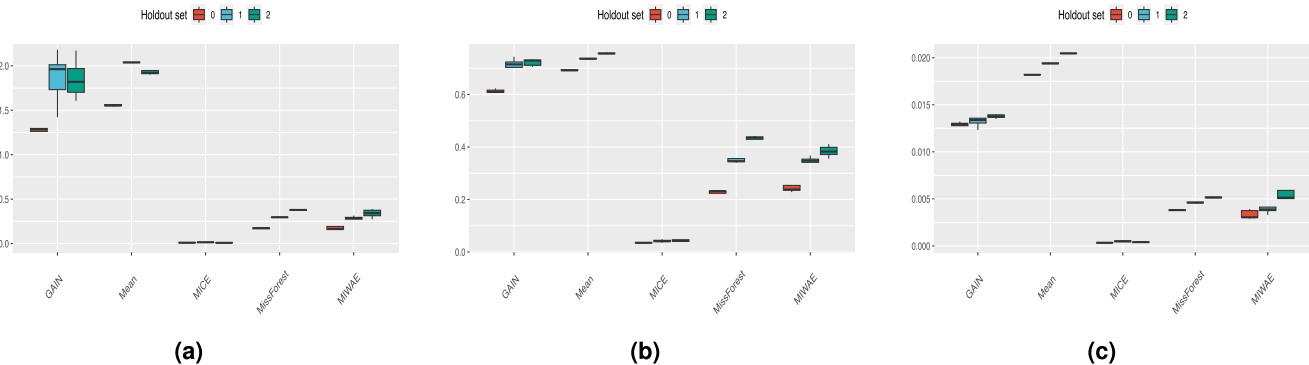

**Fig. 7 Distances derived from the Sliced Wasserstein distance distributions for the MIMIC-III dataset with 25% train and test missingness rates.**
Results for **a** Kullback–Leibler, **b** Kolmogorov–Smirnoff and **c** Wasserstein distance. Each boxplot consists of $n = 75$ datapoints, and for presentation purposes the scales between plots are different. Whiskers extend to the extreme values, with outliers omitted that are above 1.5 times the interquartile range from the median (horizontal line).

perform similarly poorly whilst for Simulated (N,C) there is no clear method which performs best.

**Sliced Wasserstein distance ratio analysis**. In Supplementary Figs. 32–34 are the boxplots of the ratios of the distances from $J_p$ to $I_p$ for the imputed data compared to the original data for the MIMIC-III, Simulated (N) and Simulated (N,C) datasets, respectively. Firstly, for MIMIC-III and Simulated (N) we see that the MICE imputation method induces a much smaller distance ratio than any other method whereas for Simulated (N,C) this is true for both MICE and MIWAE. Secondly, we note that with an increase in the train missingness rate, the ratio of the distances remains largely consistent, however, with an increase in the test missingness rate, we see a very considerable increase in the ratio.

**Outlier analysis**. It is important to understand how stable the imputation methods are when imputation is repeated and also whether the stochastic nature of the imputation methods can lead to outlier imputed values for particular features. In Fig. 7 and the Supplementary Figs. 24–29, we see that some of the imputation methods can lead to distances with large variances, especially as the missingness rates in the train and test sets increase. This variance can result from either the random projections (in some cases, the distributions match well and in others quite poorly) or stochasticity in the imputation algorithm. We are keen to understand the influence of each. In Fig. 8 and Supplementary Fig. 3, we see that the MICE method performs consistently well across all holdout and validation sets with no imputations above the distance of $10^{-7}$ and at the threshold of $1.5 \times 10^{-8}$, 90% of the MICE imputations are above this distance. This demonstrates a consistency of the MICE imputations, with most imputations at a distance between $1.5 \times 10^{-8}$ and $10^{-7}$ from the true values.

**Link between imputation quality and downstream classification performance**. In Fig. 9 and Supplementary Figs. 30 and 31, we plot the results for all class A, B and C imputation discrepancy statistics against the AUC of the downstream classification task. Our previous analysis has shown that the test missingness rate has a large influence on both imputation quality and downstream performance, so we display the correlations separately for 25% and 50% missingness.

For the sample-wise metrics, we see a clear negative correlation between discrepancy and classification AUC for the Simulated (N) and Simulated (N,C) datasets but a near-zero, if slight negative correlation, for the MIMIC-III dataset. For the feature-

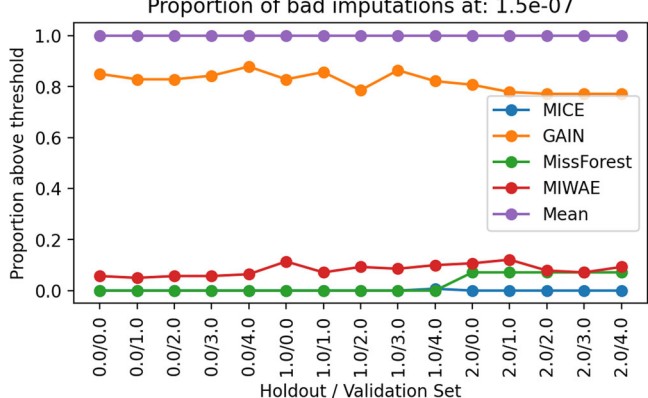

**Fig. 8 Identifying outliers in the imputations.** The proportion of repeated imputations that give outlier Wasserstein distances, at threshold $10^{-7}$, for different imputation methods.

wise and the proposed sliced Wasserstein discrepancy statistics, we see a negative correlation for all measures in both the MIMIC-III and Simulated (N) datasets but interestingly, it is not observed for the Simulated (N,C) dataset.

In Fig. 10 and Supplementary Fig. 35, we give heatmaps showing the correlations between the nine different discrepancy statistics used for the MIMIC-III, Simulated (N) and Simulated (N,C) datasets, respectively. Within each class of discrepancy metric (for A, B and C), the measures are all correlated with one another, however, the sample-wise metrics (class A) do not highly correlate with any of the feature-wise (class B) or sliced Wasserstein distances (class C). There is also a strong correlation between most of the class B and C metrics.

**Impact of imputation quality on interpretability**. For each classifier, we find that the best sliced Wasserstein distance ratio is always obtained for data imputed with MICE. For NGBoost and XGBoost, the worst distance ratio of the high-performing models is for GAIN imputed data whereas, for Random Forest, this is for mean imputed data. For each of the 25 features in the Simulated (N) dataset, we calculate the absolute value of the skew for the Shapley values and display these in Supplementary Fig. 4. For the Random Forest classifier, we see that for 21/25 features, the absolute skew of the Shapley values for the MICE imputation is smaller than that of mean imputation. For NGBoost, we see a

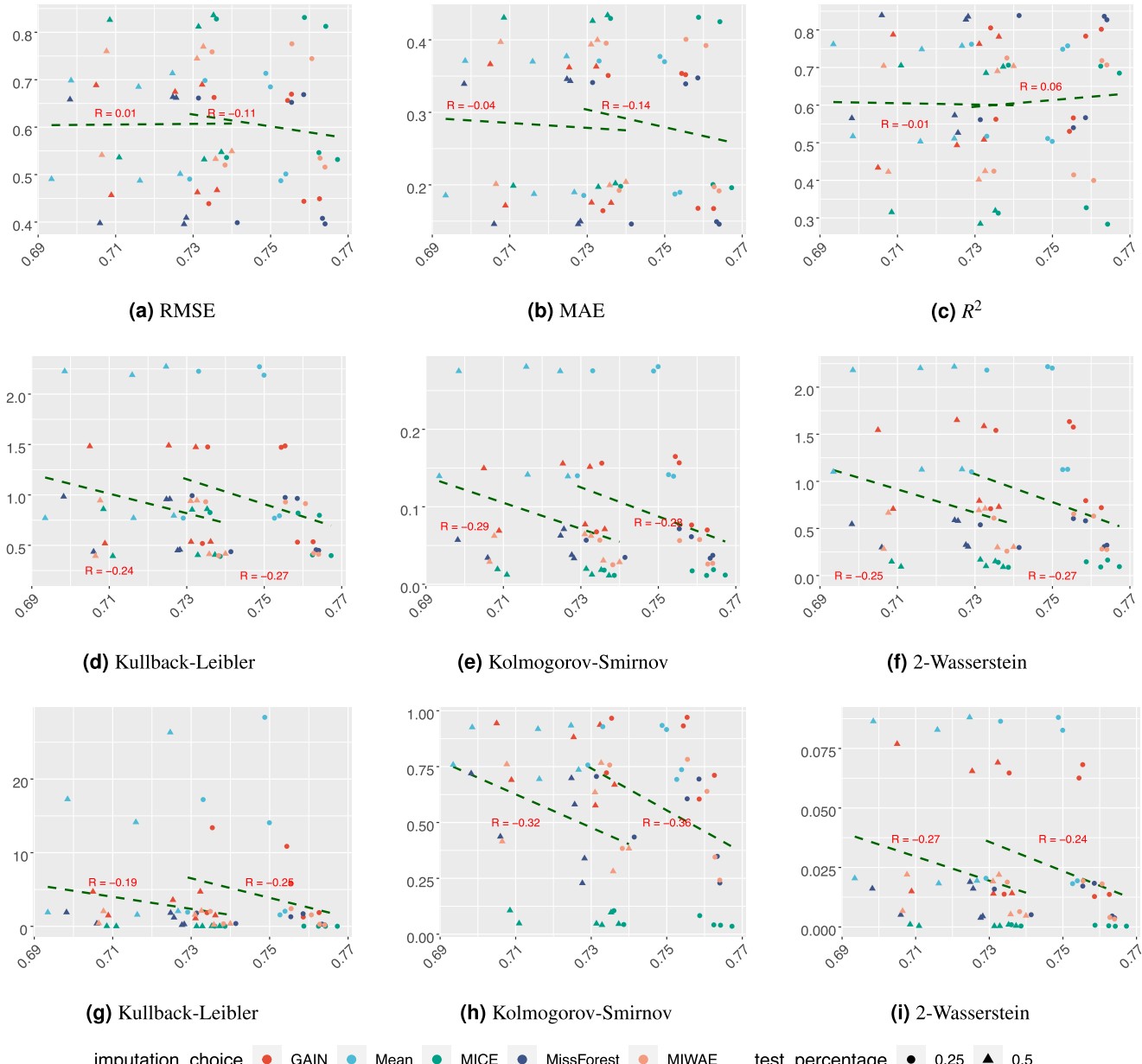

**Fig. 9 Imputation discrepancy metrics for classes A, B and C against the downstream classification AUC value for MIMIC-III. a–c** are the sample-wise measures, **d–f** are the feature-wise distances and **g–i** are the sliced Wasserstein-derived distances. The $y$ axes show the AUC values and the x-axes show the **a** RMSE, **b** MAE, **c** $R^2$, **d**, **g** Kullback–Leibler, **e**, **h** Kolmogorov–Smirnov, **f–i** 2-Wasserstein. Trend lines are shown for 25% and 50% test missingness separately.

smaller absolute skew in 19/25 features for MICE against GAIN. Finally, for XGBoost, we see a smaller absolute skew in 15/25 features for MICE against GAIN. Supplementary Tables 7–9 contain the details of the model configurations used for this analysis.

## Discussion

In this paper, we have highlighted the importance for machine learning and data science practitioners to reconsider the quality of the imputed data which is used to train a classifier. Each of the datasets considered in this study contains different missingness rates and missingness types. In particular, the NHSX COVID-19 and Breast Cancer datasets have data that is missing not at random (MNAR), while the induced missingness to the MIMIC-III, Simulated (N) and Simulated (N,C) datasets is missing completely at

random (MCAR). It was found that there is no particular classifier that outperforms all others across all of the datasets and similarly, no particular imputation method leads to the best downstream classification performance. Even for the datasets with the same types of missingness, no optimal imputation or classification method emerges. Importantly, however, models fit to poorly imputed data give rise to misleading feature importances, demonstrating the adverse impact of poor imputation quality on model interpretability. Any features judged as important from those models are therefore compromised. This is crucial to appreciate, especially where models are fit to clinical data, as incorrect conclusions may be drawn about the influence of particular features on patient outcomes. Overall, this suggests that the quality of imputation does feed through to the downstream model interpretability, i.e., whilst classifiers may be able to

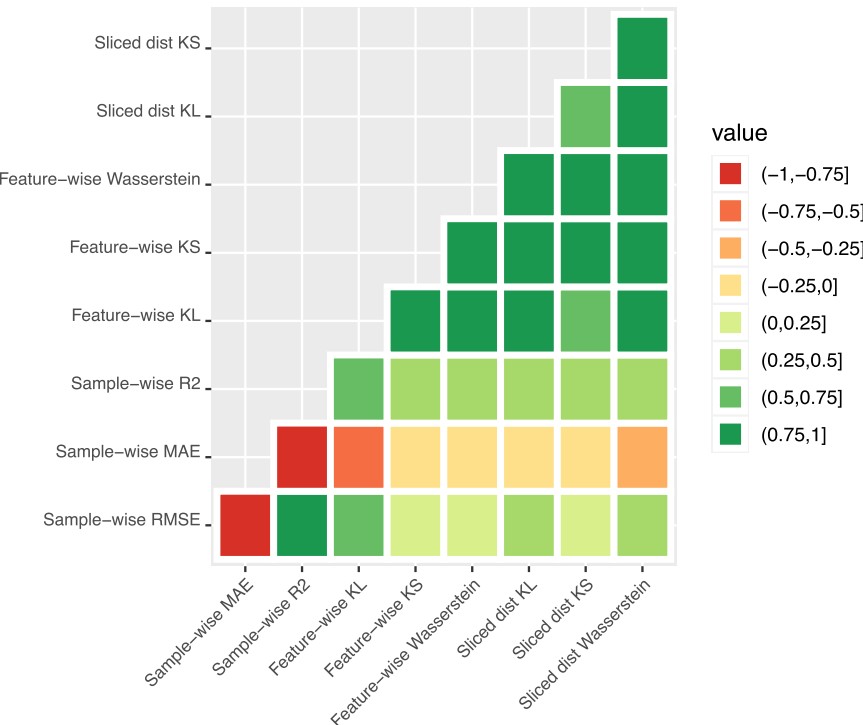

**Fig. 10 Correlations between discrepancy scores.** Heatmap showing the correlation between all discrepancy metrics considered in this paper for the MIMIC-III dataset. Correlations are computed using $n = 30$ datapoints.

achieve high performance in spite of poorly imputed data, they are compromised downstream.

Interestingly, classification models built on data imputed using mean imputation demonstrate the largest variance in the downstream performance, even though the imputed data is exactly the same in each of the multiple imputations. This is likely due to the large uncertainty introduced by mean imputation to each variable compounding with stochastic algorithms.

Using ANOVA, we found that performance is most influenced by the test missingness rate. This would suggest that when deploying a trained model to a new dataset, it is of primary importance to be conscious of the missingness rate of the new data, as classifier performance is very sensitive to it. ANOVA also highlights a minimal dependence of the downstream classification performance on the imputation method chosen.

There is no accepted approach for evaluating the quality of an imputation method. Typically, imputation methods are evaluated using sample-wise discrepancy statistics such as RMSE, MAE and $R^2$, but this approach implicitly assumes that the aim of imputation is to recover the true value of the missing data point, rather than recreating the correct distribution. In our experiments, we find that the sample-wise discrepancy scores are not sufficient to assess the quality of the reconstruction of the distribution of imputed values. In fact, for MSE, we have seen that it takes an optimal value for imputations that give a very poor distribution match. Using MSE to evaluate the imputation quality is only statistically justified in the case of imputing data from a Gaussian distribution (minimising MSE corresponds to maximising the log-likelihood). Since the Gaussian assumption often doesn't hold in practice, MSE can be a very inaccurate measure of imputation quality. Beyond sample-wise discrepancy scores, some studies have considered the reconstruction quality of distributions for datasets on a feature-by-feature basis. However, in this study, we have shown how considering these marginal distributions alone is not sufficient to understand how well the overall data distribution has been reconstructed. As stated by van Buuren[7], imputation is

not prediction. The goal of an imputation method is to recover the correct distribution, rather than the exact value of each missing feature. However, the metrics used in the literature simply do not currently enforce this. In this paper, we introduced a class of discrepancy statistics, based on the sliced Wasserstein distance, which allows us to evaluate how well all features have been reconstructed in a dataset.

Interestingly, although GAIN performs well in the sample-wise discrepancy scores, we see poor performance in the discrepancy when measured feature-wise and with our proposed class of sliced Wasserstein distance-based scores. In fact, we find that the GAIN imputation method tends to perform in line with mean imputation and gives a poor reconstruction of the underlying data distribution. It has been identified in ref. [45] that GAIN, which uses a generative adversarial approach for training, tends to rely mostly on the reconstruction error in the loss term (the mean square error) and the adversarial term contributes in a minimal way. The MICE imputation method has shown a clear dominance in recreating the distribution of the datasets as a whole, echoing[46] who find MICE outperforms machine learning-based imputation methods. In both datasets for which it could be evaluated and across all train and test missingness rates, it performs better than the competitor imputation methods at replicating the data distribution. We note that MICE and MissForest are extremely computationally expensive for high-dimensional data[45] so may be infeasible in some circumstances. In particular, MICE also suffers from a key theoretical shortcoming, that it is ignorant of whether joint distributions actually exist, but will produce imputations regardless[7] and has a tendency to crash for high-dimensional non-continuous variables[47].

When evaluating the ratio between the sliced Wasserstein distance in the imputed data versus the original data, we found that consistent with the earlier observations, an increased test missingness rate leads to a large increase in the distance induced by imputation. However, an increase in the development set missingness rate leads to a minimal increase in the induced

distance ratio. We found that MICE outperforms the other imputation methods, inducing the smallest relative increase in distance with minimal variance. It also gave highly consistent imputation results, with minimal variance in the distances between the imputed and original values. Performing multiple imputations highlights that GAIN and MIWAE suffer from mode collapse in some of the rounds of imputation and can fall into outlying local minima, likely due to the highly non-convex problem they are solving. This is a problem common to all deep neural network architectures and serves as a reminder, for imputation methods, of the importance of performing multiple imputations to overcome the risk of some poor-quality imputations.

We find that it is not necessarily the best quality imputation that leads to the best-performing classification method. This observation could be explained by the fact that imputation does not add any information that was not present in the dataset to begin with. Therefore, a powerful classifier may be able to extract all information relevant to the classification task regardless of the imputation quality. Secondly, we conjecture that an inaccurate imputation could in fact provide a form of regularisation for the classifier. It has been shown that perturbing the development data with a small amount of Gaussian noise is equivalent to $L_2$ regularisation[48] and can be beneficial to the performance in supervised learning tasks. In the same spirit, an inaccurate imputation method resulting in noisy imputed values could provide a regularising effect beneficial to performance in the downstream classification task. In addition, we observed that, with fixed train missingness, an increase in the test missingness leads to a large decrease in the performance of the classification methods, likely due to the imputed values being more variable for larger test missingness.

We also find that not only are the common discrepancy scores used by the community, i.e., the sample-wise statistics (RMSE, MAE, $R^2$), uncorrelated from the distributional discrepancy metrics (of classes B and C) but that they are also disconnected from the downstream classification performance of the model. This highlights how important it is to consider additional statistics when measuring imputation quality, not simply the sample-wise statistics, as the distribution discrepancy scores occupy a completely orthogonal space to the sample-wise ones. Importantly, we find a correlation between the proposed class of sliced Wasserstein discrepancy scores and the downstream model performance suggesting that a link has been forged from the imputed data to the downstream model performance. Given this, we would suggest that instead of focusing on optimising performance by considering the best combination of the imputation method and classifier, attention should shift towards optimising the imputation quality in terms of how well the distribution is reconstructed.

In addition to the main aims of this paper, through this work, we have also identified some issues that need the attention of the imputation community, such as how categorical and ordinal variables should be correctly imputed. For example, if a categorical variable is one-hot-encoded then a valid imputation must be in {0, 1}. However, most imputation methods will give a value in the range [0, 1] and these must be post-processed. It is not clear how this should be performed. Similarly, if a category with multiple values is one-hot encoded, e.g., nationality, then only one of these variables should equal one but no imputation method enforces this. Issues were also identified and fixed, in the public code releases of the GAIN and MIWAE imputation methods and so, to improve the quality of future benchmarking of imputation methods, we release our codebase in a GitLab repository[23]. This allows for rapid computation of the results for

several imputation methods across incomplete datasets with data preprocessing, partitioning and analysis, all built in. This should serve as a sandbox for the development and fair evaluation, of new imputation methods.

This study has been designed to highlight the importance of considering the quality of the imputation for datasets which are then used in downstream tasks, therefore there are several limitations to this study. Firstly, we only focus on classification tasks as these represent the majority of the problems encountered in machine learning research applied to clinical data, i.e., predicting death vs. survival or malignant vs. benign disease. It is our hope that by highlighting the consequences for classification models of using poorly imputed data, it will motivate the community to also focus on this for other predictive models with single or multiple outputs. Secondly, we do not aim to provide a fully exhaustive empirical analysis of all imputation methods and classifiers as the aim of this manuscript is to draw the reader's attention to the importance of measuring imputation quality before fitting models.

In conclusion, this study highlights how machine learning classification models are compromised in numerous different ways if fit to poorly imputed data. We also identify that existing common approaches for measuring imputation quality are flawed and propose to use a discrepancy measure derived from the sliced Wasserstein distance. It is also our hope that by providing an open-source codebase that standardises the imputation methods, classification methods and analysis pipelines for determining the imputation quality, future studies can build on this work to consider different data types, missingness types, different outcomes and other imputation/classification methods.

## Data availability

The data that support the findings of this study are openly available for the simulated, MIMIC-III and Breast Cancer datasets in our GitLab repository[23]. The NHSX COVID-19 dataset is available from the originators upon completion of an approved request (see https://nhsx.github.io/covid-chest-imaging-database/data-access). The MIMIC-III dataset is available after completion of a training course at https://physionet.org/content/mimic3-carevue/1.4/. The Breast Cancer dataset is freely available at https://www.cbioportal.org/study/summary?id=breast_msk_2018. All code required to pre-process the data and generate all Figures and Tables are also publicly shared[23].

## Code availability

Our GitLab repository[23] contains all data and code used to generate the results in this paper. Moreover, we provided a scheme of the whole code architecture in the GitLab repository in Supplementary Fig. 36.

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

## Acknowledgements
There is no direct funding for this study, but the authors are grateful for the following indirect funding: European Research Council (No. 716063) (T.S. and J.T.), Academy of Finland (No. 323098) (T.M.) (No. 317680) (T.S. and J.T.), BusinessFinland (T.S. and J.T.), The Sigrid Juselius Foundation (T.S. and J.T.), the EU/EFPIA Innovative Medicines Initiative project DRAGON (101005122) (T.S., M.R., J.S. J.G., S.D., E.S., AIX-COVNET, and C.-B.S.), Astra-Zeneca (M.R., P.T. and M.P.), the Trinity Challenge (M.R. and C.-B.S.), the EPSRC Cambridge Mathematics of Information in Healthcare Hub EP/T017961/1 (M.R., J.H.F.R., J.A.D.A. and C.-B.S.), the Cantab Capital Institute for the Mathematics of Information (J.S. and C.-B.S.). Aviva (J.S.), the European Research Council under the European Union's Horizon 2020 research and innovation programme grant agreement no. 777826 (M.T., C.-B.S.), the Alan Turing Institute (M.T. and C.-B.S.), Fundación Rafael del Pino (R.V.), Wellcome Trust (J.H.F.R.), The Mark Foundation for Cancer Research (E.S.), Cancer Research UK Cambridge Centre (C9685/A25177) (E.S. and C.-B.S.), British Heart Foundation (J.H.F.R.), the NIHR Cambridge Biomedical Research Centre (J.H.F.R.), HEFCE (J.H.F.R.), the Finnish Cancer Organizations (A.S.R.) and the Jane and Aatos Erkko Foundation (A.S.R.). In addition, C.-B.S. acknowledges support from the Leverhulme Trust project on 'Breaking the non-convexity barrier', the Philip Leverhulme Prize, the EPSRC grants EP/S026045/1 and EP/T003553/1 and the Wellcome Innovator Award 215733/Z/19/Z and 221633/Z/20/Z. Finally, the AIX-COVNET collaboration is also grateful to Intel for financial support.

## Author contributions
T.S., M.R., J.S., J.G., P.T., M.P., J.T. and C.B.S. designed the project. T.S., J.G. and M.R. collected and pre-processed the data. T.S., M.R., J.S., J.G. and P.T. performed the experimentation, data analysis and wrote the codebase. T.S., M.R., J.A.D.A., S.D., J.T. and J.G. interpreted the results. T.S. and M.R. wrote the initial manuscript draft. T.S., M.R., J.S., J.G., P.T., S.D., M.T., R.V.T., E.S., P.L., M.P., J.P., J.H.F.R., J.A.D.A., T.M., A.S.R. and C.B.S. provided critical revisions to the draft manuscript and suggested additional experiments. All authors contributed to the revising of the manuscript after review and approved the final version.

## Competing interests
The authors declare no competing interests.

## Additional information

## AIX-COVNET Collaboration

Michael Roberts [2,3,23], Sören Dittmer[2,4], Ian Selby[7], Anna Breger[2,15], Matthew Thorpe[5], Julian Gilbey [2,23], Jonathan R. Weir-McCall[7,16], Effrossyni Gkrania-Klotsas[9], Anna Korhonen[17], Emily Jefferson[18], Georg Langs[19], Guang Yang[20], Helmut Prosch[19], Jacobus Preller [9], Jan Stanczuk[2,23], Jing Tang [1], Judith Babar[9], Lorena Escudero Sánchez[7], Philip Teare[3,23], Mishal Patel[3,8], Marcel Wassin[21], Markus Holzer[21], Nicholas Walton[22], Pietro Lió[6], Tolou Shadbahr [1,23], James H. F. Rudd [10], Evis Sala [7] & Carola-Bibiane Schönlieb[2]

[15]Faculty of Mathematics, University of Vienna, Vienna, Austria. [16]Royal Papworth Hospital, Cambridge, Royal Papworth Hospital NHS Foundation Trust, Cambridge, UK. [17]Language Technology Laboratory, University of Cambridge, Cambridge, UK. [18]Population Health and Genomics, School of Medicine, University of Dundee, Dundee, UK. [19]Department of Biomedical Imaging and Image-guided Therapy, Computational Imaging Research Lab Medical University of Vienna, Vienna, Austria. [20]National Heart and Lung Institute, Imperial College London, London, UK. [21]contextflow GmbH, Vienna, Austria. [22]Institute of Astronomy, University of Cambridge, Cambridge, UK.

