## [Peer Review File · Communications Medicine]

Reviewers' comments:

Reviewer #1 (Remarks to the Author):

Please check the attached report.

Reviewer #2 (Remarks to the Author):

This work introduces a class of discrepancy statistics, based on the sliced Wasserstein distance, which allows assessing how well features with missing values have been reconstructed in a dataset after performing missing data imputation. Even though there are a few typos and (parts of) sentences that could be improved, the paper is well-written and follows IMRaD format, which makes sense since it was submitted to Communications Medicine. It is however very inconvenient for readers to have to jump to the Supplementary Materials repeatedly and then go back to understand the paper – in some parts, the Supplementary Materials are literally Fundamental Materials, without which the big picture cannot be understood. The authors should, therefore, try to make the paper more self-contained.

The paper addresses an interesting and important topic and shows promising results. However, I would like to invite the authors to revise their manuscript to address specific concerns (before a final decision is reached). Therefore, I invite the authors to revise the paper's focus, which should be on the quality of the imputation rather than dividing the focus between the quality of the imputation and the performance of the downstream machine learning task. In addition to blurring the discussion of the former, the latter does not contribute anything to the assessment of imputation quality. Furthermore, the paper gives no good reason why classification was selected as the machine learning task. Why not prediction?! Why not both?!

Although the main contribution of the paper is a new discrepancy measure – Sliced Wasserstein Distance – that can be used to measure how well features with missing values have been reconstructed in a dataset after the imputation of missing values, no theoretical proof was provided that the Sliced Wasserstein Distance gives, on average, better results than the Sliced KL Distance and the Sliced KS Distance. In principle, such proof is unattainable. Thus, the authors decided, and well, to perform an empirical evaluation. However, the empirical evaluation presented in the paper raises several concerns, to name but a few:

- a) Few datasets were used, and all have binary target features/variables, which weakens the evaluation.
- b) From the “Dataset Description and Preprocessing Details” Section of Supplementary Materials/Information (see page 18/48), it seems that the data for the (seven) selected continuous/numerical variables – “blood pressure (systolic, diastolic and mean), heart rate, oxygen saturation, respiratory rate, and temperature” – of the MIMIC-III dataset has missing values. Then, in a preprocessing step, some patients were dropped because they “had fewer than 5 observations in any of aforementioned variables”. If this is correct, it is incorrect to consider the resulting dataset the ground truth and then draw conclusions from that.
- c) Taking as granted that the observation made in the previous point is correct, the only (literally) complete dataset is the synthetic dataset which limits, to one out of four datasets, the evaluation by comparing against the ground truth.
- d) The dataset extracted from the MIMIC-III dataset and the synthetic dataset have only

continuous/numerical variables which limit the spectrum of the evaluation (no categorical features).

e) Although the datasets are not completely characterized (e.g., features distributions, outliers, skewness, and kurtosis), the authors decided to apply one solution fits all in terms of data normalization – “For all methods, the data in the development and holdout sets are normalized to mean zero and unit standard deviation (SD) using the development set mean and SD.”. Usually, data preprocessing transformations are applied based on the characteristics of a given dataset, and one should avoid applying blindly a data transformation because that will have an impact when using the (transformed) data. For instance, a data preprocessing transformation may have a considerable impact on the performance of third-party tools (e.g., MICE, GAIN, and so forth) or the downstream ML task. Therefore, applying a one solution fits all in terms of data normalization jeopardizes the quest of the paper – “does imputation quality matter?”.

f) In principle, amputation (i.e., the “Induced missingness”) was done in the “Pre-Processing of the Datasets” stage in the data pipeline shown in Figure 35. However, it is not clear if this data transformation was also applied to the target feature/variable of the simulated and the MIMIC-III datasets. If it was not, what was the rationale behind that decision? If it was, why confusion matrices were not computed to assess the performance of the imputation methods/tools?

g) If no detail was overlooked in the paper and the Supplementary Materials, it is not possible to know which versions of the imputation tools were used or when and from where those tools were downloaded. Even worse, one does not know which setup/configuration was used to run each of those tools. The authors did a considerable effort to find the best setup/configuration of the downstream classification tasks (by performing a grid search) but none for the imputation methods/tools. Therefore, one must assume that default configurations were used, but this has an impact on the performance of the imputation methods/tools. Yet, conclusions about the performance of those tools were drawn.

These concerns lead to the conviction that the paper does not present a thorough empirical evaluation of the proposed novel discrepancy measure. Consequently, one should take with a pinch of salt any conclusion that was drawn from the presented results.

The paper could also benefit from improving Section 2.7 – “Measuring the quality of imputation”. Although I did not fully understand part of the notation (related to I_p and J_p , see “Step 1” of the algorithm provided on page 5/48), Section 2.7 would benefit from having:

- Pseudocode representation of the algorithm
- A minimal but elucidative graphical example, using a toy dataset (for instance, 7 samples and 4 features), that eases the perception of each of the three steps.

Finally, more comments and suggestions are provided in a PDF file.

Reviewer #3 (Remarks to the Author):

This is an interesting study that claims to answer two research questions: 1. How to select imputation methods and classifier configurations, to train models using incomplete data. 2. How to best evaluate the imputation methods. The authors claim to introduce a class of discrepancy scores inspired by the sliced Wasserstein distance for evaluating how well imputed data faithfully reconstructs the overall distribution of feature values, and they shared their code on a GitLab Repo.

The authors compared the performance of five different imputation methods namely: mean imputation, multivariate imputation by chained equations (MICE), MissForest, generative adversarial imputation networks (GAIN), and the missing data importance-weighted autoencoder (MIWAE). They also compared five classification algorithms namely: logistic regression, Random Forest, XGBoost, NGBoost, and an artificial neural network.

Improvement recommendations:

1. The authors are referring to a specific cohort extracted from MIMIC III as described in supplementary material and not the whole dataset, specific reference citation need to be provided in the main manuscript as well (minor).
2. As per the cited study the preprocessing step includes missing data imputation, I'd highly recommend the authors to clearly describe the process they followed for the MIMIC cohort definition, especially that the experiment design and conclusion are highly linked to the assumption that MIMIC III cohort does not have any original missing data (major).
3. An overall study conclusion is missing, the reported results demonstrate that the best-performing imputation method and classifier will vary based on the dataset and the prediction task, which is logically expected. I'd highly recommend the authors to summarize the key findings in one or two statements to clearly convey the study innovation.(major)
4. Additionally, the authors need to highlight the advantage of their study compared to a number of additional studies besides those cited by the authors such as "Strategies for Handling Missing Data in Electronic Health Record Derived Data" by Brian J. Wells, et al, and <https://www.frontiersin.org/articles/10.3389/fgene.2021.691274/full>.

Referee Report

COMMSMED-22-0247A

“Classification of datasets with imputed missing values: does imputation quality matter?”

The paper gives a summary of findings from an extensive comparison of several imputation methods for handling missing data using four datasets. Such a comparison study is useful for practitioners as it will give a good guidance to their daily practice. While the paper is well-written, I have a few suggestions to improve the quality of the paper.

1. In addition to Little and Rubin (2019), Kim and Shao (2021) should also be cited as a reference book for handling missing data and imputation.
2. The authors consider five imputation methods, but MICE is the only statistical approach. Please consider fractional hot deck imputation (FHDI) or P-FHDI (Yang et al., 2022) also in the comparison.
3. The main finding of the paper is that the MICE consistently outperforms the ML-based imputation method for classification. This finding is also reported in Wang et al. (2022). The authors may want to cite this paper.

References

- Kim, J. K. and J. Shao (2021). *Statistical Methods for handling Incomplete Data* (2nd ed.). CRC Press.
- Little, R. J. A. and D. B. Rubin (2019). *Statistical Analysis with Missing Data* (3rd ed.). Wiley.
- Wang, Z., O. Akande, J. Poulos, and F. Li (2022). Are deep learning models superior for missing data imputation in large surveys? evidence from an empirical comparison. *Survey Methodology*. accepted (<https://arxiv.org/abs/2103.09316>).

Yang, Y., I. H. Cho, and J. K. Kim (2022). Parallel fractional hot deck imputation and variance estimation for big incomplete data curing. *IEEE transactions on Knowledge and Data Engineering* 34, 3912 – 3926.

Classification of datasets with imputed missing values: does imputation quality matter?

Tolou Shadbahr^{1,+}, Michael Roberts^{2,3,+,*}, Jan Stanczuk^{2,+}, Julian Gilbey^{2,+}, Philip Teare^{3,+}, Sören Dittmer^{2,14}, Matthew Thorpe⁴, Ramon Viñas Torné⁵, Evis Sala⁶, Pietro Lió⁵, Mishal Patel^{3,7}, AIX-COVNET Collaboration⁸, James H. F. Rudd⁹, Tuomas Mirtti^{1,10,11}, Antti Sakari Rannikko^{1,11,12}, John A. D. Aston¹³, Jing Tang¹, and Carola-Bibiane Schönlieb²

¹Research Program in Systems Oncology, Faculty of Medicine, University of Helsinki, Helsinki, Finland

²Department of Applied Mathematics and Theoretical Physics, University of Cambridge, Cambridge, UK

³Data Science & Artificial Intelligence, AstraZeneca, Cambridge, UK

⁴Department of Mathematics, University of Manchester, Manchester, UK

⁵Department of Computer Science and Technology, University of Cambridge, Cambridge, UK

⁶Department of Radiology, University of Cambridge, Cambridge, UK

⁷Clinical Pharmacology & Safety Sciences, AstraZeneca, Cambridge, UK

⁸A list of authors and their affiliations appears at the end of the paper

⁹Department of Medicine, University of Cambridge, Cambridge, UK

¹⁰Department of Pathology, University of Helsinki and Helsinki University Hospital, Finland.

¹¹iCAN-Digital Precision Cancer Medicine Flagship, Helsinki, Finland.

¹²Department of Urology, University of Helsinki and Helsinki University Hospital, Helsinki, Finland

¹³Department of Pure Mathematics and Mathematical Statistics, University of Cambridge, Cambridge, UK

¹⁴ZeTeM, University of Bremen, Bremen, Germany

*corresponding author, email: michael.roberts@maths.cam.ac.uk

+these authors contributed equally to this work

ABSTRACT

Classifying samples in incomplete datasets is a common aim for machine learning practitioners, but is non-trivial. Missing data is found in most real-world datasets and these missing values are typically imputed using established methods, followed by classification of the now complete, imputed, samples. The focus of the machine learning researcher is then to optimise the downstream classification performance. In this study, we highlight that it is imperative to consider the quality of the imputation. We demonstrate how the commonly used measures for assessing quality are flawed and propose a new class of discrepancy scores which focus on how well the method recreates the overall distribution of the data. To conclude, we highlight the compromised interpretability of classifier models trained using poorly imputed data. All code and data used in this paper are also released publicly at [inserted upon publication].

1 Introduction

Datasets with missing values are ubiquitous in many applications, and feature values can be missing for many reasons. This may be due to incomplete or inadequate data collection, corruption of the dataset, or differences in the recording of data between different cohorts or sources. Furthermore, these reasons do not always remain constant or consistent across data sources. Each of these sources of missingness can also introduce different types of missingness, e.g. missing completely at random (MCAR), missing at random (MAR) and missing not at random (MNAR)¹.

Training machine learning classification models is non-trivial if the underlying data is incomplete, as many methods require complete data². Simply excluding incomplete samples could lead to both a significant reduction in statistical power and also risks introducing a bias if the cause of the missingness is related to the outcome. The typical solution to this problem is to follow a two-stage process, firstly imputing the missing values and then using a machine learning method to classify the now-complete dataset. There is an extensive literature discussing imputation methods, most of which is focussed around imputation of data at a single time-point (as we focus on). However, more nuanced scenarios have also been considered, in particular for imputation of longitudinal data^{3,4}, imputation of decentralised datasets using a federated approach⁵ and imputation of variables which have a hierarchical (multi-level) relationship to one another⁶.

Despite the two-stage methods forming the bedrock of approaches to making predictions from incomplete data⁷⁻¹⁰,

it is still unclear which approaches perform ‘best’, and how this should be measured. In particular, it is unclear how the classification model’s performance is influenced by the underlying imputation method and how performance is affected by levels of missingness in the data. When fitting a model to incomplete data using a two-stage approach, the primary aim of many studies is to optimise the downstream classification performance, rather than carefully assessing if the imputed data reflects the underlying feature distribution. However, the latter is profoundly important, as a model trained using poorly imputed data could assign spurious importance to particular features. When we artificially induce missingness into a complete dataset, the quality of an imputation method is typically measured by comparing the imputed values with the ground truth using common metrics such as the (root / normalised) mean square error (MSE)^{2,5,11–18}, mean absolute (percentage) error^{2,13–15} or (root / normalised) square deviance^{3,5,11}. However, as other authors have commented⁶, optimal results for some metrics are achieved even when the distribution of the imputed data is far from the true distribution, see Figure 11.

There are a wide range of discrepancy scores for measuring imputation quality considered in the literature, with a distinction made for those used with categorical data and those used for continuous data. For example, for categorical data, one could consider the proportion of false classifications and Cramér’s V metric. For continuous data there are many additional metrics and discrepancy scores such as, the two-sample Kolmogorov–Smirnov statistic, the (2-)Wasserstein distance (Mallows’ L^2) and the Kullback–Leibler divergence. In particular, the paper of Thurrow et al.¹⁷ considers imputation quality in detail, reporting many of these discrepancy scores, on a feature-by-feature basis to identify distributional differences between the imputed and true missing values.

In this paper, we systematically and carefully address two open research questions. Firstly, for two-stage methods, how should one choose the optimal configuration (e.g. imputation method and classifier) for a particular dataset, to ensure optimal classifying performance for the incomplete data? To address this, we systematically evaluate the performance of several two-stage methods for classifying incomplete data. Using multi-factor ANOVA analysis, we quantify how the downstream classification performance is influenced by the imputation method, classification method and data missingness rate. Secondly, how faithfully do different data imputation methods reproduce the distribution of the underlying dataset? This is a crucial question and requires careful and extensive evaluation. We assess imputation quality using standard metrics, such as RMSE, MAE and the coefficient of determination (R^2), and also introduce a class of discrepancy scores inspired by the sliced Wasserstein distance¹⁹ for evaluating how well imputed data faithfully reconstructs the overall distribution of feature values. We demonstrate how the new proposed class of measures is more appropriate for assessing imputation quality than existing popular discrepancy statistics. We explore the link between imputation quality and downstream classification performance and show the remarkable result that a classifier built on poor imputation quality can actually give satisfactory downstream performance. We postulate that this could be due to the ability of powerful methods to overcome issues in the imputed data, or that a poor imputation is equivalent to injecting noise into the dataset. Training machine learning-based models on such noisy data can be viewed as a form of data augmentation known to improve generalisability and robustness of the models²⁰.

The stability of different imputation methods is explored and we found that the algorithms consisting of neural network components are susceptible to local minima and can give highly variable imputation results. In addition, we find that the popular discrepancy measures used to assess imputation quality are uncorrelated from the downstream model performance, whereas the measures which consider distributional discrepancies do show a correlation. Finally, we demonstrate how high-performing classification models trained using poorly imputed data assign spurious importances to particular features in the dataset. We release a codebase to the community, with example datasets, at [github.com/\[shared upon publication\]](https://github.com/[shared upon publication]) which provides a framework for practitioners to allow for easy, reproducible benchmarking of imputation method performance, for evaluating a wide range of classifiers along with assessing imputation quality in a completely transparent way.

2 Materials and Methods

In this section we briefly introduce the datasets used in the study, the benchmarking exercise and the methods for assessing imputation quality.

2.1 Datasets

In this study, we focus on four datasets denoted as **Breast Cancer**, **MIMIC-III**, **NHSX COVID-19**, and **Simulated**. The first three of these are derived from real clinical datasets, while the final one is synthetic. The **MIMIC-III** and **Simulated** datasets are complete (i.e. do not contain missing values), giving us control over the induced missingness type and rate. The **Breast Cancer** and **NHSX COVID-19** datasets exhibit their natural missingness.

- **Simulated** is a synthetic dataset created using the `scikit-learn`²¹ function `make_classification`, giving a dataset with 1000 samples and 25 informative features. This allows us to perform a simulation study to determine which factors can potentially influence classification after imputation.

- **MIMIC-III.** The Medical Information Mart for Intensive Care (MIMIC) dataset²² is a large, freely-available database comprising deidentified health-related data from patients who were admitted to the critical care units of the Beth Israel Deaconess Medical Center in the years 2001–2012. Following the preprocessing detailed in the Supplementary Materials, we obtain the **MIMIC-III** dataset used in this paper. This contains data for 7214 unique patients with 14 clinical features and a survival outcome recorded for each patient.
- **Breast Cancer.** This dataset is derived from an oncology dataset collected at Memorial Sloan Kettering Cancer Center between April 2014 and March 2017²³. The dataset contains genomic profiling of 1918 tumor samples from 1756 patients with detailed clinical variables and outcomes for each patient and the therapy administered over the time of treatment. The **Breast Cancer** dataset is obtained after the preprocessing described in the Supplementary Materials and has 16 features for 1756 patients with the natural missingness of the data retained.
- **NHSX COVID-19.** This dataset is derived from the NHSX National COVID-19 Chest Imaging Database (NCCID)²⁴ which contains clinical variables and outcomes for COVID-19 patients admitted in many hospitals around the UK. The dataset is continually updated by the NHSX; the download we performed was on 5 August 2020 which contained data for 851 unique patients. The **NHSX COVID-19** dataset contains 23 features for 851 patients.

Outcome variables. For the **MIMIC-III**, **Breast Cancer** and **NHSX COVID-19** datasets we use survival status as the outcome of interest, the **Simulated** dataset outcome is a binary variable generated at the data synthesis stage by the `make_classification` function in `scikit-learn`²¹.

2.2 Dataset partitioning

In our experiments, we simultaneously compare many combinations of imputation and classification methods, each with several hyperparameters that can be tuned. Therefore, we must be careful to avoid overfitting. To address this, we partition each dataset at two levels. At the first level, we identify the development and holdout cohorts and at the second, we partition the development sets into training and validation cohorts (see the Supplementary material for more details.)

Induced missingness. The **Simulated** and **MIMIC-III** datasets are complete and, therefore, we can induce different missingness rates in the development and holdout datasets. We randomly removed 25% and 50% of entries from each of the development and holdout datasets. This allows us to compare 4 different scenarios for development-holdout missingness rates: 25%-25%, 25%-50%, 50%-25% and 50%-50%.

2.3 Imputation methods

Data imputation is the process of substituting missing values in a dataset with new values that are, ideally, close to the true values which would have been recorded if they had been observed. To formalise the notation, we consider a dataset \mathcal{D} , consisting of N samples $\mathbf{x}_i \in \mathbb{R}^d$ drawn from some (unknown) distribution. Some of the elements in these samples may be missing and we impute them to give the complete samples $\hat{\mathbf{x}}_i$. Whilst it is not possible to be certain about the true value of the missing entries, it is desirable that the uncertainty in the imputed values be considered any downstream task which relies on the imputed data²⁵.

In general, imputation methods fall into two categories, namely ‘single imputation’ and ‘multiple imputation’. In the case of single imputation, plausible values are imputed in place of the missing values just once, whereas for multiple imputation methods^{26,27}, imputation is performed multiple times to generate a series of imputed values for each missing item. For imputation methods that have some stochastic nature, this allows for insight into the uncertainty of the imputed values. For multiple imputation methods, there are two approaches for performing a downstream classification task. The first approach involves pooling the multiple imputation results and creating a single, summary, dataset from which we perform the classification task. The second approach, which we follow in this paper, is to perform the classification task on each of the multiple imputation results separately, before pooling the predictions of the classification model. The final prediction is made using e.g. averaging or majority vote. See van Buuren⁶, for a comparison of the approaches and justification for the latter being preferable.

In our study, we consider five popular imputation methods from the literature, namely mean imputation, multivariate imputation by chained equations (MICE)^{28,29}, MissForest³⁰, generative adversarial imputation networks (GAIN)³¹, and the missing data importance-weighted autoencoder (MIWAE)³². Among the four datasets we consider, **Simulated** and **MIMIC-III** contain only numerical features whereas **Breast Cancer** and **NHSX COVID-19** contain numerical, categorical (single and multi-level) and ordinal features. In preprocessing, the categorical features are one-hot encoded and the ordinal features are coded with integer values.

2.4 Classification methods

Classification is the process of grouping items with similar characteristics into specific classes. In our case, the classification methods are machine learning-based algorithms which take an input sample $\hat{\mathbf{x}}_i \in \mathbb{R}^d$ and output a particular class label

$\ell_i \in \{0, 1\}$. The classification methods we consider in this paper are logistic regression, Random Forest, XGBoost, NGBoost and an artificial neural network (for more details see the Supplementary Materials.)

2.5 Hyperparameter optimisation

Each of the classifiers that we consider (except for logistic regression) have several tuneable hyperparameters, with classifier performance significantly dependent on them. In order to identify the best classifier for each dataset, we perform an exhaustive grid search over a wide range of hyperparameters. For each configuration of dataset, train/test missingness level, imputation method, holdout set, validation set and each of the multiple imputations, we fit a classifier, exhaustively, over a large range of hyperparameters as detailed in Supplementary Table 1. To identify the optimal hyperparameter choice for each of the these configurations, we evaluate the model on each of the five validation folds. The hyperparameter choice which results in the best average performance over these validation folds is selected as the optimal model configuration i.e. for each holdout set H1–H3 (Fig. 12) we obtain an optimal model. This exhaustive grid search required millions of experiments to ensure the comparisons are fair.

2.6 ANOVA analysis

One of the key aims of this paper is to identify and quantify the influence of key factors on the performance of the downstream classification. To quantify the impact, we used a generalized linear model (binomial logistic regression-based) to perform multi-factor ANOVA for the holdout AUC of the optimal classifiers identified in §2.5. Firstly, to establish the factors most influencing the models built on the real clinical models, we pool the results for the **MIMIC-III**, **Breast Cancer**, and **NHSX COVID-19** datasets and perform a logistic ANOVA. Additionally, each of the datasets are assessed individually. After constructing the ANOVA model including all factors and their interaction, we excluded factors not significant at the 1% level using backward elimination (see the Supplementary Materials for more details).

2.7 Measuring the quality of imputation

In addition to determining how the imputation method, missingness rates and datasets affect the downstream classification performance, we are also interested in exploring how the quality of the imputation affects the downstream classification performance. However, there is no widely accepted approach for measuring the quality of an imputation.

In the literature, many authors, such as Jadha et. al.³³ and Thurow et. al.¹⁷, assess imputation quality by taking a complete dataset, introducing missingness artificially and imputing the resulting incomplete dataset; we follow this approach here. For this purpose, we induced missingness completely at random (MCAR) with rate 25% and 50% into the **MIMIC-III** and **Simulated** datasets. In order to quantitatively assess how well an imputation method reconstructs the missing values, we must define a score that has the desirable property that it has a low value when the distribution of imputed values closely resembles that of the true values. Explicitly, our aim is to compute a distance between the original samples $\mathcal{D} = \{\mathbf{x}_i\}_{i=1}^N$ and the imputed samples $\hat{\mathcal{D}} = \{\hat{\mathbf{x}}_i\}_{i=1}^N$ that will indicate the quality of the imputation.

In this paper, we consider many of the popular discrepancy statistics used in the literature for measuring imputation quality. These typically fall into two classes: (A) measures of discrepancy between the imputed and true values of individual samples, and (B) measures of discrepancy in distributions of individual features for the imputed and true data. The class of measures which would be of most practical value to practitioners is (C) measures of discrepancy for imputed and true data across the whole data distribution. In the literature, we were unable to find any examples of discrepancy measures of this type and we propose such a class in this paper.

A. Sample-wise discrepancy. In much of the literature, the quality of imputation is determined by measuring the discrepancy in the real and imputed values sample-by-sample, then summarising over all samples in the dataset. In this paper, we consider the three discrepancy statistics, Root mean square error (RMSE), mean absolute error (MAE), and the coefficient of determination (\mathbf{R}^2). These statistics compare the imputed values explicitly to the true values.

B. Feature-wise distribution discrepancy. In the literature, some authors have considered discrepancy measures to quantify for how faithfully the distributions of individual features are reconstructed. In particular, Thurow et al.¹⁷ consider several distribution measures on a feature-by-feature basis, including the Kullback–Leibler (KL) divergence, the two-sample Kolmogorov–Smirnov (KS) statistic and 2-Wasserstein (2W) distance. We report results for all of these in this study. As these discrepancies are measured feature-by-feature, we get many scores for each dataset and report the minimum, maximum and median discrepancy for each distance over all features. More details about these statistics, including definitions and the implementations used, can be found in the Supplementary Materials.

C. Sliced Wasserstein distance. In Figure 1, we see that simply considering the feature-by-feature marginal distributions is not sufficient to quantify how well a high-dimensional data structure has been imputed. The marginal distributions of the imputed data (directions 1 and 2) match that of the original data perfectly but do not identify the discrepancy of the distributions shown in Figures 1(a) and (b). This motivates us to consider a new measure, which harnesses the multi-dimensional nature of the data to better identify distribution differences like this.

Figure 1. The effects of projecting imputed data in multiple directions. (a) The underlying 2-dimensional data distribution; (b) the distribution of imputed data; (c) some example directions: 1 and 2 are in the direction of the features, directions 3 and 4 are not; (d) and (e) show the original and imputed data distributions projected onto the four directions shown in (c): their marginals (directions 1 and 2) are indistinguishable but the distributions are clearly different when projected in directions 3 and 4.

Modelling the discrepancy of imputed and true data in high dimensions is challenging for two key reasons. Firstly, the curse of dimensionality results in computations which are infeasible for high-dimensional datasets. Secondly, high-dimensional datasets are sparse unless there are an unrealistically large number of samples. We address both of these issues by repeatedly projecting the entire data distribution to random one-dimensional subspaces, thereby increasing the density of the data while considering more axes than those simply defined by the features themselves. We then consider the distance of the imputed data from the original data in the random direction, commonly known as the sliced Wasserstein distance^{34,35}; this gives a distribution of distances across all randomly chosen directions. Initially, we perform the following steps:

- Step 1: Determine partitions and random directions.

Choose M random unit vectors (directions) $\mathbf{n}_r \in \mathbb{R}^d$, $r = 1, \dots, M$, where $M \geq d$. Choose P random partitions of the index set $\{1, 2, \dots, N\}$ into subsets I_p and J_p for $p = 1, \dots, P$ (where N is the number of samples). If N is even, then I_p and J_p are the same size, while if N is odd, these subsets have sizes $(N + 1)/2$ and $(N - 1)/2$ respectively. To account for the dimensionality differences between **MIMIC-III** and **Simulated**, we used $M = 50$ and $M = 90$ respectively. We set $P = 10$ throughout.

- Step 2: Calculate projections of data.

For each r , we project all of the data onto the one-dimensional subspace of \mathbb{R}^d spanned by \mathbf{n}_r ; this gives the projected original data $\mathbf{x}_i \cdot \mathbf{n}_r$ and projected imputed data $\hat{\mathbf{x}}_i \cdot \mathbf{n}_r$.

- Step 3: Calculate sliced Wasserstein Distances.

For each pair (r, p) , with $r \in \{1, \dots, M\}$ and $p \in \{1, \dots, P\}$, we calculate (2-)Wasserstein distances between the projected original and imputed data. The data are normalised by dividing through by the standard deviation, s , of the projected data $\mathbf{x}_i \cdot \mathbf{n}_r$ for $i \in I_p$. The data points in I_p are taken as the ‘true’ distribution and we determine the distance $w(r, p)$ of the remaining data in J_p from this. This is our baseline distance inherent to the data. We then calculate $\hat{w}(r, p)$, the 2-Wasserstein distance between $\mathbf{x}_i \cdot \mathbf{n}_r / s$ for $i \in I_p$ and the imputed data $\hat{\mathbf{x}}_j \cdot \mathbf{n}_r / s$ for $j \in J_p$.

Over all (r, p) pairs, these steps result in two distributions of distances, for $w(r, p)$ and $\hat{w}(r, p)$, as illustrated in Figure 2. Firstly, these can be regarded as probability distributions, allowing us to compute the same class B feature-wise discrepancy scores discussed previously. Secondly, we evaluate the relative change in the Wasserstein distance due to imputation for each r and p , namely $\hat{w}(r, p)/w(r, p)$, allowing us to quantify how much different imputation methods induce discrepancy in the data distribution. Finally, we assess the stability of the different imputation methods by exploring the variance in the induced sliced Wasserstein distance across repeated imputations.

Figure 2. Procedure for deriving the sliced Wasserstein discrepancy statistics.

2.8 Downstream effects of imputation quality

A critical question that this paper aimed to answer is: in what way does the quality of the data imputation affect the downstream classification performance? Firstly, we determined whether there is a correlation between the discrepancy scores in classes A–C and the performance of the **predictive** model. Secondly, we investigated whether a classification model, trained using poorly imputed data, could use unexpected features to make its **predictions**. To answer this, we analyse the models fit to the **Simulated** dataset. This is an ideal dataset, as the features are of equal importance by design, the clusters of values within each feature are normally distributed, centered at vertices of a hypercube and separated²¹.

For each classifier, we identify two models which perform well at the classification task but where one is trained using poorly imputed data and the other is fit to data which is imputed well. See the Supplementary Materials for details on the model selection. To assess the feature importance for each of the trained models, we used the popular Shapley value approach of Lundberg et al.³⁶ For each feature, this assigns an importance to every value, identifying how the **prediction** is changed on the basis of the value of this feature. A positive importance value indicates the value influences the model towards a positive **prediction**. Due to the design of the **Simulated** dataset, we expect the distribution of the Shapley values to be symmetric, as the clusters are separated by a fixed distance and are normally distributed within those clusters. Using the skewness, we measure the symmetry of the feature importances for each feature. The Python package, for computing the Shapley values, cannot currently support neural network and logistic regression models, but we consider them for all other models.

3 Results

Classifier influence on downstream performance. In Figure 3(a), we show how the classifier affects the downstream performance for each dataset across different train and test missingness rates. For the **MIMIC-III** and **Simulated** datasets, the classifier trained using the complete original dataset has performance which always exceeds that of those built on imputed data (with the sole exception of the Neural Network for **MIMIC-III**). The performances for the **Simulated** dataset are much higher than for the real datasets as, by design, there is a direct link between outcome and the feature values. For the **Simulated** and **MIMIC-III** datasets, we see that increasing the train and test missingness rates leads to a decline in performance. For all classifiers, with a fixed train missingness rate, a change in the test missingness rate affects performance very significantly compared to changing the train missingness rate for a fixed test missingness rate. For each imputed dataset, we see that when varying the missingness rates of the **development** and test data, the ranking of each classifier’s performance is almost consistent, e.g. for the **Simulated** dataset the Neural Network always performs the best while Random Forest performs the worst. The XGBoost classifier performs best for the **MIMIC-III** and **NHSX COVID-19** datasets, with consistency in performance ranking at all missingness rates. The **Breast Cancer** dataset follows the trend of the other real-world datasets, with the exception that the logistic regression classifier performs the best. We see that increasing the test missingness rate leads to an increase in the standard deviation. For most datasets, the variance of the classifier performance is similar across the classifiers, the exception being the **Breast Cancer** dataset for which the Random Forest has a small variance and Neural Network a relatively large variance.

Imputation influence on downstream performance. In Figure 3(b), we show the dependence of the downstream classification performance on the imputation methods used to generate the complete datasets. For the **Simulated** and **NHSX COVID-19** datasets, MIWAE imputation gives the best downstream performance for all levels of **development** and test missingness rates. For the **Simulated** dataset, as the missingness rate increases, the performance of models trained using MissForest imputed data significantly declines. In the real-world datasets, there is no ‘best’ imputation method that consistently leads to a model

which outperforms the others. For the **MIMIC-III** dataset, we see that the simple mean imputation method gives the worst downstream performance for all missingness rates but for **NHSX COVID-19** is competitive with the best performing MIWAE method. For the **MIMIC-III** dataset, MICE imputation leads to the best performance for most missingness rates but is the worst for the **Breast Cancer** dataset. The variance in the performances is consistent across the real-world datasets, in the range $[0.01, 0.02]$, but is significantly higher for all levels of missingness in the **Simulated** dataset, in the range of $[0.02, 0.04]$.

Figure 3. Dependence of downstream classification performance on classification and imputation methods. These plots show the dependence of the downstream performance on the classification and imputation methods. For the **Simulated** and **MIMIC-III** datasets we show performance for 25% and 50% levels of missingness in the **development** and test data. The size of each marker corresponds to the standard deviation

ANOVA for downstream classification performance. In Figure 4, we show the significant factors identified for the pooled data and individual datasets. For the pooled data, it is clear that the missingness rate of the test set explains most of the deviance in the results of our **predictive** models, confirming the observations previously derived from Figure 3. The dataset under consideration and the classification method used are the next most important factors. The ANOVA is also performed for the individual datasets, shown in Figure 4(b)–(d). For the **Simulated** dataset, we see that many factors affect the classification performance. Only the **Simulated** and **MIMIC-III** datasets have induced missingness, and we see that the test set missingness rate is the most significant source of deviance in the downstream classification performance, followed by the choice of classification

method. For both the **NHSX COVID-19** and **Breast Cancer** datasets, the classification method is the primary source of deviance. Indeed for the **Breast Cancer** dataset, it is the only significant factor at the 1% level.

Figure 4. Pooled and dataset-segregated ANOVA analysis. In these plots we show the significant factors in the ANOVA analysis for the pooled and segregated datasets (we do not display for **Breast Cancer**, as the choice of the classifier is the only significant factor.).

Comparing imputation quality

A. Sample-wise statistics. In Figure 5 and Supplementary Figures 18–19 the sample-wise discrepancy scores are shown for the **MIMIC-III** and **Simulated** datasets for different train and test missingness levels. RMSE and MAE are generally consistent across the different imputation methods for fixed train and test missingness rates. There is a minimal performance difference between holdout sets. Using these sample-wise measures for the **MIMIC-III** dataset, the MissForest imputation method performs the best overall, followed by GAIN. The gap between the performance of MissForest and GAIN narrows as the train and test missingness rates increase. Mean imputation performs worst at 25% test missingness, whilst at 50%, MICE is the worst. For the **Simulated** dataset, the best performing methods are MissForest and mean imputation whilst MICE is the worst. MICE generally attains the lowest R^2 score at all missingness rates for both datasets. The poor performance of MICE by the RMSE metric can also be seen in Figure 11 although, qualitatively, the distribution is better recreated using this imputation method.

B. Feature distribution metrics. In Figure 6 and Supplementary Figures 20–26 we show the minimum, median and maximum feature-wise discrepancies statistics for the **MIMIC-III** and **Simulated** datasets. The mean imputation method is the worst in all metrics for minimum, median and maximum at all rates of train and test missingness in both datasets. MICE imputation, which performed worst by the sample-wise discrepancy scores, is the best performing method by the Kolmogorov-Smirnoff statistic and Wasserstein distance for all missingness rates across all the minimum, median and maximum discrepancies. It is generally the best method by Kullback-Leibler divergence, with MissForest and MIWAE also competitive. For GAIN, the minimum discrepancies are competitive with the other imputation methods. However, when considering the maximum

Figure 5. The sample-wise statistics for the MIMIC-III dataset with 25% train and test missingness rates.

discrepancy, we note that the difference in performance of the mean and GAIN methods narrows significantly. Increasing the test missingness rate leads to a significant increase in the feature-wise distances, whereas there is a more subtle increase in the distances with an increase in the train missingness rate from 25% to 50%.

Figure 6. The feature-wise statistics for the MIMIC-III dataset with 25% train and test missingness rates.

C. Sliced Wasserstein distribution metrics. In Figure 7 and Supplementary Figures 27–30, we show the discrepancies between

the distributions of the sliced Wasserstein distances for the imputation methods at different train and test missingness rates for the **MIMIC-III** and **Simulated** datasets. For the **MIMIC-III** dataset, over all discrepancy scores and missingness rates, the MICE imputation method shows a clear dominance with the mean and GAIN methods performing poorly. For the **Simulated** dataset, the MICE method again performs the best overall by all measures. At the 25% test missingness rate the MIWAE imputation method is competitive with, and sometimes outperforms, MICE. Mean imputation performs the worst, with GAIN and MissForest performing similarly poorly.

Figure 7. The distribution discrepancy statistics of the sliced Wasserstein distances for the **MIMIC-III** dataset with 25% train and test missingness rates.

Sliced Wasserstein distance ratio analysis. In Supplementary Figures 32–33 are the boxplots of the ratios of the distances from J_p to I_p for the imputed data compared to the original data for the **MIMIC-III** and **Simulated** datasets respectively. Firstly, we see that the MICE imputation method induces a much smaller distance ratio than any other method. Secondly, we note that with an increase in the train missingness rate the ratio of the distances remains largely consistent, however with an increase in the test missingness rate, we see a very significant increase in the ratio.

Outlier analysis. It is important to understand how stable the imputation methods are when imputation is repeated and also whether the stochastic nature of the imputation methods can lead to outlier imputed values for particular features. In Figure 7 and the Supplementary Figures 27–28 we see that some of the imputation methods can lead to distances with large variances, especially as the missingness rates in the train and test sets increase. This variance can result from either the random projections (in some cases the distributions match well and in others quite poorly) or stochasticity in the imputation algorithm. We are keen to understand the influence of each. In Figure 8 and Supplementary Figure 13, we see that the MICE method performs consistently well across all holdout and validation sets with no imputations above the distance of 10^{-7} and at the threshold of 1.5×10^{-8} , 90% of the MICE imputations are above this distance. This demonstrates a consistency in MICE imputations, with most imputations at a distance between 1.5×10^{-8} and 10^{-7} from the true values.

Figure 8. The proportion of repeated imputations that give outlier Wasserstein distances, at threshold 10^{-7} , for different imputation methods.

Link between imputation quality and downstream classification performance. In Figure 9 and Supplementary Figure 31, we plot the results for all class A, B and C imputation discrepancy statistics against the AUC of the downstream classification task. Our previous analysis has shown that the test missingness rate has a large influence on both imputation quality and downstream performance, so we display the correlations separately for 25% and 50% missingness. For the sample-wise statistics, we see a negative correlation between imputation discrepancy and downstream classifier performance only for the **Simulated** dataset. In the **MIMIC-III** dataset, we see no clear correlation between imputation quality, by any of these discrepancy scores, and the classifier performance. However, for the feature-wise and the proposed sliced Wasserstein classes, we see a negative correlation for all statistics in both datasets. In Figure 10 and Supplementary Figure 34 we give heatmaps showing the correlations between the nine different discrepancy statistics used for the **MIMIC-III** and **Simulated** datasets, respectively. Within each class of discrepancy metric (for A, B and C), the measures are all correlated with one another, however the sample-wise metrics (class A) do not correlate significantly with any of the feature-wise (class B) or sliced Wasserstein distances (class C). There is also strong correlation between most of the class B and C metrics.

Figure 9. The different imputation discrepancy metrics from classes A, B and C are shown against the downstream AUC value for the classification task of the **MIMIC-III** dataset. Trend lines are shown for 25% and 50% test missingness separately.

Impact of imputation quality on interpretability. For each classifier, we find that the best sliced Wasserstein distance ratio is always obtained for data imputed with MICE. For NGBoost and XGBoost, the worst distance ratio of the high performing models is for GAIN imputed data whereas, for Random Forest, this is for mean imputed data. For each of the 25 features in

Figure 10. Correlation heatmap for all discrepancy metrics considered in this paper for the **MIMIC-III** dataset.

the **Simulated** dataset, we calculate the absolute value of the skew for the Shapley values and display these in Supplementary Figure 14. For the Random Forest classifier, we see that for 21/25 features, the absolute skew of the Shapley values for the MICE imputation is smaller than that of mean imputation. For NGBoost, we see the a smaller absolute skew in 19/25 features for MICE against GAIN. Finally, for XGBoost, we see a smaller absolute skew in 15/25 features for MICE against GAIN.

4 Discussion

In this paper, we have highlighted the importance for machine learning and data science practitioners to reconsider the quality of the imputed data which is used to train a classifier. Each dataset we considered presents with different missingness rates and missingness types. In particular, the **NHSX COVID-19** and **Breast Cancer** datasets have data that is missing not at random, while the induced missingness to the **MIMIC- III** and **Simulated** datasets is missing completely at random (MCAR). Perhaps as a consequence of this, it was found that there is no particular classifier which performs best across all of the datasets and similarly, no particular imputation method leads to the best downstream classification performance. Even for the datasets with the same types of missingness, no optimal imputation or classification method emerges. Interestingly, classification models built on data imputed using mean imputation demonstrate the largest variance in the downstream performance, even though the imputed data is exactly the same in each of the multiple imputations. Using ANOVA, we found that performance is most influenced by the test missingness rate. This would suggest that when deploying a trained model to a new dataset, it is of primary importance to be conscious of the missingness rate of the new data, as classifier performance is very sensitive to it. ANOVA also highlights a minimal dependence of the downstream classification performance on the imputation method chosen. Therefore, we conclude that it is unlikely that it is possible to heuristically identify the optimal imputation and classification method for a particular dataset.

There is no accepted approach for evaluating the quality of an imputation method. Typically, imputation methods are evaluated using sample-wise discrepancy statistics such as RMSE, MAE and R^2 , but this approach implicitly assumes that the aim of imputation is to recover the true *value* of the missing datapoint, rather than recreating the correct *distribution*. In our experiments, we find that the sample-wise discrepancy scores are not sufficient to assess the quality of the reconstruction of the distribution of imputed values. In fact, for MSE in particular, we have seen that it is optimal for imputations that give a very

poor distribution match. Using MSE to evaluate the imputation quality is only statistically justified in the case of imputing data from a Gaussian distribution (minimizing MSE corresponds to maximizing the log-likelihood). Since the Gaussian assumption often doesn't hold in practice, MSE can be a very inaccurate measure of imputation quality. Beyond sample-wise discrepancy scores, some studies have considered the reconstruction quality of distributions for datasets on a feature-by-feature basis. However, in this study, we have shown how considering these marginal distributions alone is not sufficient to understand how well the overall data distribution has been reconstructed. Echoing van Buuren⁶, "*imputation is not prediction*". The goal of an imputation method is to recover the correct distribution, rather than the exact value of each missing feature. However, the metrics used in the literature simply do not currently enforce this. In this paper, we introduced a class of discrepancy statistics, based on the sliced Wasserstein distance, which allow us to evaluate how well *all* features have been reconstructed in a dataset.

Interestingly, although GAIN performs well in the sample-wise discrepancy scores, we see a poor performance in the discrepancy when measured feature-wise and with our proposed class of sliced Wasserstein distance-based scores. In fact, we find that the GAIN imputation method tends to perform in line with mean imputation and give a poor reconstruction of the underlying data distribution. It has been identified in³⁷ that GAIN, which uses a generative adversarial approach for training, tends to rely mostly on the reconstruction error in the loss term (the mean square error) and the adversarial term contributes in a minimal way. The MICE imputation method has shown a clear dominance for recreating the distribution of the datasets as a whole. In both datasets for which it could be evaluated, and across all train and test missingness rates, it performs better than the competitor imputation methods. When evaluating the ratio between the sliced Wasserstein distance in the imputed data versus the original data, we found that, consistent with the earlier observations, an increased test missingness rate leads to a significant increase in the distance induced by imputation. However, an increase in the train missingness rate leads to a minimal increase in the induced distance ratio. We found that MICE outperforms the other imputation methods, inducing the smallest relative increase in distance with minimal variance. It also gave highly consistent imputation results, with minimal variance in the distances between the imputed and original values. Performing multiple imputations highlights that GAIN and MIWAE suffer from mode collapse in some of the rounds of imputation and can fall into outlying local minima, likely due to the highly non-convex problem they are solving. This is a problem common to all deep neural network architectures and serves as a reminder, for imputation methods, of the importance of using multiple imputation to overcome the risk of some poor quality imputations.

We find that it is not necessarily the best quality imputation that leads to the best performing classification method. This observation could be explained by the fact that imputation does not add any information that was not present in the data set to begin with. Therefore, a powerful classifier may be able to extract all information relevant to the classification task regardless of the imputation quality. Secondly, we conjecture that an inaccurate imputation could in fact provide a form of regularization for the classifier. It has been shown that perturbing the training data with a small amount of Gaussian noise is equivalent to L_2 regularization³⁸ and can be beneficial to the performance in a supervised learning tasks. In the same spirit, an inaccurate imputation method resulting in noisy imputed values could provide a regularizing effect beneficial to performance in the downstream classification task. We also find that not only are the common discrepancy scores used by the community, i.e. the sample-wise statistics (RMSE, MAE, R^2), uncorrelated from the distributional discrepancy metrics (of classes B and C) but that they are also disconnected from the downstream classification performance of the model. This highlights how important it is to consider additional statistics when measuring imputation quality, not simply the sample-wise statistics, as the distribution discrepancy scores occupy a completely orthogonal space to the sample-wise ones. Importantly, we find a correlation between the proposed class of, sliced Wasserstein distance derived, discrepancy scores and the downstream model performance suggesting that a link has been forged from the imputed data to the downstream model performance. Given this, we would suggest that instead of focusing on optimising performance by considering the best combination of imputation method and classifier, attention should shift towards optimising the imputation quality in terms of how well the distribution is reconstructed. In our assessment of how the imputation quality can affect the model interpretability downstream, we find that, for comparably high performing classifiers, those fit to poorly imputed data give misleading feature importance's in the downstream classification task. Any features judged as important from these models are therefore compromised. This is crucial to appreciate, especially where models are fit to clinical data, as we may draw incorrect conclusions about the influence of particular features on patient outcomes. Overall, this suggests that the quality of imputation does feed through to the downstream model interpretability.

In addition to main aims of this paper, through this work, we have also identified some issues that need the attention of the imputation community, such as how categorical and ordinal variables should be correctly imputed. For example, if a categorical variable is one-hot-encoded then a valid imputation must be in $\{0, 1\}$. However, most imputation methods will give a value in the range $[0, 1]$ and these must be post-processed. It is not clear how this should be performed. Similarly, if a category with multiple values is one-hot encoded, e.g. nationality, then only one of these variables should equal one but no imputation method enforces this. Issues were also identified, and fixed, in the public code releases of the GAIN and MIWAE imputation methods and so, to improve the quality of future benchmarking of imputation methods, we release our codebase at

github.com/ [updated upon publication]. This allows for rapid computation of the results for several imputation methods across incomplete datasets with data preprocessing, partitioning and analysis all built in. This should serve as a sandbox for the development, and fair evaluation, of new imputation methods.

It is our hope that by standardising the imputation methods, multiple imputation, classification and analysis pipelines for determining the imputation quality, future studies have all the tools necessary to fit classifiers to incomplete data that are not only achieving high performance, but are built on high quality imputed data. This study highlights how these concepts are not separable and if we do not understand the quality of the imputation, the model fit to the data is compromised.

Acknowledgements

There is no direct funding for this study, but the authors are grateful for the following indirect funding: European Research Council (No. 716063) (T.S., J.T.), Academy of Finland (No. 317680) (T.S., J.T.), BusinessFinland (T.S., J.T.), The Sigrid Juselius Foundation (T.S., J.T.), the EU/EFPIA Innovative Medicines Initiative project DRAGON (101005122) (T.S., M.R., J.S. J.G., S.D., E.S., AIX-COVNET, C.-B.S.), AstraZeneca (M.R., P.T., M.P.), the Trinity Challenge (M.R., C.-B.S.), the EPSRC Cambridge Mathematics of Information in Healthcare Hub EP/T017961/1 (M.R., J.H.F.R., J.A.D.A, C.-B.S.), the Cantab Capital Institute for the Mathematics of Information (J.S., C.-B.S.), Aviva (J.S.), the European Research Council under the European Union's Horizon 2020 research and innovation programme grant agreement no. 777826 (M.T., C.-B.S.), the Alan Turing Institute (M.T., C.-B.S.), Fundación Rafael del Pino (R.V.), Wellcome Trust (J.H.F.R.), The Mark Foundation for Cancer Research (E.S.), Cancer Research UK Cambridge Centre (C9685/A25177) (E.S., C.-B.S.), British Heart Foundation (J.H.F.R.), the NIHR Cambridge Biomedical Research Centre (J.H.F.R.), HEFCE (J.H.F.R.), the Finnish Cancer Organizations (A.S.R.) and the Jane and Aatos Erkko Foundation (A.S.R.). In addition, C.-B.S. acknowledges support from the Leverhulme Trust project on 'Breaking the non-convexity barrier', the Philip Leverhulme Prize, the EPSRC grants EP/S026045/1 and EP/T003553/1 and the Wellcome Innovator Award RG98755. Finally, the AIX-COVNET collaboration is also grateful to Intel for financial support.

AIX-COVNET

Michael Roberts^{2,3}, Sören Dittmer^{2,14}, Ian Selby⁶, Anna Breger^{2,15}, Matthew Thorpe⁴, Julian Gilbey², Jonathan R. Weir-McCall^{6,16}, Effrossyni Gkrania-Klotsas¹⁷, Anna Korhonen¹⁸, Emily Jefferson¹⁹, Georg Langs²⁰, Guang Yang²¹, Helmut Prosch²⁰, Jacobus Preller¹⁷, Jan Stanczuk², Jing Tang¹, Judith Babar¹⁷, Lorena Escudero Sánchez⁶, Philip Teare³, Mishal Patel^{3,7}, Marcel Wassin²², Markus Holzer²², Nicholas Walton²³, Pietro Lió⁵, Tolou Shadbahr¹, James H. F. Rudd⁹, Evis Sala⁶ and Carola-Bibiane Schönlieb².

¹⁵ Faculty of Mathematics, University of Vienna, Austria ¹⁶ Royal Papworth Hospital, Cambridge, Royal Papworth Hospital NHS Foundation Trust, Cambridge, UK ¹⁷ Addenbrooke's Hospital, Cambridge University Hospitals NHS Trust, Cambridge, UK. ¹⁸ Language Technology Laboratory, University of Cambridge, Cambridge, UK. ¹⁹ Population Health and Genomics, School of Medicine, University of Dundee, Dundee, UK. ²⁰ Department of Biomedical Imaging and Image-guided Therapy, Computational Imaging Research Lab Medical University of Vienna, Vienna, Austria. ²¹ National Heart and Lung Institute, Imperial College London, London, UK. ²² contextflow GmbH, Vienna, Austria. ²³ Institute of Astronomy, University of Cambridge, Cambridge, UK.

References

1. Eekhout, I., de Boer, R. M., Twisk, J. W. R., de Vet, H. C. W. & Heymans, M. W. Missing Data: A Systematic Review of How They Are Reported and Handled. *Epidemiology* **23**, 729–732, DOI: [10.1097/EDE.0b013e3182576cdb](https://doi.org/10.1097/EDE.0b013e3182576cdb) (2012).
2. Emmanuel, T. *et al.* A survey on missing data in machine learning. *J. Big Data* **8**, 140, DOI: [10.1186/s40537-021-00516-9](https://doi.org/10.1186/s40537-021-00516-9) (2021).
3. Luo, Y. Evaluating the state of the art in missing data imputation for clinical data. *Briefings Bioinforma.* **23**, bbab489, DOI: [10.1093/bib/bbab489](https://doi.org/10.1093/bib/bbab489) (2022).
4. Huque, M. H., Carlin, J. B., Simpson, J. A. & Lee, K. J. A comparison of multiple imputation methods for missing data in longitudinal studies. *BMC Med. Res. Methodol.* **18**, 168, DOI: [10.1186/s12874-018-0615-6](https://doi.org/10.1186/s12874-018-0615-6) (2018).
5. Chang, C., Deng, Y., Jiang, X. & Long, Q. Multiple imputation for analysis of incomplete data in distributed health data networks. *Nat. Commun.* **11**, 5467, DOI: [10.1038/s41467-020-19270-2](https://doi.org/10.1038/s41467-020-19270-2) (2020).
6. van Buuren, S. *Flexible Imputation of Missing Data* (CRC Press, 2018), 2nd edn.
7. Roberts, M. *et al.* Common pitfalls and recommendations for using machine learning to detect and prognosticate for COVID-19 using chest radiographs and CT scans. *Nat. Mach. Intell.* **3**, 199–217, DOI: [10.1038/s42256-021-00307-0](https://doi.org/10.1038/s42256-021-00307-0) (2021).

8. Wynants, L. *et al.* Prediction models for diagnosis and prognosis of covid-19: systematic review and critical appraisal. *BMJ* **369**, m1328, DOI: [10.1136/bmj.m1328](https://doi.org/10.1136/bmj.m1328) (2020).
9. Li, J. *et al.* Predicting breast cancer 5-year survival using machine learning: A systematic review. *PLOS ONE* **16**, e0250370, DOI: [10.1371/journal.pone.0250370](https://doi.org/10.1371/journal.pone.0250370) (2021).
10. SCORE2 working group and ESC Cardiovascular risk collaboration. SCORE2 risk prediction algorithms: new models to estimate 10-year risk of cardiovascular disease in Europe. *Eur. Hear. J.* **42**, 2439–2454, DOI: [10.1093/eurheartj/ehab309](https://doi.org/10.1093/eurheartj/ehab309) (2021).
11. Deng, Y., Chang, C., Ido, M. S. & Long, Q. Multiple Imputation for General Missing Data Patterns in the Presence of High-dimensional Data. *Sci. Reports* **6**, 21689, DOI: [10.1038/srep21689](https://doi.org/10.1038/srep21689) (2016).
12. Schmitt, P., Mandel, J. & Guedj, M. A Comparison of Six Methods for Missing Data Imputation. *J. Biom. & Biostat.* **06**, DOI: [10.4172/2155-6180.1000224](https://doi.org/10.4172/2155-6180.1000224) (2015).
13. Muzellec, B., Josse, J., Boyer, C. & Cuturi, M. Missing Data Imputation using Optimal Transport. In *Proceedings of the 37th International Conference on Machine Learning*, 7130–7140 (PMLR, 2020).
14. Lin, W.-C. & Tsai, C.-F. Missing value imputation: a review and analysis of the literature (2006–2017). *Artif. Intell. Rev.* **53**, 1487–1509, DOI: [10.1007/s10462-019-09709-4](https://doi.org/10.1007/s10462-019-09709-4) (2020).
15. Platias, C. & Petasis, G. A Comparison of Machine Learning Methods for Data Imputation. In *11th Hellenic Conference on Artificial Intelligence*, SETN 2020, 150–159, DOI: [10.1145/3411408.3411465](https://doi.org/10.1145/3411408.3411465) (2020).
16. Armina, R., Zain, A. M., Ali, N. A. & Sallehuddin, R. A Review On Missing Value Estimation Using Imputation Algorithm. *J. Physics: Conf. Ser.* **892**, 012004, DOI: [10.1088/1742-6596/892/1/012004](https://doi.org/10.1088/1742-6596/892/1/012004) (2017).
17. Thurow, M., Dumpert, F., Ramosaj, B. & Pauly, M. Goodness (of fit) of Imputation Accuracy: The GoodImpact Analysis. *arXiv* DOI: [10.48550/arXiv.2101.07532](https://doi.org/10.48550/arXiv.2101.07532) (2021).
18. Jäger, S., Allhorn, A. & Bießmann, F. A Benchmark for Data Imputation Methods. *Front. Big Data* **4**, DOI: [10.3389/fdata.2021.693674](https://doi.org/10.3389/fdata.2021.693674) (2021).
19. Kantorovich, L. V. Mathematical Methods of Organizing and Planning Production. *Manag. Sci.* **6**, 366–422, DOI: [10.1287/mnsc.6.4.366](https://doi.org/10.1287/mnsc.6.4.366) (1960).
20. Goodfellow, I., Bengio, Y. & Courville, A. *Deep Learning* (MIT Press, 2016).
21. Pedregosa, F. *et al.* Scikit-learn: Machine Learning in Python. *J. Mach. Learn. Res.* **12**, 2825–2830 (2011).
22. Johnson, A. E. W. *et al.* MIMIC-III, a freely accessible critical care database. *Sci. Data* **3**, 160035, DOI: [10.1038/sdata.2016.35](https://doi.org/10.1038/sdata.2016.35) (2016).
23. Razavi, P. *et al.* The Genomic Landscape of Endocrine-Resistant Advanced Breast Cancers. *Cancer Cell* **34**, 427–438.e6, DOI: [10.1016/j.ccell.2018.08.008](https://doi.org/10.1016/j.ccell.2018.08.008) (2018).
24. Cushnan, D. *et al.* Towards nationally curated data archives for clinical radiology image analysis at scale: Learnings from national data collection in response to a pandemic. *DIGITAL HEALTH* **7**, 20552076211048654, DOI: [10.1177/20552076211048654](https://doi.org/10.1177/20552076211048654) (2021).
25. Little, R. J. A. & Rubin, D. B. *Statistical Analysis with Missing Data*. Wiley Series in Probability and Statistics (Wiley, 2019), 3rd edn.
26. Rubin, D. B. An Overview of Multiple Imputation. In *Proceedings of the Survey Research Methods Section, American Statistical Association*, 79–84 (1988).
27. Rubin, D. B. *Multiple Imputation for Nonresponse in Surveys*. Wiley Series in Probability and Statistics (John Wiley & Sons, 1987).
28. van Buuren, S. & Oudshoorn, K. Flexible Multivariate Imputation by MICE. Tech. Rep. PG/VGZ/99.054, Netherlands Organization for Applied Scientific Research (TNO), Leiden, The Netherlands (1999).
29. van Buuren, S. & Groothuis-Oudshoorn, K. mice: Multivariate Imputation by Chained Equations in R. *J. Stat. Softw.* **45**, 1–67, DOI: [10.18637/jss.v045.i03](https://doi.org/10.18637/jss.v045.i03) (2011).
30. Stekhoven, D. J. & Bühlmann, P. MissForest—non-parametric missing value imputation for mixed-type data. *Bioinformatics* **28**, 112–118, DOI: [10.1093/bioinformatics/btr597](https://doi.org/10.1093/bioinformatics/btr597) (2012).
31. Yoon, J., Jordon, J. & van der Schaar, M. GAIN: Missing Data Imputation using Generative Adversarial Nets. In *Proceedings of the 35th International Conference on Machine Learning*, vol. 80 of *Proceedings of Machine Learning Research*, 5689–5698 (PMLR, 2018).

32. Mattei, P.-A. & Frellsen, J. MIWAE: Deep Generative Modelling and Imputation of Incomplete Data Sets. In *Proceedings of the 36th International Conference on Machine Learning*, vol. 97 of *Proceedings of Machine Learning Research*, 4413–4423 (PMLR, 2019).
33. Jadhav, A., Pramod, D. & Ramanathan, K. Comparison of Performance of Data Imputation Methods for Numeric Dataset. *Appl. Artif. Intell.* **33**, 913–933, DOI: [10.1080/08839514.2019.1637138](https://doi.org/10.1080/08839514.2019.1637138) (2019).
34. Rabin, J., Peyré, G., Delon, J. & Bernot, M. Wasserstein barycenter and its application to texture mixing. In *International Conference on Scale Space and Variational Methods in Computer Vision*, 435–446 (Springer, 2011).
35. Bonneel, N., Rabin, J., Peyré, G. & Pfister, H. Sliced and radon wasserstein barycenters of measures. *J. Math. Imaging Vis.* **51**, 22–45 (2015).
36. Lundberg, S. M. & Lee, S.-I. A Unified Approach to Interpreting Model Predictions. In *Advances in Neural Information Processing Systems*, vol. 30 (Curran Associates, Inc., 2017).
37. Viñas, R., Azevedo, T., Gamazon, E. R. & Liò, P. Deep Learning Enables Fast and Accurate Imputation of Gene Expression. *Front. Genet.* **12**, DOI: [10.3389/fgene.2021.624128](https://doi.org/10.3389/fgene.2021.624128) (2021).
38. Bishop, C. M. Training with Noise is Equivalent to Tikhonov Regularization. *Neural Comput.* **7**, 108–116, DOI: [10.1162/neco.1995.7.1.108](https://doi.org/10.1162/neco.1995.7.1.108) (1995). Conference Name: Neural Computation.
39. Wang, S. *et al.* MIMIC-Extract: A Data Extraction, Preprocessing, and Representation Pipeline for MIMIC-III. *Proc. ACM Conf. on Heal. Inference, Learn.* 222–235, DOI: [10.1145/3368555.3384469](https://doi.org/10.1145/3368555.3384469) (2020).
40. Breiman, L. Random Forests. *Mach. Learn.* **45**, 5–32, DOI: [10.1023/A:1010933404324](https://doi.org/10.1023/A:1010933404324) (2001).
41. Choudhury, A. & Kosorok, M. R. Missing Data Imputation for Classification Problems. *arXiv* DOI: [10.48550/arXiv.2002.10709](https://doi.org/10.48550/arXiv.2002.10709) (2020).
42. Bhattarai, A. missingpy (2018). <https://github.com/epsilon-machine/missingpy> version 0.2.0.
43. van Buuren, S. Multiple imputation of discrete and continuous data by fully conditional specification. *Stat. Methods Med. Res.* **16**, 219–242, DOI: [10.1177/0962280206074463](https://doi.org/10.1177/0962280206074463) (2007).
44. Raghunathan, T. E., Lepkowski, J. M., Van Hoewyk, J. & Solenberger, P. A Multivariate Technique for Multiply Imputing Missing Values Using a Sequence of Regression Models. *Surv. Methodol.* **27**, 85–95 (2001).
45. Enders, C. K., Mistler, S. A. & Keller, B. T. Multilevel multiple imputation: A review and evaluation of joint modeling and chained equations imputation. *Psychol. Methods* **21**, 222–240, DOI: [10.1037/met0000063](https://doi.org/10.1037/met0000063) (2016).
46. Ding, Y. & Ross, A. A comparison of imputation methods for handling missing scores in biometric fusion. *Pattern Recognit.* **45**, 919–933, DOI: [10.1016/j.patcog.2011.08.002](https://doi.org/10.1016/j.patcog.2011.08.002) (2012).
47. White, I. R., Royston, P. & Wood, A. M. Multiple imputation using chained equations: Issues and guidance for practice. *Stat. Medicine* **30**, 377–399, DOI: [10.1002/sim.4067](https://doi.org/10.1002/sim.4067) (2011).
48. Tran, C. T., Zhang, M., Andreae, P., Xue, B. & Bui, L. T. An effective and efficient approach to classification with incomplete data. *Knowledge-Based Syst.* **154**, 1–16, DOI: [10.1016/j.knosys.2018.05.013](https://doi.org/10.1016/j.knosys.2018.05.013) (2018).
49. Goodfellow, I. *et al.* Generative Adversarial Nets. In *Advances in Neural Information Processing Systems*, vol. 27 (Curran Associates, Inc., 2014).
50. Wang, Z., Akande, O., Poulos, J. & Li, F. Are deep learning models superior for missing data imputation in large surveys? Evidence from an empirical comparison. *arXiv* DOI: [10.48550/arXiv.2103.09316](https://doi.org/10.48550/arXiv.2103.09316) (2022).
51. Kingma, D. P. & Welling, M. Auto-Encoding Variational Bayes. In *2nd International Conference on Learning Representations, ICLR 2014*, DOI: [10.48550/arXiv.1312.6114](https://doi.org/10.48550/arXiv.1312.6114) (Banff, AB, Canada, 2014).
52. Rezende, D. J., Mohamed, S. & Wierstra, D. Stochastic Backpropagation and Approximate Inference in Deep Generative Models. In *Proceedings of the 31st International Conference on Machine Learning*, 1278–1286 (PMLR, 2014).
53. Burda, Y., Grosse, R. & Salakhutdinov, R. Importance Weighted Autoencoders. *arXiv* DOI: [10.48550/arXiv.1509.00519](https://doi.org/10.48550/arXiv.1509.00519) (2016).
54. Cox, D. R. The regression analysis of binary sequences. *J. Royal Stat. Soc. Ser. B (Methodological)* **20**, 215–232, DOI: [10.1111/j.2517-6161.1958.tb00292.x](https://doi.org/10.1111/j.2517-6161.1958.tb00292.x) (1958). *eprint*: <https://onlinelibrary.wiley.com/doi/pdf/10.1111/j.2517-6161.1958.tb00292.x>.
55. Breiman, L. Consistency for a simple model of random forests. Technical Report 670, Statistics Department, University of California at Berkeley (2004).

56. Chen, T. & Guestrin, C. XGBoost: A Scalable Tree Boosting System. In *Proceedings of the 22nd ACM SIGKDD International Conference on Knowledge Discovery and Data Mining*, KDD '16, 785–794, DOI: [10.1145/2939672.2939785](https://doi.org/10.1145/2939672.2939785) (Association for Computing Machinery, New York, NY, USA, 2016).
57. Duan, T. *et al.* NGBoost: natural gradient boosting for probabilistic prediction. In *Proceedings of the 37th International Conference on Machine Learning*, Proceedings of Machine Learning Research, 2690–2700 (PMLR, 2020).
58. Martens, J. New Insights and Perspectives on the Natural Gradient Method. *J. Mach. Learn. Res.* **21**, 1–76 (2020).
59. Hastie, T., Tibshirani, R. & Friedman, J. *The Elements of Statistical Learning: Data Mining, Inference, and Prediction*. Springer Series in Statistics (Springer, New York, NY, USA, 2009), 2nd edn.
60. Rumelhart, D. E., Hinton, G. E. & Williams, R. J. Learning representations by back-propagating errors. *Nature* **323**, 533–536, DOI: [10.1038/323533a0](https://doi.org/10.1038/323533a0) (1986).
61. Agarap, A. F. Deep Learning using Rectified Linear Units (ReLU). *arXiv* DOI: [10.48550/arXiv.1803.08375](https://doi.org/10.48550/arXiv.1803.08375) (2019).
62. Kingma, D. P. & Ba, J. Adam: A Method for Stochastic Optimization. In *3rd International Conference on Learning Representations (ICLR)* (San Diego, CA, USA, 2015).
63. Villani, C. *Optimal Transport: Old and New*. Grundlehren der mathematischen Wissenschaften (Springer Berlin, Heidelberg, 2009), 1st edn.
64. Arjovsky, M., Chintala, S. & Bottou, L. Wasserstein Generative Adversarial Networks. In *Proceedings of the 34th International Conference on Machine Learning*, 214–223 (PMLR, 2017).

Supplementary Materials

Supplementary Information

Comparison between MSE and Wasserstein distance as discrepancy scores

In Figure 11 below, we show an example data distribution, created with the `make_classification` function, similar to the **Simulated** dataset. We evaluate the MSE and Wasserstein Distance between true and imputed data for (b) a method which optimises for MSE and (c) using MICE. It is clear that optimising for MSE leads to a poor distribution reconstruction, however MICE reflects the underlying distribution well. This simple example demonstrates how MSE is insufficient for assessing imputation quality, as although MICE is clearly preferable in this case, it has a poor MSE.

(a) True Data Distribution	(b) Imputation optimising mean square error	(c) Imputation with the MICE method
			Mean Square Error = 0.9074 Wasserstein Distance = 2.68×10^{-4}	Mean Square Error = 1.6533 Wasserstein Distance = 1.55×10^{-4}

Figure 11. Comparing imputation for MSE and MICE. In (a) we see a simulated dataset, (b) shows the optimal imputation against the mean square error (MSE) and (c) shows the MICE imputation result. The MSE and Wasserstein distances are quoted for each.

Dataset Description and Preprocessing Details

All of the datasets used in this study are publicly available (upon reasonable request for **MIMIC-III** and **NHSX COVID-19**). For the **MIMIC-III** and **Breast Cancer** datasets, we include some details below for how the **MIMIC-III** and **Breast Cancer** datasets were preprocessed for use in this study. The code for performing this preprocessing is also available in the codebase.

MIMIC-III. In preprocessing the MIMIC-III²² dataset, we only considered information for the first ICU admission of patients, and restricted to patients who are over 15 years old, spent at least 3 days in the ICU in the data collection period and were admitted as ‘Emergency’ or ‘Urgent’ cases. Furthermore, we extracted data on seven clinical variables: blood pressure (systolic, diastolic and mean), heart rate, oxygen saturation, respiratory rate and temperature; these were the only variables recorded for over 50% of the patients. In our preprocessing of the dataset, we only considered data from the first 10 days in the ICU (or the whole ICU stay if shorter), and excluded any patients who had fewer than 5 observations in any of aforementioned variables. We then calculated the mean and standard deviation for each of these seven variables, giving a total of 14 numerical variables **per patient**. The outcome variable we use is the survival of patients in the 30 days after admission to the ICU. Our preprocessing code is based on MIMIC-Extract³⁹ and is available at **[shared upon publication]**. **Due to the size of the dataset (and the number of computations we ultimately perform), we then select one third of the patients randomly, resulting in a dataset with 7214 patients.**

Breast Cancer. This dataset is derived from an oncology dataset collected at Memorial Sloan Kettering Cancer Center between April 2014 and March 2017²³. The biopsy samples were collected prior or during or after treatment from different primary or metastatic sites which leads to several samples for **the** each patient. We only considered the first occurrence of the patients ID. The dataset, consisting of 16 different features, is assembled using data stored in three different sub-datasets: `data_clinical_sample`, `data_clinical_patient`, and `breast_msk_2018_clinical_data` from Razavi et al.²³. Specifically, the features we consider are the ‘Fraction Genome Altered’ and ‘Mutation Count’, taken from

breast_msk_2018_clinical_data, 'ER Status of the Primary', 'Invasive Carcinoma Diagnosis Age', 'Oncotree Code', 'PR Status of the Primary', 'Overall Primary Tumor Grade' and 'Stage At Diagnosis', taken from `data_clinical_sample`, and 'Metastatic Disease at Last Follow-up', 'Metastatic Recurrence Time', 'M Stage', 'N Stage', 'T Stage', 'Overall Patient HER2 Status', 'Overall Patient HR Status' and 'Overall Patient Receptor Status' from `data_clinical_patient`. The features of this dataset are of several different types, 'ER Status of the Primary', 'Metastatic Disease at Last Follow-up', 'M Stage', 'Overall Patient HER2 Status', 'Overall Patient HR Status' and 'PR Status of the Primary' are binary; 'N Stage', 'T Stage', 'Stage At Diagnosis' and 'PR Status of the Primary' are ordinal; 'Oncotree Code', and 'Overall Patient Receptor Status' are multilevel categorical, and 'Fraction Genome Altered', 'Invasive Carcinoma Diagnosis Age', 'Metastatic Recurrence Time' and 'Mutation Count' are numerical. The outcome variable we choose was 'Overall Survival Status' for the classification task. Moreover, features 'N Stage' and 'T Stage' are described with 14 and 15 different levels respectively, and some of the levels only have **only** one or two samples. To ensure features are meaningfully represented, we only consider the parent family of each level which results in only 4 different levels for each feature.

All the ordinal variables are treated in the same manner as numeric variables if the imputation method was not capable of addressing the ordinal variables. The majority of categorical variables are binary, or have only two options (e.g. sex, death) and are simply encoded to zero and one and treated as numeric variables or binary (if the imputation method was capable of modelling the binary variables).

Descriptions of the imputation methods

Below, we give a detailed summary for each imputation method used in this paper, in particular, we detail how the categorical variables are encoded for each method.

Mean Imputation is one of the simplest imputation methods, in which the missing values are replaced with the sample mean. This is a single imputation method, as there is no stochasticity in its computation. Mean imputation was performed using the `SimpleImputer` implementation in the Python package `scikit-learn`²¹. Mean imputation has no special treatment for the multi-level categorical variables, therefore we one-hot encode all the multi-level categorical features as part of the preprocessing.

MissForest is an iterative imputation method which employs a random forest (a non-parametric model) for non-linear modeling of mixed data types. Therefore, MissForest is applicable for both numerical, categorical and ordinal data, whilst making few assumptions about the structure of the data^{30,40}. MissForest imputation works by initially training a random forest using the observed (i.e. not missing) data values to predict the missing entries of a given variable. This procedure continues until either a maximum number of iterations is reached or the out-of-bag error increases^{30,41}. MissForest imputation was performed using the Python package `missingpy`⁴². In our implementation, the multi-level categorical variables in **Breast Cancer** are specified to the algorithm.

Multivariate Imputation by Chained Equations (MICE), also known as 'Fully Conditional Specification'⁴³ or 'Sequential Regression Multivariate Imputation'⁴⁴, is an imputation method which iteratively imputes the missing values of each variable one at the time using a method based on the type of the corresponding variable (such as linear regression)^{28,41,45}. In its initialisation, MICE sets the missing values using values derived from the observed values, for example the mean of the feature or a random observed value. Next, a model is fit which takes one feature as the output/target variable and all other features are used as input variables. This model is then applied to predict the target variable and those which correspond to the missing values are replaced. This process is repeated, updating all features in turn. This gives a dataset in which all missing values of the features have been updated once. This whole process can be repeated many times until the imputed values converge to within some error⁴⁶⁻⁴⁸. We perform our computation using the R package `mice`²⁹. In our model, all the numerical variables are initially imputed using predictive mean matching (pmm)²⁹. In the **Breast Cancer** dataset, there are variables with multi-level categorical values. These variables are one-hot encoded and imputed using logistic regression along with binary variables. Finally, the ordinal variable are imputed using a proportional odds model.

Generative Adversarial Imputation Networks (GAIN)³¹ is an imputation method whose framework is based on generative adversarial networks⁴⁹. This approach adversarially trains a pair of neural networks, first a generator whose purpose is to generate realistic samples from the training distribution, and a discriminator which aims to identify whether a sample is from the training distribution. Training in this adversarial manner ensures that the generator creates more realistic samples as it tries to "fool" the discriminator. For the GAIN method, a binary mask is also provided to the generator function where the zeros correspond to the missing values and ones indicate an observed value. A "hint" matrix is generated from this mask for which some proportion of entries (decided by a parameter) are set to 0.5, i.e. for these entries we do not know if the value is observed or missing. Initially, all missing values are replaced by random noise sampled from a normal distribution. The

generated dataset is input to the discriminator along with the hint matrix. The task of discriminator is to predict the mask, i.e. identify both the observed values and imputed values^{31,50}. For GAIN, the official implementation provided in the paper³¹ is used. As GAIN cannot directly use multi-level categorical variables as input, in our implementation these are one-hot encoded.

Missing Data Importance-Weighted Autoencoder (MIWAE) is an imputation method proposed by Mattei and Frellsen³² which uses a deep latent variable model (DLVM)^{51,52} for the imputation of the missing values in a given dataset. This method builds upon the importance-weighted autoencoder (IWAE)⁵³, which aims to maximise a lower bound of the log-likelihood of the observed data. The lower bound of IWAE is a k -sample importance weighting estimate of the log-likelihood and is a generalization of the variational lower bound used in variational autoencoders⁵¹ (which corresponds to the case of $k = 1$). In MIWAE, the lower bound of IWAE is further generalized to the case of incomplete data (and coincides with the IWAE bound in the case of complete data). During training, the missing data is replaced with zeros before being passed into the encoder network to obtain the latent codes. The codes are passed to the decoder network and the output is compared with the observed data (only in non-missing dimensions) to compute the aforementioned lower bound. Once the model is trained, multiple imputation is possible via *sampling importance resampling* (SIR) from the trained model. We used the official Python implementation³² in our study. All the multi-level categorical variables are first one-hot encoded and the ordinal features are coded with numerical values before imputation with MIWAE.

Descriptions of the classification methods

In the following, we briefly describe each of the classification methods used in this paper. Moreover, we detail the hyperparameters, including the ranges of values, that are tuned in the benchmarking exercise for obtaining the optimal performing classifier.

Logistic Regression. This is simple and efficient classification method with high prediction performance for datasets that have linearly separable classes. This method use a logistic function $\frac{1}{1+e^{-(mx+b)}}$ to generate a binary output, where x is the input and m and b are learned⁵⁴. We used the `scikit-learn` library implementation of the logistic regression classifier. The only hyperparameter to tune is the maximum number of iterations used, we search over $\{50, 100, 150, 200, 250\}$.

Random Forest. This classifier is one of the most popular and successful algorithms. It was proposed by Breiman⁴⁰ and involves creating an ensemble of randomised decision trees whose predictions are aggregated to obtain the final results. In training, for each tree m samples are randomly selected by bootstrapping. Then, this subset of the dataset is used to train a randomized tree. This procedure is repeated k times. The Random Forest is the aggregation of these k decision trees⁵⁵. For our study, we used the Random Forest implementation in open source `scikit-learn` library²¹ in Python. The Random Forest can be applied to variety of different prediction problems, and few parameters need to be tuned. For the number of estimators, we tried 8 different values in the range $[20, 90]$ with equal step width as 10. For the maximum depth, we limited our search to $\{3, 4\}$. For minimum sample split and the minimum samples in each leaf, we search over the set $\{2, 3, 4\}$. This results in 144 different hyperparameter combinations.

XGBoost. This method, proposed by Chen and Guestrin⁵⁶, is an efficient gradient tree boosting algorithm which builds a chain of ‘weak learner’ trees to give a high-performing classification or regression model. In this method, a base model is constructed by training an initial tree. Then, the second tree is obtained through combination with initial tree. This procedure is repeated until the maximum number of trees is reached. These additive trees are selected based on a greedy algorithm, in each step adding the tree that most minimizes the loss function. Therefore, the training of the model is in additive manner⁵⁶. For the implementation of XGBoost, we used the official `xgboost` Python package provided in⁵⁶. For the maximum depth of trees we consider 3 different values $\{3, 4, 5\}$, and for the number of subsamples we selected among 6 different values in the range $[0.5, 1]$ with equal step size 0.1. The number of the trees are selected from 8 different values, in the range $[50, 400]$ with equal step size 50. This results in 144 different hyperparameter combinations.

NGBoost. This recent method, proposed by Duan et al.⁵⁷, gives probabilistic predictions for the outcome variable y for input x . It assumes there is an underlying probability distribution $P_{\theta}(y|x)$ with a parametric form, described by $\theta(x)$. This method considers natural gradient boosting, by replacing the loss function with a scoring rule. This scoring rule, compares the estimated probability distribution to the observed data, by using a predicted probability distribution P and the observed value y (outcome). The proper scoring rule S returns the best score for the true distribution of the outcomes. The parameter that minimize value of the score rule is obtained through natural gradient descend⁵⁸. For the implementation of the NGBoost method, we used provided code on the official repository at <https://github.com/stanfordmlgroup/ngboost>. An NGBoost model has three key hyperparameters which can be tuned, namely the learning rate, the minibatch fraction and the number of estimators. For the learning rate we used 4 different values $\{0.0001, 0.001, 0.01, 0.1\}$. For the minibatch fraction we

used 6 different values in the range $[0.5, 1]$ with equal step size 0.1. The number of the estimator is selected from 8 different values in the range $[50, 400]$ with equal step size 50. This results in 192 different hyperparameter combinations.

(Artificial) Neural Network (NN). The artificial neural network we employ is a multi-layer perceptron (MLP)⁵⁹ consisting of layers of neurons. Each neuron is assigned a value, based on a weighted combination the values for neurons in the previous layer. Activation functions are employed to introduce non-linearities into the weighted sum calculations. The training of the NN is through backpropagation⁶⁰, where the weights of the edges between each neuron are adjusted to minimize the error between the true labels and output. We use the ReLu⁶¹ activation function and the ADAM⁶² optimiser method with binary cross-entropy as the loss function. We used the implementation in the open source `scikit-learn` library²¹ in Python. An NN has three key hyperparameters which must be tuned, specifically, the initial learning rate for the optimizer, the number of the hidden layers, and the number of the neurons in each hidden layer. The initial learning rate selected from 3 different values $\{0.001, 0.01, 0.1\}$, the number of the hidden layers is set to one of the 3 values $\{1, 2, 3\}$. 5 different values are considered for the number of the neurons in each hidden layer from the range $[20, 100]$ with equal step size 20. This results in 45 different hyperparameters combinations.

Data partitioning and hyperparameter selection

We partition each dataset at two levels as shown in Figure 12. In the first level, we randomly partition the dataset into three holdout sets, each consisting of one third of the samples. These are used for reporting the performance of each imputation method and classifier. Each holdout set has a complementary development set and at the second level of partitioning, we divide the development set into five non-overlapping cohorts. We use five-fold cross-validation on the development set to select the optimal hyperparameters for each combination of imputation method and classifier using the mean area under the receiver operating characteristic curve (AUC) over the five validation folds. In Table 1, we detail the hyperparameters combinations considered for each of the classifiers.

Figure 12. A schematic illustrating the hierarchical dataset split. Key: H = Holdout, D = Development, V = Validation.

Classifier (K)	Tuned Hyperparameters
Random Forest (144)	Number of trees (8), maximum depth of trees (2), minimum samples needed for splits in each individual leaf (3), minimum number of samples in each leaf (3).
XGBoost (144)	Number of the trees (8), maximum depth of each tree (3), sub-sample of the training instances used in each iteration of boosting (6).
NGBoost (192)	Number of estimators (8), learning rate (4), subsample of the training instances used in each iteration of boosting (6).
Artificial Neural Network (45)	Number of neurons per layer (5), number of hidden layers (3), learning rate (3).

Table 1. Each classifier is fit with the K hyperparameter combinations shown above. The key hyperparameters for each model were identified and the number in brackets indicates how many values were tested for each hyperparameter.

Description of the ANOVA analysis

In our multi-factor ANOVA analysis, for the **MIMIC-III** dataset we included the missingness rate of the development data and holdout data as separate factors to allow us to evaluate their individual effects. However, the **Breast Cancer** and **NHSX**

COVID-19 datasets are real datasets with their inherent missingness rates. Therefore, the missingness rate is introduced as a factor (i.e. categorical 25%, 50%, inherent) to the ANOVA model.

Description of the sample-wise and feature-wise discrepancy measures

In this paper we use nine statistics to measure the discrepancy between the imputed and true data. These fall into three classes; class A are sample-wise, class B are feature-wise and class C are derived from the sliced Wasserstein distances. For all methods, the data in the development and holdout sets are normalised to mean zero and unit standard deviation (SD) using the development set mean and SD.

Sample-wise metrics and their implementation. We briefly describe the sample-wise statistics used in the paper, along with giving details of the implementations used:

- A1. Root MSE (RMSE). This is simply the square root of the mean square error, which is the average of the squared discrepancy errors between imputed and original samples. In this paper, we use the function `mean_squared_error` implemented in `sklearn` and take the square root of it.
- A2. Mean absolute error (MAE). This is the average absolute difference between the imputed and the original values. In this paper, we use the function `mean_absolute_error` implemented in `sklearn`.
- A3. R^2 . This is the coefficient of determination, implemented in `sklearn` as `r2_score`, which measures the proportion of the variation in the imputed values that is predictable from the original values. This is expressed as a percentage. Note that this is not a metric.

Feature-wise metrics and their implementation. We briefly describe the distribution comparison measures used in the paper, along with giving details of the implementations used:

- B1. Kullback-Leibler (KL). The KL divergence measures how different two probability distributions are from one another. This is implemented in Python using the standard calculation shown in `kl.py` in our codebase.
- B2. Kolmogorov-Smirnov (KS). The KS test is used to assess whether two one-dimensional probability distributions differ. We use the function `ks_2samp` in the Python package `scipy.stats`.
- B3. Two-Wasserstein (2W). The 2W distance⁶³ also measures the distance between two probability distributions using optimal transport. We use the function `emd2_1d` in the POT package in Python.

Outlier Analysis Details

To isolate whether the large variances are due to stochasticity in the algorithms, we now go back and consider the original feature distributions, rather than the projected distributions. If an imputation algorithm occasionally imputes poorly in particular features, it will be identified here. For each holdout and validation set, we compute the Wasserstein distance between the imputed data and the true data for all features in the 10 repeats, i.e. for each feature we have 10 Wasserstein distances. We want to understand how often these distances are very large relative to how often they are small. In Figure 8, we show the proportion of the imputations that have Wasserstein distances above a threshold of 10^{-7} for each holdout set and validation set. The plots for other thresholds are in Figure 13. It can be seen that the Mean imputation method leads to feature imputations that always have relatively large distances from the true values. This is not surprising as it is the baseline model. It is surprising that GAIN is often imputing with high distance (80% at 10^{-7} and 40% at 10^{-6}), indicating some stochasticity in the imputation method which causes poor imputations for some computations and better imputation for others. MIWAE demonstrates the same stochasticity to a lesser extent, followed by MissForest. This highlights the importance of performing multiple imputation runs for models which have stochasticity integral to them. For GAIN and MIWAE, this is particularly true as deep neural networks will occasionally find local minima at their optimum and generative adversarial networks are liable to mode collapse⁶⁴.

Interpretability

First, for each classifier, we find the top ten configurations of validation set, holdout set and imputation methods that achieve the best performance. We then rank these configurations by the distance ratio induced by the imputation method and take the model with the smallest and largest induced distance ratio for the sliced Wasserstein distance. This gives us the two models which are high performing but which are trained using data of different imputation quality (see Tables 6–8). The features that are important for a model’s prediction can be found using many interpretability techniques. In this paper we employ Shapley values³⁶ implemented in the Python package `shap`.

Figure 13. The proportion of repeated imputations that give outlier Wasserstein distances, at the thresholds shown, for different imputation methods.

Figure 14. For each classifier, we give the absolute skew of the Shapley values for each feature for the two candidate models identified.

Additional performance metrics for the downstream classification task

In addition to the exploring the AUC values for downstream performance of classifiers, we also report the **Accuracy**, **Brier Score**, **Precision**, **Sensitivity** and **Specificity**. These results are presented in Figures 15–17. For the **Simulated** and **MIMIC-III** datasets, we show the performance for missingness rates of 25% and 50% in the development and test data.

Dependence of the accuracy

(a) Classifier dependence

(b) Imputation dependence

Figure 15. Dependence of Accuracy of downstream classification performance on classification and imputation methods. These plots show the **Accuracy** of the downstream performance on the classification (a) and imputation methods (b). The size of each marker indicates the standard deviation.

Dependence of the Brier score

(c) Classifier dependence

(d) Imputation dependence

Figure 15. Dependence of Brier Score of downstream classification performance on classification and imputation methods.

Dependence of the precision

(e) Classifier dependence

(f) Imputation dependence

Figure 15. Dependence of Precision of downstream classification performance on classification and imputation methods.

Dependence of the sensitivity

(a) Classifier dependence

(b) Imputation dependence

Figure 16. Dependence of Sensitivity of downstream classification performance on classification and imputation methods.

Dependence of the specificity

(a) Classifier dependence

(b) Imputation dependence

Figure 17. Dependence of Specificity of downstream classification performance on classification and imputation methods.

Supplementary Figures for the Sample-wise Discrepancy Statistics

A: Sample-wise discrepancy for the MIMIC-III dataset at different train and test missingness rates

Figure 18. The sample-wise statistics for the MIMIC-III dataset at the different train and test missingness rates considered.

A: Sample-wise discrepancy for the Simulated dataset at different train and test missingness rates

Figure 19. The sample-wise statistics for the **Simulated** dataset at the different train and test missingness rates considered.

Supplementary Figures for the Feature-wise Discrepancy Statistics

B: Feature-wise discrepancy for the MIMIC-III dataset at the respective train and test missingness rates of 25% and 50%.

Figure 20. Feature-wise 25% train missingness and 50% test missingness.

B: Feature-wise discrepancy for the MIMIC-III dataset at the respective train and test missingness rates of 50% and 25%.

Figure 21. Feature-wise 50% train missingness and 25% test missingness.

B: Feature-wise discrepancy for the MIMIC-III dataset at the respective train and test missingness rates of 50% and 50%.

Figure 22. Feature-wise 50% train missingness and 50% test missingness.

B: Feature-wise discrepancy for the Simulated dataset at the respective train and test missingness rates of 25% and 25%.

Figure 23. Feature-wise 25% train missingness and 25% test missingness.

B: Feature-wise discrepancy for the Simulated dataset at the respective train and test missingness rates of 25% and 50%.

Figure 24. Feature-wise 25% train missingness and 50% test missingness.

B: Feature-wise discrepancy for the Simulated dataset at the respective train and test missingness rates of 50% and 25%.

Figure 25. Feature-wise 50% train missingness and 25% test missingness.

B: Feature-wise discrepancy for the Simulated dataset at the respective train and test missingness rates of 50% and 50%.

Figure 26. Feature-wise 50% train missingness and 50% test missingness.

Supplementary Figures for the Sliced Wasserstein Discrepancy Statistics

C: Sliced Wasserstein discrepancy for the MIMIC-III dataset at different train and test missingness rates

Figure 27. The class C discrepancies for the sliced Wasserstein distances of the MIMIC-III data at the 25% missingness rate for the training data along with 25% and 50% for the test data. The original values and logarithms are shown for clarity.

C: Sliced Wasserstein discrepancy for the MIMIC-III dataset at different train and test missingness rates

Figure 28. The class C discrepancies for the sliced Wasserstein distances of the MIMIC-III data at the 50% missingness rate for the training data along with 25% and 50% for the test data. The original values and logarithms are shown for clarity.

C: Sliced Wasserstein discrepancy for the Simulated dataset at different train and test missingness set rates

Figure 29. The class C discrepancies for the sliced Wasserstein distances of the Simulated data at the 25% missingness rate for the training data along with 25% and 50% for the test data. The original values and logarithms are shown for clarity.

C: Sliced Wasserstein discrepancy for the Simulated dataset at different train and test missingness rates

Figure 30. The class C discrepancies for the sliced Wasserstein distances of the Simulated data at the 50% missingness rate for the training data along with 25% and 50% for the test data. The original values and logarithms are shown for clarity.

Link between quality and downstream classification performance

Figure 31. The different imputation discrepancy metrics from classes A, B and C are shown against the downstream AUC value for the classification task of the Simulated dataset. Trend lines are shown for 25% and 50% test missingness separately.

Supplementary Distance Ratio Figure

Figure 32. Ratio of the Wasserstein distance for the imputed data compared to the original data for the MIMIC-III dataset at different train and test missingness rates.

Figure 33. Ratio of the Wasserstein distance for the imputed data compared to the original data for the Simulated dataset at different train and test missingness rates.

ANOVA Results for the Breast Cancer, NHSX COVID-19 and MIMIC-III datasets

Variables	Estimate	Std. Error	t value	Pr(> t)	
classifier choice LogisticReg	1.0824	0.0188	57.50	<2e-16	***
classifier choice NeuralNetwork	0.9841	0.0184	53.52	<2e-16	***
classifier choice NGBoost	1.0046	0.0185	54.37	<2e-16	***
classifier choice RandomForest	1.0475	0.0187	56.12	<2e-16	***
classifier choice XGBoost	1.0419	0.0186	55.90	<2e-16	***

Table 2. Table of parameter coefficients for **Breast Cancer** dataset.

Variables	Estimate	Std. Error	t value	Pr(> t)	
imputation choice GAIN	1.1323	0.0340	33.33	<2e-16	***
imputation choice Mean	1.1689	0.0342	34.22	<2e-16	***
imputation choice MICE	1.1299	0.0340	33.27	<2e-16	***
imputation choice MissForest	1.0688	0.0337	31.74	<2e-16	***
imputation choice MIWAE	1.1764	0.0342	34.40	<2e-16	***
classifier choice Neural Network	-0.0959	0.0354	-2.71	0.0085	**
classifier choice NGBoost	0.0357	0.0360	0.99	0.3239	
classifier choice Random Forest	0.0305	0.0359	0.85	0.3989	
classifier choice XGBoost	0.0749	0.0361	2.07	0.0423	*

Table 3. Table of parameter coefficients for **NHSX COVID-19** dataset

Variables	Estimate	Std. Error	t value	Pr(> t)	
imputation choice GAIN	1.0530	0.0130	81.03	<2e-16	***
imputation choice Mean	1.0119	0.0129	78.21	<2e-16	***
imputation choice MICE	1.0721	0.0130	82.33	<2e-16	***
imputation choice MissForest	1.0467	0.0130	80.60	<2e-16	***
imputation choice MIWAE	1.0661	0.0130	81.92	<2e-16	***
classifier choice Neural Network	0.0237	0.0123	1.93	0.0548	.
classifier choice NGBoost	0.0963	0.0124	7.78	1.3e-13	***
classifier choice RandomForest	0.0610	0.0123	4.94	1.3e-06	***
classifier choice XGBoost	0.1321	0.0124	10.62	<2e-16	***
train percentage 0.5	-0.0175	0.0079	-2.23	0.0268	*
test percentage 0.5	-0.1531	0.0079	-19.47	<2e-16	***

Table 4. Table of parameter coefficients for **MIMIC-III** dataset.

ANOVA results for the Simulated dataset

Variables	Estimate	Std. Error	t value	Pr(> t)	
imputation choice GAIN	1.6777	0.0306	54.79	<2e-16	***
imputation choice Mean	1.6191	0.0302	53.61	<2e-16	***
imputation choice MICE	1.6253	0.0302	53.74	<2e-16	***
imputation choice MissForest	1.6552	0.0302	54.85	<2e-16	***
imputation choice MIWAE	1.7848	0.0313	57.01	<2e-16	***
classifier choice Neural Network	0.9962	0.0459	21.69	<2e-16	***
classifier choice NGBoost	-0.0190	0.0387	-0.49	0.6241	
classifier choice Random Forest	-0.0482	0.0385	-1.25	0.2112	
classifier choice XGBoost	0.2913	0.0404	7.22	6.31e-12	***
train percentage 0.5	-0.1057	0.0330	-3.21	0.0015	**
test percentage 0.5	-0.4593	0.0320	-14.34	<2e-16	***
imputation choice Mean:train percentage 0.5	0.0130	0.0269	0.48	0.6285	
imputation choice MICE:train percentage 0.5	-0.0301	0.0271	-1.11	0.2675	
imputation choice MissForest:train percentage 0.5	-0.0334	0.0266	-1.26	0.2105	
imputation choice MIWAE:train percentage 0.5	-0.0671	0.0276	-2.43	0.0157	*
imputation choice Mean:classifier choice Neural Network	-0.0534	0.0456	-1.17	0.2429	
imputation choice MICE:classifier choice Neural Network	-0.0842	0.0453	-1.86	0.0642	.
imputation choice MissForest:classifier choice Neural Network	-0.2190	0.0445	-4.92	1.58e-06	***
imputation choice MIWAE:classifier choice Neural Network	-0.0732	0.0464	-1.58	0.1159	
imputation choice Mean:classifier choice NGBoost	-0.0507	0.0410	-1.23	0.2182	
imputation choice MICE:classifier choice NGBoost	0.0408	0.0411	0.99	0.3226	
imputation choice MissForest:classifier choice NGBoost	-0.0666	0.0407	-1.64	0.1030	
imputation choice MIWAE:classifier choice NGBoost	-0.0090	0.0419	-0.22	0.8298	
imputation choice Mean:classifier choice RandomForest	-0.1012	0.0408	-2.48	0.0137	*
imputation choice MICE:classifier choice RandomForest	0.0109	0.0409	0.27	0.7907	
imputation choice MissForest:classifier choice RandomForest	-0.1162	0.0405	-2.87	0.0044	**
imputation choice MIWAE:classifier choice RandomForest	-0.0416	0.0417	-1.00	0.3196	
imputation choice Mean:classifier choice XGBoost	-0.1364	0.0420	-3.25	0.0013	**
imputation choice MICE:classifier choice XGBoost	0.0125	0.0423	0.29	0.7684	
imputation choice MissForest:classifier choice XGBoost	-0.1325	0.0417	-3.18	0.0017	**
imputation choice MIWAE:classifier choice XGBoost	-0.0124	0.0432	-0.29	0.7743	
classifier choice NeuralNetwork:train percentage 0.5	-0.4091	0.0466	-8.79	2.40e-16	***
classifier choice NGBoost:train percentage 0.5	-0.0428	0.0394	-1.09	0.2786	
classifier choice RandomForest:train percentage 0.5	-0.0728	0.0389	-1.87	0.0626	.
classifier choice XGBoost:train percentage 0.5	-0.1845	0.0409	-4.52	9.70e-06	***
imputation choice Mean:test percentage 0.5	0.0415	0.0272	1.52	0.1286	
imputation choice MICE:test percentage 0.5	0.0186	0.0274	0.68	0.4966	
imputation choice MissForest:test percentage 0.5	-0.0984	0.0269	-3.65	0.0003	***
imputation choice MIWAE:test percentage 0.5	0.0305	0.0279	1.09	0.2753	
classifier choice Neural Network:test percentage 0.5	-0.5158	0.0442	-11.67	<2e-16	***
classifier choice NGBoost:test percentage 0.5	-0.0235	0.0378	-0.62	0.5335	
classifier choice RandomForest:test percentage 0.5	0.0170	0.0375	0.45	0.6513	
classifier choice XGBoost:test percentage 0.5	-0.1445	0.0392	-3.69	0.0003	***
train percentage 0.5:test percentage 0.5	0.0569	0.0374	1.52	0.1292	
classifier choice Neural Network:train percentage 0.5:test percentage 0.5	0.1876	0.0591	3.17	0.0017	**
classifier choice NGBoost:train percentage 0.5:test percentage 0.5	0.0169	0.0524	0.32	0.7468	
classifier choice RandomForest:train percentage 0.5:test percentage 0.5	0.0260	0.0520	0.50	0.6172	
classifier choice XGBoost:train percentage 0.5:test percentage 0.5	0.0778	0.0539	1.44	0.1501	

Table 5. Table of parameter coefficients for Simulated dataset.

Configurations of the models used in the interpretability analysis

Classifier	Imputation	Train Missing-ness	Test Missing-ness	Holdout Set	Distance Ratio	Mean Test AUC	Max Depth	Min Samples Split	Min Samples Leaf	Estimators
Random Forest	MICE	0.25	0.25	1	1.16	0.83	4	2	4	60
Random Forest	Mean	0.25	0.25	0	1.68	0.83	4	2	3	90

Table 6. The model configurations used in the interpretability analysis for the Random Forest.

Classifier	Imputation	Train Missing-ness	Test Missing-ness	Holdout Set	Distance Ratio	Mean Test AUC	Max Depth	Estimators	Subsampling
XGBoost	MICE	0.25	0.25	2	1.11	0.87	5	400	0.6
XGBoost	GAIN	0.25	0.25	0	1.51	0.88	5	400	0.6

Table 7. The model configurations used in the interpretability analysis for XGBoost.

Classifier	Imputation	Train Missing-ness	Test Missing-ness	Holdout Set	Distance Ratio	Mean Test AUC	Estimators	Learning Rate	Minibatch Fraction
NGBoost	MICE	0.25	0.25	1	1.16	0.84	400	0.01	0.5
NGBoost	GAIN	0.25	0.25	0	1.51	0.85	350	0.01	0.5

Table 8. The model configurations used in the interpretability analysis for NGBoost.

Correlation between discrepancy statistics for the Simulated dataset

Figure 34. Correlation heatmap for all discrepancy metrics considered in this paper for the **Simulated** dataset.

Codebase flowchart

Figure 35. Flowchart of the codebase. In yellow are shown the individual modules with the associated codes labelled. In pink, we indicate the Figures generated at each step by these codes.

Title: Classification of datasets with imputed missing values: does imputation quality matter?

Responses to the reviewers

Reviewer #1

The paper gives a summary of findings from an extensive comparison of several imputation methods for handling missing data using four datasets. Such a comparison study is useful for practitioners as it will give a good guidance to their daily practice.

We thank the reviewer for their kind feedback on our manuscript.

While the paper is well-written, I have a few suggestions to improve the quality of the paper. 1. In addition to Little and Rubin (2019), Kim and Shao (2021) should also be cited as a reference book for handling missing data and imputation.

We thank the reviewer for their comment and agree. The book "Statistical methods for handling incomplete data" by Kim and Shao (2021) has now been added to our references. You can find the changes for this comment in the **Imputation Methods** section in our revised version of the manuscript.

2. The authors consider five imputation methods, but MICE is the only statistical approach. Please consider fractional hot deck imputation (FHDI) or P-FHDI (Yang et al., 2022) also in the comparison.

We appreciate the reviewer's valuable comment, we have considered carefully whether to include any more types of imputation methods. However, the aim of our study is not a comprehensive and systematic review of all imputation methods, rather we want to focus attention on the importance of assessing imputation quality – rather than just focusing on downstream classifier performance. We have added this as a limitation of the study at the end of the discussion.

3. The main finding of the paper is that the MICE consistently outperforms the ML-based imputation method for classification. This finding is also reported by Wang et al. (2022). The authors may want to cite this paper.

We appreciate the reviewer drawing our attention to this paper which we have now cited. In our paper we find both MICE and MIWAE performing extremely well, and quite similarly to one another, for classification performance. However, MICE really performs extremely well by discrepancy scoring (as found in the Wang 2022 paper).

Reviewer #2

This work introduces a class of discrepancy statistics, based on the sliced Wasserstein distance, which allows assessing how well features with missing values have been reconstructed in a dataset after performing missing data imputation. Even though there are a few typos and (parts of) sentences that could be improved, the paper is well-written and follows IMRaD format, which makes sense since it was submitted to Communications Medicine. It is however very inconvenient for readers to have to jump to the Supplementary Materials repeatedly and then go back to understand the paper – in some parts, the Supplementary Materials are literally Fundamental Materials, without which the big picture cannot be understood. The authors should, therefore, try to make the paper more self-contained.

We thank the reviewer for their kind comments on our paper. We acknowledge that many fundamental parts of the paper, such as detailed descriptions of the datasets, imputation methods, hyperparameter searches and the classifiers are kept to the Supplementary Materials. However, within the strict word limits, we needed to compromise to keep the high-level information in the main manuscript, whilst keeping in the details for reproducibility for the Supplementary Materials. However, we hope our changes have made the manuscript more self-contained.

The paper addresses an interesting and important topic and shows promising results. However, I would like to invite the authors to revise their manuscript to address specific concerns (before a final decision is reached). Therefore, I invite the authors to revise the paper's focus, which should be on the quality of the imputation rather than dividing the focus between the quality of the imputation and the performance of the downstream machine learning task. In addition to blurring the discussion of the former, the latter does not contribute anything to the assessment of imputation quality. Furthermore, the paper gives no good

reason why classification was selected as the machine learning task. Why not prediction?! Why not both?!

We thank the reviewer for their kind comments and highlight that we are very appreciative of their very thorough review of the manuscript. We did carefully consider whether the two focusses of the paper could be separated but we believe they cannot. Our aim is to highlight to the machine learning community the importance of assessing the imputation quality when fitting models to data, in our case focussing on classifier models. A key question is whether imputation quality actually matters and has an effect on model performance, whilst also isolating whether it compromises the interpretability of downstream models. We believe that by examining the whole pipeline from imputation through to the downstream consequences, the argument is more convincing. This also allows the article to appeal to both the imputation and general machine learning communities – it may be missed by the latter if we focussed only on imputation quality scoring.

We completely agree with the reviewer that other downstream tasks also deserve attention, in particular prediction models. We hope that our paper serves to draw the community to focus on imputation quality and we can build on this foundation to investigate other downstream tasks. We have explicitly drawn this as one of our conclusions.

Although the main contribution of the paper is a new discrepancy measure – Sliced Wasserstein Distance – that can be used to measure how well features with missing values have been reconstructed in a dataset after the imputation of missing values, no theoretical proof was provided that the Sliced Wasserstein Distance gives, on average, better results than the Sliced KL Distance and the Sliced KS Distance. In principle, such proof is unattainable. Thus, the authors decided, and well, to perform an empirical evaluation.

We thoroughly agree with this reviewer's comment.

However, the empirical evaluation presented in the paper raises several concerns, to name but a few: a) Few datasets were used, and all have binary target features/variables, which weakens the evaluation.

We thank the reviewer for this comment. This paper is motivated by the aim to draw the community's attention to the importance of imputation quality and the effects it may have on downstream models. In the clinical machine learning field, the dominant machine learning methods used in practice are classifier models and they are often fit to important binary outcomes such as death/survival, malignant/benign, etc. Whilst we agree that our outcomes are binary, and this is a limitation of the study, we do not claim or aim to evaluate every imputation method on every type of dataset. Rather we aim to be the first paper to consider this type of analysis, highlight the importance of the issues to the community and now we and others can build on this paper to focus on, among other things, datasets with different outcomes, datasets with majority categorical variables, multiple outcome prediction, non-classification problems. This has been added as a limitation of the study at the end of the discussion section.

b) From the "Dataset Description and Preprocessing Details" Section of Supplementary Materials/Information (see page 18/48), it seems that the data for the (seven) selected continuous/numerical variables – "blood pressure (systolic, diastolic and mean), heart rate, oxygen saturation, respiratory rate, and temperature" – of the MIMIC-III dataset has missing values. Then, in a preprocessing step, some patients were dropped because they "had fewer than 5 observations in any of aforementioned variables". If this is correct, it is incorrect to consider the resulting dataset the ground truth and then draw conclusions from that.

The reviewer makes a good point here. The MIMIC-III dataset contains the clinical records of intensive care patients collected over a period of several years. Unlike in an experimental context, there are no pre-specified data values that are "expected" to be collected, and for those which are standard (such as the clinical variables we have chosen to consider), there is not a specified or standard frequency with which these are recorded. So a patient who was discharged from intensive care after 3 days might only have 3 or 4 measurements of one of these variables recorded, or perhaps for some reason the staff did not record some of the measurements correctly so they were removed during the data cleaning step. Since the data we are using for our experiments is summary data (mean and standard deviation) rather than individual data values, we decided to exclude these patients as the standard deviation, in particular, might be quite extreme. It turns out that less than 1% of patients (only 169 out of 21 812) were excluded in this way, which will have negligible impact on the qualitative or quantitative nature of the dataset. These figures have been added to the Supplementary Materials in the description of the MIMIC-III dataset.

c) Taking as granted that the observation made in the previous point is correct, the only (literally) complete dataset is the synthetic dataset which limits, to one out of four datasets, the evaluation by comparing against the ground truth.

Please see the previous comment.

d) The dataset extracted from the MIMIC-III dataset and the synthetic dataset have only continuous/numerical variables which limit the spectrum of the evaluation (no categorical features).

We thank the reviewer for highlighting this. We have added a comment to the conclusion section to highlight that whilst we have considered categorical and continuous variables over the datasets, a limitation is that the outcomes are binary and that other datasets with other qualities must be considered for further analysis. This paper serves as a motivation to the community to focus on the importance of the imputation quality. This has been added as a limitation of the study at the end of the discussion section.

e) Although the datasets are not completely characterized (e.g., features distributions, outliers, skewness, and kurtosis), the authors decided to apply one solution fits all in terms of data normalization – “For all methods, the data in the development and holdout sets are normalized to mean zero and unit standard deviation (SD) using the development set mean and SD.”. Usually, data preprocessing transformations are applied based on the characteristics of a given dataset, and one should avoid applying blindly a data transformation because that will have an impact when using the (transformed) data. For instance, a data preprocessing transformation may have a considerable impact on the performance of third-party tools (e.g., MICE, GAIN, and so forth) or the downstream ML task. Therefore, applying a one solution fits all in terms of data normalization jeopardizes the quest of the paper – “does imputation quality matter?”.

We thank the reviewer for their insightful comment. We completely agree with the reviewer that any preprocessing step will have a huge impact on the downstream classification results and should be adopted carefully by studying the dataset. In this paper, we carefully considered when data should and should not be normalised. In particular, at imputation time, it is only in the case of GAIN that normalisation is required and employed before feeding the dataset to the network – we then reverse the normalisation for classification. We apply no normalisation before the classification task to boost performance, as we found all methods (in particular the Neural Network) converged very well without it. Moreover, we highlight that we did perform normalisation before calculating the 3 different classes of discrepancies for the MIMIC-III and Simulated datasets. We have updated the text in our revised manuscript to clarify these points.

f) In principle, amputation (i.e., the “Induced missingness”) was done in the “Pre-Processing of the Datasets” stage in the data pipeline shown in Figure 35. However, it is not clear if this data transformation was also applied to the target feature/variable of the simulated and the MIMIC-III datasets. If it was not, what was the rationale behind that decision? If it was, why confusion matrices were not computed to assess the performance of the imputation methods/tools?

Amputation was only performed on the “input” features, not on the target feature/variable. The rationale behind this is that the imputation step is almost always performed without access to the target variable, so amputating the target variable would have had no effect on imputation; for the classification step, the training is all done on labelled data (the methods are all “fully supervised”) and the evaluation (validation and testing data) is all performed without access to the target variable, so amputating the target would simply have reduced the amount of available data for the experiment. This also addresses the comment about confusion matrices.

g) If no detail was overlooked in the paper and the Supplementary Materials, it is not possible to know which versions of the imputation tools were used or when and from where those tools were downloaded. Even worse, one does not know which setup/configuration was used to run each of those tools. The authors did a considerable effort to find the best setup/configuration of the downstream classification tasks (by performing a grid search) but none for the imputation methods/tools. Therefore, one must assume that default configurations were used, but this has an impact on the performance of the imputation methods/tools. Yet, conclusions about the performance of those tools were drawn.

We thank the reviewer for their valuable comment. Regarding the comment on the version of the imputation tools, we would like to state that a requirement.txt file can be found in our GitHub repository which contains all the packages (and their version numbers) needed for running our experiment. Additionally, to ensure the paper is self-contained, we have added a section to our Supplementary Material to state the version of Python, RStudio, along with versions of Python packages and other important information for the imputation methods in that section.

Regarding the second section of the comment, the hyperparameter of the imputation method, we confirm that this is indeed the case. For GAIN and MIWAE, with configurable parameters, we used the default values after finding that varying

the hyperparameters led to nominal changes in the losses. Additionally, a joint optimisation of imputation and classifier hyperparameters was deemed unjustifiably computationally expensive without any noticeable improvement. We have added a section to the Supplementary Materials to explain and justify this.

These concerns lead to the conviction that the paper does not present a thorough empirical evaluation of the proposed novel discrepancy measure. Consequently, one should take with a pinch of salt any conclusion that was drawn from the presented results.

We thank the reviewer for detailing their concerns. We do not aim in this paper, to fully examine the performance of the proposed sliced Wasserstein discrepancy for all datatypes and imputation methods, but our ambition is to draw attention to the importance of imputation quality whilst showing that existing discrepancy scoring is not reflective of the underlying data distributions. We have added the limitation to the conclusion that we only focus on binary outcomes and four diverse datasets, but agree that this is not intended to be fully exhaustive.

The paper could also benefit from improving Section 2.7 – “Measuring the quality of imputation”. Although I did not fully understand part of the notation (related to I_p and J_p , see “Step 1” of the algorithm provided on page 5/48), Section 2.7 would benefit from having:

- Pseudocode representation of the algorithm
- A minimal but elucidative graphical example, using a toy dataset (for instance, 7 samples and 4 features), that eases the perception of each of the three steps.

We thank the reviewer for this comment and agree this would be helpful. We have added a step-by-step example to the Supplementary Materials and reference this in the main text.

Finally, more comments and suggestions are provided in a PDF file.

We thank the reviewer for their thorough and diligent review. We believe that all of the comments have given a much improved manuscript.

Reviewer #3

This is an interesting study that claims to answer two research questions: 1. How to select imputation methods and classifier configurations, to train models using incomplete data. 2. How to best evaluate the imputation methods. The authors claim to introduce a class of discrepancy scores inspired by the sliced Wasserstein distance for evaluating how well imputed data faithfully reconstructs the overall distribution of feature values, and they shared their code on a GitLab Repo. The authors compared the performance of five different imputation methods namely: mean imputation, multivariate imputation by chained equations (MICE), MissForest, generative adversarial imputation networks (GAIN), and the missing data importance-weighted autoencoder (MIWAE). They also compared five classification algorithms namely: logistic regression, Random Forest, XGBoost, NGBoost, and an artificial neural network.

Improvement recommendations: 1. The authors are referring to a specific cohort extracted from MIMIC III as described in supplementary material and not the whole dataset, specific reference citation need to be provided in the main manuscript as well (minor).

We thank the reviewer for the comment and have updated the detail in the Supplementary Materials. The MIMIC-III dataset is not publicly available without training, but to allow for reproducibility we have provided scripts in the GitLab and GitHub repositories which allows for processing of the MIMIC-III data and for anyone to obtain the same cohort that we used in this study. We believe this is a reasonable compromise to allow reproducibility, whilst avoiding issues with the data originator by not releasing their dataset directly.

2. As per the cited study the preprocessing step includes missing data imputation, I'd highly recommend the authors to clearly describe the process they followed for the MIMIC cohort definition, especially that the experiment design and conclusion are highly linked to the assumption that MIMIC III cohort does not have any original missing data (major).

Please see the previous response which addresses this point also.

3. An overall study conclusion is missing, the reported results demonstrate that the best-performing imputation method and classifier will vary based on the dataset and the prediction task, which is logically expected. I'd highly recommend the authors to summarize the key findings in one or two statements to clearly convey the study innovation.(major)

We thank the reviewer for highlighting this oversight and have now added a concluding paragraph at the end of the manuscript.

4. Additionally, the authors need to highlight the advantage of their study compared to a number of additional studies besides those cited by the authors such as [1] "Strategies for Handling Missing Data in Electronic Health Record Derived Data" by Brian J. Wells, et al, and [2] <https://www.frontiersin.org/articles/10.3389/fgene.2021.691274/full>.

We thank the reviewer for sharing these references. Paper [1] discusses MICE imputation and gives an overview of issues with using real-world EHR data with missingness, but gives no scoring for the imputation quality. For paper [2], this only considers RMSE as a discrepancy score. In the current manuscript we have shown the issues with sample-wise discrepancy metrics and the importance of considering imputation quality in relation to the downstream classification performance. We have discussed both of these references in the text.

Reviewers' comments:

Reviewer #2 (Remarks to the Author):

I am pleased to note that the authors considered the majority of the comments and suggestions made by the reviewers. The references to the suggested literature, the references to the suggested related works, the example for calculating sliced Wasserstein distances, and the mention of software versions and imputation tools are a few examples. As a result (and as the authors acknowledged), the manuscript has significantly improved. However, some other comments and suggestions were not considered in the gracious revision made by the authors upon the comments and suggestions provided, including but not limited to:

- The suggestion to move the last sentence of Section 2.3 to Section 2.1.
- The justification of the captions of Figures 1 and 3 not done.
- Although the authors considered the comments previously made to the representation of the information contained in Figure 4, the new layout needs to be improved because the charts at the top row – (a) and (b) – and not aligned with the ones of the bottom row – (c) and (d).
- Figure 5 à y-axis different scale ranges.
- Figure 6 à middle row à y-axis right chart has a slightly different scale range.
- Figure 7 à y-axis different scale ranges.
- The same is also observed in other charts of other figures where they are side by side to enable (a somehow misleading) comparison of results.
- In Section 3 – Results – the sentence/claim “For all classifiers, with a fixed train missingness rate, a change in the test missingness rate affects performance very significantly compared to changing the train missingness rate for a fixed test missingness rate.” remains without explanation. And this is a critical (and easy) explanation that would help to understand better the presented results.
- In Section 3 à Comparing imputation quality à B. Feature distribution metrics à The last comment – “What is the explanation for the Mean imputation method to narrow the differences to all other methods when measuring the Wasserstein distance?” – was not considered.
- In Section 3 à Comparing imputation quality à C. Sliced Wasserstein distribution metrics à The first comment – “Figure 7 shows more than just the sliced Wasserstein distances.” – was not considered.
 - o Taking as granted that no detail was overlooked, it is not “the sliced Wasserstein distances”, but rather “the sliced distances”. In fact, besides “the sliced Wasserstein distances”, there are also the sliced KL distances as well as the sliced KS distances. In each of these cases, the name of a distance comes from the metric used to compute that distance, no matter if sliced or not.
 - o This could imply several other modifications across the entire paper due to inappropriate usage of the term “sliced Wasserstein distance(s)”.
- In Section 3 à Impact of imputation quality on interpretability à The comment “I do not understand the criteria to select these classifiers and imputation methods. For instance, why not KNN?” was not considered.
 - o This should be explained because KNN classifiers (e.g., the one available in scikit-learn) are a natural choice and widely used when the ML task is a classification problem.
- The first paragraph of Section 4 – Discussion –, specially at the end of it, clearly states that this study does not allow to provide a conclusive answer to the quest of the study – “does imputation quality matter?”
 - o IMHO, this contradicts in some way a sentence that is now repeated a few times in the new version of the manuscript – “The interest of this paper is not to assess all imputation methods, but to draw attention to the importance of assessing imputation quality when fitting classification models to

incomplete data.”.

o The authors decided (and well) to perform an important study, but, besides the obvious answer to the quest “does imputation quality matter?” – Yes, it matters!

Pretty much as generic quality matters for any other scientific problem and/or challenge. –, the first/main conclusion of the study is: “It was found that there is no particular classifier which performs best that outperforms all others across all of the datasets and similarly, no particular imputation method leads to the best downstream classification performance. Even for the datasets with the same types of missingness, no optimal imputation or classification method emerges.”.

o Therefore, it becomes difficult to accept the underlying idea that gives the perception that MICE is better than the other imputation tools that were used in the study.

o Although the authors decided to take into consideration a comment which pinpointed that MICE is not free of caveats and, as a result, included the sentence “We note that MICE and MissForest are extremely computationally expensive for high-dimensional data⁴⁴ so may be infeasible in some circumstances.”, it needs to be stressed out the caveats mentioned in [44, 45] (in the previous version of the manuscript [37, 50]).

§ This is a verbatim copy from [45] ([50]): MICE is appealing in large-scale survey data because it is simple and flexible in imputing different types of variables. However, MICE has a key theoretical drawback that the specified conditional distributions may be incompatible, that is, they do not correspond to a joint distribution (Arnold & Press 1989; Gelman & Speed 1993; Li et al. 2012). Despite this drawback, MICE works remarkably well in real applications and numerous simulations have demonstrated it outperforms many theoretically sound JM-based methods; see van Buuren 2018 for case studies. However, MICE is also computationally intensive (White et al. 2011) and generally cannot be parallelized. Moreover, popular software packages for implementing MICE with GLMs, e.g. mice in R (van Buuren & Groothuis-Oudshoorn 2011), often crash in settings with high dimensional non-continuous variables, e.g., categorical variables with many categories (Akande et al. 2017).

o Also, from countless experiments with MICE, even after parallelizing it, I was able to observe that MICE produce out of range and sometimes inadmissible (imputed) values.

• The comment to the last sentence of the second paragraph of Section 4 – Discussion – was addressed as follows:

o The comment was: “IMHO, more evidence is needed through evaluation with more datasets that have (much) less and (much) more samples than the ones used in this study, with very different dimensions as well as characteristics. The results of such evaluation should be provided as summaries of the data as well as plots that show the distribution of each feature, outliers, correlations (e.g., features heatmaps), and so forth.”.

o The reply/revision is: “This study has been designed to highlight the importance of considering the quality of the imputation for datasets which are then used in downstream tasks, therefore there are several limitations to this study. Firstly, we only focus on classification tasks as these represent the majority of the problems encountered in machine learning research applied to clinical data, i.e. predicting death vs. survival or malignant vs. benign disease. It is our hope that by standardising the imputation methods, multiple imputation, classification and analysis pipelines for determining the imputation quality, future studies have all the tools necessary to fit classifiers to incomplete data that are not only achieving high performance, but are built on high quality imputed data. This highlighting the consequences for classification models of using poorly imputed data, it will motivate the community to also focus on this for other predictive models with a single or multiple outputs. Secondly, we do not aim to provide a fully exhaustive empirical analysis all imputation

methods and classifiers as the aim of this manuscript is to draw the reader’s attention to the

importance of measuring imputation quality before fitting models. A third limitation is that the assessment of imputation quality was performed only for the Synthetic and MIMIC-III datasets with continuous numerical variables but an extension to datasets with categorical variables is envisaged.”.

o Although I understand the statement(s) of the authors, the lack of exhaustive validation remains the major flaw of this work.

Finally, I also challenge the authors to proofread the new version of the manuscript due to the several typos and a few sentences that can be improved. For instance, at the end of Section 4 – Discussion – it is written: “Firstly, we only focus on classification tasks as these represent the majority of the problems encountered in machine learning research applied to clinical data, i.e. predicting death vs. survival or malignant vs. benign disease.”.

Reviewer #3 (Remarks to the Author):

The authors did a good job considering our previous recommendations. Thank you.

I am pleased to note that the authors considered the majority of the comments and suggestions made by the reviewers. The references to the suggested literature, the references to the suggested related works, the example for calculating sliced Wasserstein distances, and the mention of software versions and imputation tools are a few examples. As a result (and as the authors acknowledged), the manuscript has significantly improved. However, some other comments and suggestions were not considered in the gracious revision made by the authors upon the comments and suggestions provided, including but not limited to:

- The suggestion to move the last sentence of Section 2.3 to Section 2.1.
- The justification of the captions of Figures 1 and 3 not done.
- Although the authors considered the comments previously made to the representation of the information contained in Figure 4, the new layout needs to be improved because the charts at the top row – (a) and (b) – and not aligned with the ones of the bottom row – (c) and (d).
- Figure 5 → y-axis different scale ranges.
- Figure 6 → middle row → y-axis right chart has a slightly different scale range.
- Figure 7 → y-axis different scale ranges.
- The same is also observed in other charts of other figures where they are side by side to enable (a somehow misleading) comparison of results.
- In Section 3 – Results – the sentence/claim *“For all classifiers, with a fixed train missingness rate, a change in the test missingness rate affects performance very significantly compared to changing the train missingness rate for a fixed test missingness rate.”* remains without explanation. And this is a critical (and easy) explanation that would help to understand better the presented results.
- In Section 3 → Comparing imputation quality → *B. Feature distribution metrics* → The last comment – *“What is the explanation for the Mean imputation method to narrow the differences to all other methods when measuring the Wasserstein distance?”* – was not considered.
- In Section 3 → Comparing imputation quality → *C. Sliced Wasserstein distribution metrics* → The first comment – *“Figure 7 shows more than just the sliced Wasserstein distances.”* – was not considered.
 - Taking as granted that no detail was overlooked, it is **not** “the sliced Wasserstein distances”, but rather “the sliced distances”. In fact, besides “the sliced Wasserstein distances”, there are also *the sliced KL distances* as well as *the sliced KS distances*. In each of these cases, the name of a distance comes from the metric used to compute that distance, no matter if sliced or not.
 - This could imply several other modifications across the entire paper due to inappropriate usage of the term *“sliced Wasserstein distance(s)”*.
- In Section 3 → Impact of imputation quality on interpretability → The comment *“I do not understand the criteria to select these classifiers and imputation methods. For instance, why not KNN?”* was not considered.
 - This should be explained because KNN classifiers (e.g., the one available in scikit-learn) are a natural choice and widely used when the ML task is a classification problem.
- The first paragraph of Section 4 – Discussion –, specially at the end of it, clear states that this study does not allow to provide a conclusive answer to the quest of the study – ***“does imputation quality matter?”***
 - IMHO, this contradicts in somehow a sentence that is now repeated a few times in the new version of the manuscript – *“The interest of this paper is not to assess all imputation methods, but to draw attention to the importance of assessing imputation quality when fitting classification models to incomplete data.”*
 - The authors decided (and well) to perform an important study, but, besides the obvious answer to the quest ***“does imputation quality matter?”*** – Yes, it matters!

Pretty much as generic quality matters for any other scientific problem and/or challenge. –, the first/main conclusion of the study is: *“It was found that there is no particular classifier which performs best that outperforms all others across all of the datasets and similarly, no particular imputation method leads to the best downstream classification performance. Even for the datasets with the same types of missingness, no optimal imputation or classification method emerges.”*.

- Therefore, it becomes difficult to accept the underlying idea that gives the perception that MICE is better than the other imputation tools that were used in the study.
- Although the authors decided to take into consideration a comment which pinpointed that MICE is not free of caveats and, as a result, included the sentence *“We note that MICE and MissForest are extremely computationally expensive for high-dimensional data⁴⁴ so may be infeasible in some circumstances.”*, it needs to be stressed out the caveats mentioned in [44, 45] (in the previous version of the manuscript [37, 50]).
 - This is a verbatim copy from [45] ([50]): MICE is appealing in large-scale survey data because it is simple and flexible in imputing different types of variables. However, **MICE has a key theoretical drawback that the specified conditional distributions may be incompatible**, that is, they do not correspond to a joint distribution (Arnold & Press 1989; Gelman & Speed 1993; Li et al. 2012). Despite this drawback, MICE works remarkably well in real applications and numerous simulations have demonstrated it outperforms many the-oretically sound JM-based methods; see van Buuren 2018 for case studies. However, **MICE is also computationally intensive** (White et al. 2011) and **generally cannot be parallelized**. Moreover, **popular software packages for implementing MICE with GLMs**, e.g. mice in R (van Buuren & Groothuis-Oudshoorn 2011), **often crash in settings with high dimensional non-continuous variables**, e.g., categorical variables with many categories (Akande et al. 2017).
- Also, from countless experiments with MICE, even after parallelizing it, I was able to observe that MICE produce out of range and sometimes inadmissible (imputed) values.
- The comment to the last sentence of the second paragraph of Section 4 – Discussion – was addressed as follows:
 - The comment was: *“IMHO, more evidence is needed through evaluation with more datasets that have (much) less and (much) more samples than the ones used in this study, with very different dimensions as well as characteristics. The results of such evaluation should be provided as summaries of the data as well as plots that show the distribution of each feature, outliers, correlations (e.g., features heatmaps), and so forth.”*.
 - The reply/revision is: *“This study has been designed to highlight the importance of considering the quality of the imputation for datasets which are then used in downstream tasks, therefore there are several limitations to this study. Firstly, we only focus on classification tasks as these represent the majority of the problems encountered in machine learning research applied to clinical data, i.e. predicting death vs. survival or malignant vs. benign disease. It is our hope that by standardising the imputation methods, multiple imputation, classification and analysis pipelines for determining the imputation quality, future studies have all the tools necessary to fit classifiers to incomplete data that are not only achieving high performance, but are built on high quality imputed data. This highlighting the consequences for classification models of using poorly imputed data, it will motivate the community to also focus on this for other predictive models with a single or multiple outputs. Secondly, we do not aim to provide a fully exhaustive empirical analysis all imputation*

methods and classifiers as the aim of this manuscript is to draw the reader's attention to the importance of measuring imputation quality before fitting models. A third limitation is that the assessment of imputation quality was performed only for the Synthetic and MIMIC-III datasets with continuous numerical variables but an extension to datasets with categorical variables is envisaged."

- Although I understand the statement(s) of the authors, the lack of exhaustive validation remains the major flaw of this work.

Finally, I also challenge the authors to proofread the new version of the manuscript due to the several typos and a few sentences that can be improved. For instance, at the end of Section 4 – Discussion – it is written: *"Firstly, we only focus on **classification** tasks as these represent the majority of the problems encountered in machine learning research applied to clinical data, i.e. **predicting** death vs. survival or malignant vs. benign disease."*

Title: Classification of datasets with imputed missing values: does imputation quality matter?

Responses to the reviewers for the revised manuscript

Reviewer #2

I am pleased to note that the authors considered the majority of the comments and suggestions made by the reviewers. The references to the suggested literature, the references to the suggested related works, the example for calculating sliced Wasserstein distances, and the mention of software versions and imputation tools are a few examples. As a result (and as the authors acknowledged), the manuscript has significantly improved.

We sincerely thank the reviewer for their detailed review and appreciate the time they have spent giving such thorough and valuable feedback.

However, some other comments and suggestions were not considered in the gracious revision made by the authors upon the comments and suggestions provided, including but not limited to: • The suggestion to move the last sentence of Section 2.3 to Section 2.1.

We thank the reviewer for highlighting this and have now moved this sentence to the end of Section 2.1.

- The justification of the captions of Figures 1 and 3 not done.

We thank the reviewer for highlighting this and have now justified all captions in the manuscript.

- Although the authors considered the comments previously made to the representation of the information contained in Figure 4, the new layout needs to be improved because the charts at the top row – (a) and (b) – and not aligned with the ones of the bottom row – (c) and (d).

We thank the reviewer for highlighting this and we hope that the revised Figure is acceptable with equal scaling of the axes. Figure 5 - y-axis different scale ranges.

We thank the reviewer for this comment. In Figure 5, we are comparing three different discrepancy scores which each have different ranges. The important aspect of these three plots is to compare how each imputation method performs within each of the scores, rather than comparing between plots. We have highlighted in the caption that the scales are different in case a reader does try to compare between the three plots. If we were to adjust the axes of the plots, we would lose much of the information as the MAE and R2 plots would be compressed significantly. We hope this sufficiently addresses the reviewers concerns and would value any alternative perspective.

Figure 6 - middle row - y-axis right chart has a slightly different scale range.

We thank the reviewer for identifying this oversight. We have corrected the scale on the axes and re-reviewed all figures in the paper to ensure there is consistency whenever the same discrepancy scores are being compared.

Figure 7 - y-axis different scale ranges.

We thank the reviewer for this comment and believe we have addressed this with the Figure 5 comment above.

The same is also observed in other charts of other figures where they are side by side to enable (a somehow misleading) comparison of results.

We thank the reviewer for this comment. We have checked all Figures in the paper now where there are side-by-side comparisons and corrected the axes.

In Section 3 – Results – the sentence/claim “For all classifiers, with a fixed train missingness rate, a change in the test missingness rate affects performance very significantly compared to changing the train missingness rate for a fixed test missingness rate.” re-

mains without explanation. And this is a critical (and easy) explanation that would help to understand better the presented results.

We thank the reviewer for this comment but would appreciate further details on what they would like included in the manuscript. It is not necessarily easy to say why e.g. a model fit to training data with 50% missingness would have a significant performance drop when evaluated on the holdout data varying from 25% to 50% missingness.

In Section 3 “Comparing imputation quality” B. Feature distribution metrics - The last comment – “What is the explanation for the Mean imputation method to narrow the differences to all other methods when measuring the Wasserstein distance?” – was not considered.

We thank the reviewer for this comment. These boxplots summarise over the different validation sets and the repeated imputations. As mean imputation generates the same data for each repeated imputation, there is minimal variance expected in the distances we summarise over. Therefore the plots have very tight boxplots for mean, in comparison to the other imputation methods.

In Section 3 “Comparing imputation quality” C. Sliced Wasserstein distribution metrics. The first comment – “Figure 7 shows more than just the sliced Wasserstein distances.” – was not considered.

We thank the reviewer for their comment. I think maybe this needs some clarification from our side. We measure the sliced Wasserstein distance between random 1D projections and the imputed/original data for all random directions, repeat imputations and validation sets. This process generates two distributions one for the imputed distances and one for the original data distances. From these sliced Wasserstein distance distributions we then compute final discrepancy scores using KL, KS and Wasserstein distance. In the revised version, we update the Figures caption to incorporate some extra information, highlighting that three discrepancy scores were being used, and also updated it in the manuscript text explicitly.

“In Figure 7 and Supplementary Figures 27–30, we show the discrepancies between the distributions of the sliced Wasserstein distances, measured using the Kullback-Leibler divergence, the Kolmogorov-Smirnov statistic and the Wasserstein distance for the imputation methods at different train and test missingness rates for the **MIMIC-III** and **Simulated** datasets.”

It is useful terminology for us to refer to ‘sliced Wasserstein distances’ for the class C scores as these are the source values for the final distributions that we measure KL, KS, Wasserstein from. We hope this is sufficient for the reviewer.

Taking as granted that no detail was overlooked, it is not “the sliced Wasserstein distances”, but rather “the sliced distances”. In fact, besides “the sliced Wasserstein distances”, there are also the sliced KL distances as well as the sliced KS distances. In each of these cases, the name of a distance comes from the metric used to compute that distance, no matter if sliced or not. This could imply several other modifications across the entire paper due to inappropriate usage of the term “sliced Wasserstein distance(s)”.

We thank the reviewer for this. I hope we have clarified in the previous answer that the KL, KS and 2W distances have been derived from sliced Wasserstein distances and for simplicity we have referred to them throughout as the sliced Wasserstein distances.

In Section 3 “Impact of imputation quality on interpretability” The comment “I do not understand the criteria to select these classifiers and imputation methods. For instance, why not KNN?” was not considered. This should be explained because KNN classifiers (e.g., the one available in scikit-learn) are a natural choice and widely used when the ML task is a classification problem.

We thank the reviewer for this comment. We picked a variety of methods for this study, from simple ones to complex ones, to draw attention of the readers to focus that imputation quality is worth focussing on and show effects across methods. We don’t intend to be exhaustive over all imputation methods and have considered several simple and popular methods along with newer and complex ones. We highlight that the project was intensely expensive computationally and believe that the added value of adding one further imputation method may be minimal to the conclusions of the paper.

The first paragraph of Section 4 – Discussion –, specially at the end of it, clear states that this study does not allow to provide a conclusive answer to the quest of the study – “does imputation quality matter?” IMHO, this contradicts in somehow a

sentence that is now repeated a few times in the new version of the manuscript – “The interest of this paper is not to assess all imputation methods, but to draw attention to the importance of assessing imputation quality when fitting classification models to incomplete data.” The authors decided (and well) to perform an important study, but, besides the obvious answer to the quest “does imputation quality matter?” – Yes, it matters!

We thank the reviewer for this comment. At the start of the Discussion, we conclude that given a particular dataset, it is impossible to apriori select the optimal imputation and classification method which would deliver optimal performance for the classifier performance. We did not intend for this to mean “imputation quality is irrelevant”. We have incorporated the interpretability discussion into this paragraph to emphasise that whilst downstream classifier performance is not dependent on imputation quality, the model you will derive is compromised and therefore imputation quality is important in the modelling process.

Pretty much as generic quality matters for any other scientific problem and/or challenge. –, the first/main conclusion of the study is: “It was found that there is no particular classifier which performs best that outperforms all others across all of the datasets and similarly, no particular imputation method leads to the best downstream classification performance. Even for the datasets with the same types of missingness, no optimal imputation or classification method emerges.”. Therefore, it becomes difficult to accept the underlying idea that gives the perception that MICE is better than the other imputation tools that were used in the study.

We thank the reviewer for this comment. In the previous response we have addressed the nuance in this statement that we make in the paper, and have updated the leading paragraph of the Discussion to emphasise that imputation quality does matter and does feed through into the resulting model – in e.g. compromised interpretability. We conclude that MICE better reflects the underlying data distribution than other methods, along with being more stable to outliers in our experiments.

Although the authors decided to take into consideration a comment which pinpointed that MICE is not free of caveats and, as a result, included the sentence “We note that MICE and MissForest are extremely computationally expensive for high-dimensional data⁴⁴ so may be infeasible in some circumstances.”, it needs to be stressed out the caveats mentioned in [44, 45] (in the previous version of the manuscript [37, 50]). This is a verbatim copy from [45]([50]): MICE is appealing in large-scale survey data because it is simple and flexible in imputing different types of variables. However, MICE has a key theoretical drawback that the specified conditional distributions may be incompatible, that is, they do not correspond to a joint distribution (Arnold & Press 1989; Gelman & Speed 1993; Li et al. 2012). Despite this drawback, MICE works remarkably well in real applications and numerous simulations have demonstrated it outperforms many theoretically sound JM-based methods; see van Buuren 2018 for case studies. However, MICE is also computationally intensive (White et al. 2011) and generally cannot be parallelized. Moreover, popular software packages for implementing MICE with GLMs, e.g. mice in R (van Buuren & Groothuis-Oudshoorn 2011), often crash in settings with high dimensional non-continuous variables, e.g., categorical variables with many categories (Akande et al. 2017). Also, from countless experiments with MICE, even after parallelizing it, I was able to observe that MICE produce out of range and sometimes inadmissible (imputed) values.

We thank the reviewer for this comment. We read with interest the passage they repeated. The key outcomes are (1) MICE have theoretical drawbacks, (2) MICE is computationally expensive, (3) MICE can crash for high-dimensional non-continuous variables. We have highlighted (2) previously and now highlight (1) and (3) in the revised manuscript. We hope this addresses the concerns of the reviewer.

The comment to the last sentence of the second paragraph of Section 4 – Discussion – was addressed as follows: o The comment was: “IMHO, more evidence is needed through evaluation with more datasets that have (much) less and (much) more samples than the ones used in this study, with very different dimensions as well as characteristics. The results of such evaluation should be provided as summaries of the data as well as plots that show the distribution of each feature, outliers, correlations (e.g., features heatmaps), and so forth.”. o The reply/revision is: “This study has been designed to highlight the importance of considering the quality of the imputation for datasets which are then used in downstream tasks, therefore there are several limitations to this study. Firstly, we only focus on classification tasks as these represent the majority of the problems encountered in machine learning research applied to clinical data, i.e. predicting death vs. survival or malignant vs. benign disease. It is our hope that by standardising the imputation methods, multiple imputation, classification and analysis pipelines for determining the imputation quality, future studies have all the tools necessary to fit classifiers to incomplete data that are not only achieving high performance, but are built on high quality imputed data. This highlighting the consequences for classification models of using poorly imputed data, it will motivate the community to also focus on this for other predictive models with a single or multiple outputs. Secondly, we do not aim to provide a fully exhaustive empirical analysis all imputation methods and

classifiers as the aim of this manuscript is to draw the reader's attention to the importance of measuring imputation quality before fitting models. A third limitation is that the assessment of imputation quality was performed only for the Synthetic and MIMIC-III datasets with continuous numerical variables but an extension to datasets with categorical variables is envisaged." o Although I understand the statement(s) of the authors, the lack of exhaustive validation remains the major flaw of this work.

We appreciate the reviewer's point of view here. Our study aims not to be exhaustive and we have added the limitations identified by the reviewer. We have incorporated experiments for a **Synthetic (N,C)** dataset that contains numerical, categorical and ordinal variables where we observe consistent results to the prior version of the manuscript. In addition to the three real datasets and the synthetic numerical dataset, we believe we have established the additional validation requested.

Finally, I also challenge the authors to proofread the new version of the manuscript due to the several typos and a few sentences that can be improved. For instance, at the end of Section 4 – Discussion – it is written: "Firstly, we only focus on classification tasks as these represent the majority of the problems encountered in machine learning research applied to clinical data, i.e. predicting death vs. survival or malignant vs. benign disease."

We have carefully reviewed the article to improve typos and clarity and believe the revised manuscript is much improved. We hope that the reviewer agrees.

Reviewer #3

The authors did a good job considering our previous recommendations. Thank you.

We thank the reviewer for their careful consideration of our manuscript and the significant improvements they proposed.

REVIEWERS' COMMENTS:

Reviewer #2 (Remarks to the Author):

I am pleased to notice how much the paper has improved! Although I am providing here a few comments, from my point of view, these suggestions should not prevent the paper from moving to the next phase (i.e., getting accepted to be published).

Here are a few comments/suggestions:

In 2.7 Measuring the quality of imputation Section \diamond C. Sliced Wasserstein distance \diamond it is said: "Secondly, high-dimensional (complete) datasets are very sparse in the space R^d unless there are an unrealistically large number of samples." \diamond [comment] Is it only the sparsity that matters, or does the density also matter?

In 3 Results Section \diamond Comparing imputation quality \diamond B. Feature distribution metrics \diamond it is said: "MICE imputation, which performed worst by the sample-wise discrepancy scores, is the best-performing method by the Kolmogorov-Smirnoff statistic and Wasserstein distance for all missingness rates across all the minimum, median and maximum discrepancies." \diamond [comment] While this is true, when doing this sort of comparison, one should also consider the magnitude of the differences. Although MICE imputation performs best by the Kolmogorov-Smirnoff statistic and Wasserstein distance, the difference to MissForest is minimal when compared to the counterpart differences by the sample-wise discrepancy scores.

In 3 Results Section \diamond Comparing imputation quality \diamond C. Sliced Wasserstein distribution metrics. \diamond it is said: "For the MIMIC-III dataset, over all discrepancy scores and missingness rates, the MICE imputation method shows a clear dominance with the mean and GAIN methods performing poorly." \diamond [comment] Again, this is true! However, one should also put into perspective the magnitude of the differences in what is being measured using the different metrics.

Also, something that I am missing from the very beginning – I apologize if I am pointing out this only now – is the scale of what is being measured. In other words, it would be extremely helpful to know for each plot which is the minimum possible value (I would say, zero) and the maximum (which is not clear). Having these values in mind would, probably, give a different perspective on the achieved results. For instance, in Figure 7, it is obvious that MICE outperforms the other imputation methods, but the question is: is that result negligible? The same goes for other comparisons.

Then it is said: "For the Simulated (N) dataset, the MICE method again performs the best overall by all measures ..." \diamond [comment] This is not true, see first row of Figure 37.

Also, in the same sentence is said: "while for the Simulated (N,C) dataset, MICE and MIWAE are the best overall." \diamond [comment] I suggest changing the formulation and saying "in general". Furthermore, this is another example where, typically, the differences seem to be neglectable.

A little bit after it is said: "For MIMIC-III and Simulated (N), mean imputation performs the worst, with GAIN and MissForest performing similarly poorly whilst for Simulated (N,C) all these methods perform equally poorly." \diamond [comment] I would be more cautious with the adjectives since I do not observe differences that justify the adjective poorly, at least not for MissForest.

I also suggest the authors fix the following typos and/or sentences:

- At the end of 1 Introduction Section, after the GitHub URL \diamond [suggestion] A comma is missing, and

the rest of the sentence reads a bit weird.

- In 2.6 ANOVA analysis Section: "... named in §2.4. Firstly, ..." ◇ [suggestion] A space is missing.
- In 2.7 Measuring the quality of imputation Section:
 - o "... we must define a score that has the desirable property that it has a low value when the distribution of imputed values closely resembles that of the true values." ◇ [suggestion] "... we must define a desirable score's property that achieves a low value when the distribution of imputed values closely resembles that of the true values."
 - o "Explicitly, our aim is to compute a distance between the original samples $D = \{x_i\}_{N_i=1}$ and the imputed samples $D^{\wedge} = \{\hat{x}_i\}_{N_i=1}$ that will indicate the quality of the imputation." ◇ [suggestion] "... that gives the quality of the imputation."
 - o "The class of measures which would be of most practical value to practitioners is (C) measures of discrepancy for imputed and true data across the whole data distribution." ◇ [suggestion] "However, we strongly believe that the class of measures which would be of most practical value to practitioners ...".
 - o "B. Feature-wise distribution discrepancy. In the literature, some authors have considered discrepancy measures to quantify for how faithfully the distributions of individual features are reconstructed." ◇ [suggestion] Delete "for".
 - o "C. Sliced Wasserstein distance. In Figure 1, we see that simply considering ..." ◇ [suggestion] Replace "see" with "show".
 - o "Step1: ... We set $P = 10$ throughout" ◇ [suggestion] Delete "throughout" and end with a dot (i.e., a period), at least put the dot there.
- In 3 Results Section:
 - o Comparing imputation quality
 - ♣ A. Sample-wise statistics.
 - "In Figure 5 and Supplementary Figures 20–22 the sample-wise ..." ◇ [suggestion] A comma is missing after "Figures 20-22".
 - "Simulated (N)and Simulated(N,C)" ◇ [suggestion] A space is missing.
 - ♣ B. Feature distribution metrics.
 - "In Figure 6 and Supplementary Figures 23–33" ◇ [suggestion] Again, a comma is missing after "Figures 23-33".
 - "In Figure 6 and Supplementary Figures 23–33 we show the minimum, median and maximum feature-wise discrepancies statistics for the MIMIC-III, Simulated (N) and Simulated (N,C) datasets. The mean imputation method is the worst in all metrics for minimum, median and maximum at all rates of train and test missingness in both datasets." ◇ [suggestion] Not "both" but "all".
 - Also, "The mean imputation method is the worst in all metrics for minimum, median and maximum at all rates of train and test missingness ..." ◇ [suggestion] "In general, the mean imputation method..." because there is a few cases in which it is not the worst.
 - ♣ C. Sliced Wasserstein distribution metrics.
 - "At the 25% test missingness rate, the MIWAE imputation method is competitive with and sometimes outperforms, MICE." ◇ [suggestion] Delete the comma after "outperforms".
 - o In 4 Discussion Section:
 - o "In fact, for MSE in particular, ..." ◇ [suggestion] MAE.
 - o [suggestion] Please, change accordingly onwards that point.
 - o "Secondly, we do not aim to provide a fully exhaustive empirical analysis all imputation methods and classifiers as the aim of this manuscript is to draw the reader's attention to the importance of measuring imputation quality before fitting models." ◇ [suggestion] "... a fully exhaustive empirical

analysis of all imputation methods ...”.

I am pleased to notice how much the paper has improved! Although I am providing here a few comments, from my point of view, these suggestions should not prevent the paper from moving to the next phase (i.e., getting accepted to be published).

Here are a few comments/suggestions:

In **2.7 Measuring the quality of imputation** Section → **C. Sliced Wasserstein distance** → it is said: “Secondly, high-dimensional (complete) datasets are very sparse in the space R^d unless there are an unrealistically large number of samples.” → [comment] Is it only the sparsity that matters, or does the density also matter?

In **3 Results** Section → **Comparing imputation quality** → *B. Feature distribution metrics* → it is said: “MICE imputation, which performed worst by the sample-wise discrepancy scores, is the best-performing method by the Kolmogorov-Smirnoff statistic and Wasserstein distance for all missingness rates across all the minimum, median and maximum discrepancies.” → [comment] While this is true, when doing this sort of comparisons, one should also consider the magnitude of the differences. Although MICE imputation performs best by the Kolmogorov-Smirnoff statistic and Wasserstein distance, the difference to MissForest is minimal when compared to the counterpart differences by the sample-wise discrepancy scores.

In **3 Results** Section → **Comparing imputation quality** → *C. Sliced Wasserstein distribution metrics*. → it is said: “For the MIMIC-III dataset, over all discrepancy scores and missingness rates, the MICE imputation method shows a clear dominance with the mean and GAIN methods performing poorly.” → [comment] Again, this is true! However, one should also put into perspective the magnitude of the differences in what is being measured using the different metrics.

Also, something that I am missing from the very beginning – I apologize if I am pointing out this only now – is the scale of what is being measured. In other words, it would be extremely helpful to know for each plot which is the minimum possible value (I would say, zero) and the maximum (which is not clear). Having these values in mind would, probably, give a different perspective on the achieved results. For instance, in Figure 7, it is obvious that MICE outperforms the other imputation methods, but the question is: is that result negligible? The same goes for other comparisons.

Then it is said: “For the Simulated (N) dataset, the MICE method again performs the best overall by all measures ...” → [comment] This is not true, see first row of Figure 37.

Also, in the same sentence is said: “while for the Simulated (N,C) dataset, MICE and MIWAE are the best overall.” → [comment] I suggest changing the formulation and saying “in general”. Furthermore, this is another example where, typically, the differences seem to be neglectable.

A little bit after it is said: “For MIMIC-III and Simulated (N), mean imputation performs the worst, with GAIN and MissForest performing similarly poorly whilst for Simulated (N,C) all these methods perform equally poorly.” → [comment] I Would be more cautious with the

adjectives since I do not observe differences that justify the adjective poorly, at least not for MissForest.

I also suggest the authors fix the following typos and/or sentences:

- At the end of **1 Introduction** Section, after the GitHub URL → [suggestion] A comma is missing, and the rest of the sentence reads a bit weird.
- In **2.6 ANOVA analysis** Section: "... named in §2.4. Firstly, ..." → [suggestion] A space is missing.
- In **2.7 Measuring the quality of imputation** Section:
 - "... we must define a score that has the desirable property that it has a low value when the distribution of imputed values closely resembles that of the true values." → [suggestion] "... we must define a desirable score's property that achieves a low value when the distribution of imputed values closely resembles that of the true values."
 - "Explicitly, our aim is to compute a distance between the original samples $D = \{x_i\}_{i=1}^N$ and the imputed samples $\hat{D} = \{\hat{x}_i\}_{i=1}^N$ that will indicate the quality of the imputation." → [suggestion] "... that gives the quality of the imputation."
 - "The class of measures which would be of most practical value to practitioners is (C) measures of discrepancy for imputed and true data across the whole data distribution." → [suggestion] "However, we strongly believe that the class of measures which would be of most practical value to practitioners ...".
 - "**B. Feature-wise distribution discrepancy.** In the literature, some authors have considered discrepancy measures to quantify for how faithfully the distributions of individual features are reconstructed." → [suggestion] Delete "for".
 - "**C. Sliced Wasserstein distance.** In Figure 1, we see that simply considering ..." → [suggestion] Replace "see" with "show".
 - "Step1: ... We set $P = 10$ throughout" → [suggestion] Delete "throughout" and end with a dot (i.e., a period), at least put the dot there.
- In **3 Results** Section:
 - **Comparing imputation quality**
 - *A. Sample-wise statistics.*
 - "In Figure 5 and Supplementary Figures 20–22 the sample-wise ..." → [suggestion] A comma is missing after "Figures 20-22".
 - "**Simulated (N)** and **Simulated(N,C)**" → [suggestion] A space is missing.
 - *B. Feature distribution metrics.*
 - "In Figure 6 and Supplementary Figures 23–33" → [suggestion] Again, a comma is missing after "Figures 23-33".
 - "In Figure 6 and Supplementary Figures 23–33 we show the minimum, median and maximum feature-wise discrepancies statistics for the MIMIC-III, Simulated (N) and Simulated (N,C) datasets. The mean imputation method is the worst in all metrics for minimum, median and maximum at all rates of train

and test missingness in both datasets.” → [suggestion] Not “both” but “all”.

- Also, “The mean imputation method is the worst in all metrics for minimum, median and maximum at all rates of train and test missingness ...” → [suggestion] “In general, the mean imputation method...” because there is a few cases in which it is not the worst.
 - *C. Sliced Wasserstein distribution metrics.*
 - “At the 25% test missingness rate, the MIWAE imputation method is competitive with and sometimes outperforms, MICE.” → [suggestion] Delete the comma after “outperforms”.
- **In 4 Discussion Section:**
 - “In fact, for MSE in particular, ...” → [suggestion] MAE.
 - [suggestion] Please, change accordingly onwards that point.
 - “Secondly, we do not aim to provide a fully exhaustive empirical analysis all imputation methods and classifiers as the aim of this manuscript is to draw the reader’s attention to the importance of measuring imputation quality before fitting models.” → [suggestion] “... a fully exhaustive empirical analysis of all imputation methods ...”.

Title: Classification of datasets with imputed missing values: does imputation quality matter?

Responses to the reviewers for the revised manuscript

Reviewer #2:

I am pleased to notice how much the paper has improved! Although I am providing here a few comments, from my point of view, these suggestions should not prevent the paper from moving to the next phase (i.e., getting accepted to be published).

Here are a few comments/suggestions:

In 2.7 Measuring the quality of imputation Section C. Sliced Wasserstein distance it is said: “Secondly, high-dimensional (complete) datasets are very sparse in the space R^d unless there are an unrealistically large number of samples.” [comment] Is it only the sparsity that matters, or does the density also matter?

Thank you for this comment, I have updated the paper to mention sparsity and density are both responsible.

In 3 Results Section Comparing imputation quality B. Feature distribution metrics it is said: “MICE imputation, which performed worst by the sample-wise discrepancy scores, is the best-performing method by the Kolmogorov-Smirnoff statistic and Wasserstein distance for all missingness rates across all the minimum, median and maximum discrepancies.” [comment] While this is true, when doing this sort of comparison, one should also consider the magnitude of the differences. Although MICE imputation performs best by the Kolmogorov-Smirnoff statistic and Wasserstein distance, the difference to MissForest is minimal when compared to the counterpart differences by the sample-wise discrepancy scores.

Thank you for this point. I have added a sentence to mention that MissForest does best for RMSE and MAE, and then is competitive for the feature-wise discrepancy scores.

In 3 Results Section Comparing imputation quality C. Sliced Wasserstein distribution metrics. it is said: “For the MIMIC-III dataset, over all discrepancy scores and missingness rates, the MICE imputation method shows a clear dominance with the mean and GAIN methods performing poorly.” [comment] Again, this is true! However, one should also put into perspective the magnitude of the differences in what is being measured using the different metrics.

Thanks for this comment. I have added a part to the sentence to say that MissForest and MIWAE are competitive with MICE.

Also, something that I am missing from the very beginning – I apologize if I am pointing out this only now – is the scale of what is being measured. In other words, it would be extremely helpful to know for each plot which is the minimum possible value (I would say, zero) and the maximum (which is not clear). Having these values in mind would, probably, give a different perspective on the achieved results. For instance, in Figure 7, it is obvious that MICE outperforms the other imputation methods, but the question is: is that result negligible? The same goes for other comparisons.

Thank you for this comment, it is an important question but I believe we may have mostly addressed this in our ratio work in Supp Figures 32–34. Here we see that the ‘induced’ distance for MICE is similar to the natural distance observed for the real data, with a ratio around 1 for the **Simulated (N)** and **MIMIC-III** datasets. For MissForest and MIWAE, this ratio hovers a little higher, around a factor of 2 in most instances. GAIN and Mean induce a high multiplier on the distance between the imputed data and the natural variability.

Then it is said: “For the Simulated (N) dataset, the MICE method again performs the best overall by all measures . . .” [comment] This is not true, see first row of Figure 37.

Thank you for highlighting this. I’ve updated the text to mention that relative performance is less clear for 50% train missingness.

Also, in the same sentence is said: “while for the Simulated (N,C) dataset, MICE and MIWAE are the best overall.” [comment] I suggest changing the formulation and saying “in general”. Furthermore, this is another example where, typically, the differences

seem to be neglectable.

Thanks for this comment, I have updated the text as suggested.

A little bit after it is said: “For MIMIC-III and Simulated (N), mean imputation performs the worst, with GAIN and MissForest performing similarly poorly whilst for Simulated (N,C) all these methods perform equally poorly.” [comment] I Would be more cautious with the adjectives since I do not observe differences that justify the adjective poorly, at least not for MissForest.

Thanks for this comment. As suggested I have reworded the text to say that GAIN generally performs similarly poorly. The MissForest performance seems erratic and often poor, but I agree with the reviewers comment as MIWAE gives similarly erratic behaviour.

I also suggest the authors fix the following typos and/or sentences: • At the end of 1 Introduction Section, after the GitHub URL [suggestion] A comma is missing, and the rest of the sentence reads a bit weird.

Thank you for this comment. I have split the sentence in two after the URL.

• In 2.6 ANOVA analysis Section: “. . . named in §2.4.Firstly, . . .” [suggestion] A space is missing.

Thank you for bringing this typo to our attention, it is now corrected.

• In 2.7 Measuring the quality of imputation Section: o “. . . we must define a score that has the desirable property that it has a low value when the distribution of imputed values closely resembles that of the true values.” [suggestion] “. . . we must define a desirable score’s property that achieves a low value when the distribution of imputed values closely resembles that of the true values.”.

Thank you for this comment. I have updated the text based on the reviewer’s suggestion.

o “Explicitly, our aim is to compute a distance between the original samples $D = x_i^{N_i=1}$ and the imputed samples $D^\wedge = x^\wedge_i^{N_i=1}$ that will indicate the quality of the imputation.” [suggestion] “. . . that gives the quality of the imputation.”.

Thank you for this comment. I have updated the text based on the reviewer’s suggestion.

o “The class of measures which would be of most practical value to practitioners is (C) measures of discrepancy for imputed and true data across the whole data distribution.” [suggestion] “However, we strongly believe that the class of measures which would be of most practical value to practitioners . . .”.

Thank you for this comment. I have updated the text based on the reviewer’s suggestion.

o “B. Feature-wise distribution discrepancy. In the literature, some authors have considered discrepancy measures to quantify for how faithfully the distributions of individual features are reconstructed.” [suggestion] Delete “for”.

Thank you for bringing this typo to our attention, it is now corrected.

o “C. Sliced Wasserstein distance. In Figure 1, we see that simply considering . . .” [suggestion] Replace “see” with “show”.

Thank you for bringing this typo to our attention, it is now corrected.

o “Step1: . . . We set $P = 10$ throughout” [suggestion] Delete “throughout” and end with a dot (i.e., a period), at least put the dot there.

Thank you for bringing this typo to our attention, it is now corrected.

• In 3 Results Section: o Comparing imputation quality A. Sample-wise statistics. • “In Figure 5 and Supplementary Figures 20–22 the sample-wise . . .” [suggestion] A comma is missing after “Figures 20-22”.

Thank you for bringing this typo to our attention, it is now corrected.

- “Simulated (N)and Simulated(N,C)” [suggestion] A space is missing.

Thank you for bringing this typo to our attention, it is now corrected.

B. Feature distribution metrics. • “In Figure 6 and Supplementary Figures 23–33” [suggestion] Again, a comma is missing after “Figures 23-33”.

Thank you for bringing this typo to our attention, it is now corrected.

- “In Figure 6 and Supplementary Figures 23–33 we show the minimum, median and maximum feature-wise discrepancies statistics for the MIMIC-III, Simulated (N) and Simulated (N,C) datasets. The mean imputation method is the worst in all metrics for minimum, median and maximum at all rates of train and test missingness in both datasets.” [suggestion] Not “both” but “all”.

Thank you for bringing this typo to our attention, it is now corrected.

- Also, “The mean imputation method is the worst in all metrics for minimum, median and maximum at all rates of train and test missingness . . .” [suggestion] “In general, the mean imputation method. . .” because there is a few cases in which it is not the worst.

Thank you for this comment. I have updated the text based on the reviewer’s suggestion.

C. Sliced Wasserstein distribution metrics. • “At the 25% test missingness rate, the MIWAE imputation method is competitive with and sometimes outperforms, MICE.” [suggestion] Delete the comma after “outperforms”.

Thanks for this comment, I have restructured the sentence to remove the comma issue.

- In 4 Discussion Section: o “In fact, for MSE in particular, . . .” [suggestion] MAE.

Thank you for the comment. I believe this is correct as MSE, as we show in Supplementary Figure 1 that MSE is optimised for this imputation method but the distribution is poorly recreated. I have reworded slightly to “In fact, for MSE, we have seen that it takes an optimal value for imputations that give a very poor distribution match.”

- o [suggestion] Please, change accordingly onwards that point.

It is not clear to me what is being requested with this point. Please clarify and I would be happy to address,

- o “Secondly, we do not aim to provide a fully exhaustive empirical analysis all imputation methods and classifiers as the aim of this manuscript is to draw the reader’s attention to the importance of measuring imputation quality before fitting models.” [suggestion] “. . . a fully exhaustive empirical analysis of all imputation methods . . .”.

Thank you for bringing this typo to our attention, it is now corrected.